# Optimizing Ammonia Emissions for PM<sub>2.5</sub> Mitigation:

1

15

17

### **Environmental and Health Co-Benefits in Eastern China**

3 Keqin Tang <sup>1</sup>, Haoran Zhang <sup>2</sup>, Ge Xu <sup>1</sup>, Fengyi Chang <sup>1</sup>, Yang Xu <sup>1</sup>, Ji Miao <sup>3</sup>, Xian 4 Cui 4, Jianbin Jin 1, Baojie Li 1, Ke Li 1, Hong Liao 1, Nan Li 1,\* 5 6 <sup>1</sup> Jiangsu Key Laboratory of Atmospheric Environment Monitoring and Pollution 7 Control, Jiangsu Collaborative Innovation Center of Atmospheric Environment and 8 9 Equipment Technology, School of Environmental Science and Engineering, Nanjing University of Information Science & Technology, Nanjing, 210044, China 10 <sup>2</sup> School of Atmospheric Sciences, Nanjing University, Nanjing, China 11 <sup>3</sup> Department of Colorectal Surgery, Nanjing Drum Tower Hospital, Affiliated Hospital 12 13 of Medical School, Nanjing University, Nanjing, China

- <sup>4</sup> Rugao Hospital of Chinese Medicine, Nantong, China
- \* Correspondence to: Nan Li, <u>linan@nuist.edu.cn</u>

#### Abstract.

Ammonia (NH<sub>3</sub>) is a key precursor of PM<sub>2.5</sub>, contributing to the formation of secondary inorganic aerosols and playing a crucial role in haze events. However, current bottomup emission inventories in China often underestimate NH<sub>3</sub> emissions, particularly with significant uncertainties in urban areas. This study developed a "top-down" iterative algorithm that integrates the IASI satellite observations with the WRF-Chem model to optimize bottom-up NH<sub>3</sub> emissions, and further quantified the impacts of sourcespecific emission reductions on PM<sub>2.5</sub> pollution. The result reveals that the updated NH<sub>3</sub> emissions in Eastern China for 2016 amounted to 4.2 Tg yr<sup>-1</sup>, 27.3% higher than prior estimations. The optimized NH<sub>3</sub> emissions peak in summer at 463.1 Gg month<sup>-1</sup>, with agricultural sources accounting for 85%, while winter emissions drop to 217 Gg month <sup>-1</sup> when the contribution from non-agricultural sources (e.g., industry, vehicle) significantly increases. The optimized NH<sub>3</sub> emission significantly improved the simulation of both total column and surface NH<sub>3</sub> concentrations, with improvements in magnitude (31%–42%) and variations (17%–55%). Sensitivity simulations show that a 30%–60% reduction in NH<sub>3</sub> emission led to decreases of 1.5–8.8 µg·m<sup>-3</sup> in city-level PM<sub>2.5</sub> concentrations and the potential effect of reducing non-agricultural emissions is comparable with that from agricultural sources. Furthermore, the NH3 reduction positively impacts public health, resulting in a 6.5%-10.3% decrease in premature deaths attributed to PM<sub>2.5</sub> exposure. Our study evaluated NH<sub>3</sub> emissions from various sources in Eastern China, emphasizing the impact of reducing non-agricultural ammonia emissions on air quality and public health benefits.

**Keywords:** NH<sub>3</sub> emission, PM<sub>2.5</sub>, satellite retrieval, WRF-Chem, top-down

42

### 1 Introduction

In recent years, China has continued to face significant challenges associated with PM<sub>2.5</sub> pollution (Geng et al., 2024; Lei et al., 2022). This issue adversely affects 45 atmospheric environment via reducing visibility (Hu et al., 2021; Yang et al., 2022) and 46 47 deteriorating air quality (Lei et al., 2024; Song et al., 2025), impacts climate change by altering radiation balance(Tang et al., 2025) and cloud formation (Gao et al., 2023; Yang 48 49 et al., 2021), and poses substantial threats to human health (Du et al., 2024; Feng et al., 2016; Liu et al., 2025; Xiao et al., 2022; Zhu et al., 2025). Ammonia (NH<sub>3</sub>), a key 50 precursor of PM<sub>2.5</sub>, neutralizes sulfuric acid (H<sub>2</sub>SO<sub>4</sub>) and nitric acid (HNO<sub>3</sub>), leading to 51 the formation of secondary inorganic aerosols (SIA), which contributes 19.4%–55.0% 52 of the total PM<sub>2.5</sub> (Huang et al., 2014; Liu et al., 2022b; Wang et al., 2016; Wei et al., 53 2023; Zheng et al., 2015; Zhou et al., 2022). Reducing NH<sub>3</sub> emissions is a highly 54 effective strategy for mitigation of PM<sub>2.5</sub> pollution (Bessagnet et al., 2014; Xu et al., 55 2022), particularly in light of the successful control of sulfur dioxide (SO<sub>2</sub>) and nitrogen 56 57 dioxide (NO<sub>2</sub>) in China over the past decade (Li et al., 2023b; Wang et al., 2017; Zhang et al., 2019; Zheng et al., 2018). 58 59 The anthropogenic sources of NH<sub>3</sub> include agriculture, industry, power generation, transportation and residential activities. Numerous studies have estimated NH<sub>3</sub> 60 emissions using a bottom-up approach, reporting emissions in China ranging from 9.7 61 Tg yr<sup>-1</sup> to 13.2 Tg yr<sup>-1</sup> (Chen et al., 2021; Huang et al., 2012; Kang et al., 2016; Li et 62 al., 2021; Ma, 2020). Among these sources, the agricultural (AGR) sector is identified 63 as the dominant contributor nationwide, accounting for 75.0%-94.5% of total NH<sub>3</sub> 64 emissions (Guo et al., 2020; Ma, 2020; Zhou et al., 2021). Additionally, some studies 65 have highlighted that in densely populated regions, NH<sub>3</sub> from non-agricultural (non-66 AGR) activities, such as industrial production/slip, vehicles, and waste disposal, 67 contributing up to 50% of regional emissions and should not be overlooked (Chang et 68 al., 2015, 2016; Chen et al., 2022; Feng et al., 2022; Pan et al., 2016, 2018b; Pu et al., 69 2020; Song et al., 2021; Sun et al., 2017; Van Damme et al., 2018; Wu et al., 2020). 70 However, despite considerable progress, bottom-up estimates still exhibit considerable 71

discrepancies and are often outdated, with a time lag of 1–2 years, mainly due to the lack of accurate and timely statistical data.

The uncertainty in the emission estimation further contributes to significant discrepancies, reflecting the range of results (1%–50%) reported in the literature, in assessing the impacts of NH<sub>3</sub> reduction on PM<sub>2.5</sub> level (Guo et al., 2018, 2024; Li et al., 2024; Liu et al., 2019, 2021, 2023; Pan et al., 2024; Zhang et al., 2022). Cheng et al (2021) employed WRF-Chem simulations to demonstrate a 24.6% reduction in PM<sub>2.5</sub> from the removal of AGR NH<sub>3</sub> emissions. Concurrently, Ti et al. (2022) determined that a 74% decrease in AGR NH<sub>3</sub> resulted in a 34.9% reduction in PM<sub>2.5</sub> in China.

To enhance the accuracy and reliability of bottom-up emission estimations, air quality monitoring satellites are increasingly regarded as valuable tools from a top-down perspective, offering advantages in both magnitude and timeliness (Chen et al., 2025, 2021; Guo et al., 2020; Jin et al., 2023; Qi et al., 2017; Xia et al., 2025; Zhou et al., 2021, 2017). Many studies have estimated optimized NH<sub>3</sub> emissions in China to be between 10.0 Tg yr<sup>-1</sup> and 18.9 Tg yr<sup>-1</sup> by coupling chemical transport models, mass balance approaches, or machine learning techniques with various NH<sub>3</sub> measurements (satellite retrieval or ground monitoring). Some studies have also improved the description of the spatial and monthly variations of NH<sub>3</sub> emissions (Kong et al., 2019; Liu et al., 2022a; Paulot et al., 2014; Zhang et al., 2018, 2017). However, most top-down studies lack further investigation into the source-specific allocation of emissions based on the optimal total emission assessment (Fu et al., 2015; Sun et al., 2017; Zhang et al., 2024). Hence, a more comprehensive understanding of NH<sub>3</sub> emissions from diverse sources across varying seasons is needed to improve existing top-down inventories and enhance the scientific accuracy of NH<sub>3</sub> emission reduction assessments.

In this study, we used satellite and surface NH<sub>3</sub> measurements alongside the regional chemical model WRF-Chem to constrain bottom-up and source-specific NH<sub>3</sub> emission estimates over Eastern China, with the aim of more accurately assessing the impacts of NH<sub>3</sub> emission reductions from different sources on PM<sub>2.5</sub> concentrations. The paper is structured as follows: Section 2 describes the detailed methodology, Section 3 presents the simulated NH<sub>3</sub> with prior emission, Section 4 provides a top-

down estimate of NH<sub>3</sub> emissions, and Section 5 demonstrates the direct correlation between NH<sub>3</sub> emission reductions and PM<sub>2.5</sub> concentration levels, as well as the associated health benefits. Our work differs from previous studies in that we constrain NH<sub>3</sub> emissions by sector, season, and region, and further assess the potential mitigation effects of NH<sub>3</sub> based on the optimized NH<sub>3</sub> inventory.

In this study, the chemical transport model WRF-Chem v3.9.1 (Grell et al. 2005)

# 2 Methodology

#### 2.1 Air Quality Model

was utilized to constrain the NH<sub>3</sub> emissions and to assess the impact of reduced NH<sub>3</sub> emission on PM<sub>2.5</sub> concentrations. Spatially, two nested domains were configured with horizontal resolutions of  $54 \times 54 \text{ km}^2$  and  $18 \times 18 \text{ km}^2$ . The outer domain covered entire China and the inner domain focused on Eastern China, characterized by intensive anthropogenic activities and elevated pollution levels (Pendergrass et al., 2025; Peng et al., 2025), including the Beijing-Tianjin-Hebei (BTH) region, Henan, Shandong, and the Yangtze River Delta (YRD) region (Figure 1). The initial and boundary conditions of meteorological parameters were derived from FNL reanalysis datasets provided by the National Centers for Environmental Prediction (NCEP) of the United States (https://rda.ucar.edu/datasets/). The initial and boundary conditions of chemical species were obtained from the global chemical transport model MOZART (Emmons et al. 2010). We conducted simulations for the entire year of 2016. The physical and chemical parameterizations describing sub-grid processes, such as radiation, microphysics, and gas-phase reaction schemes, are listed in Table S1. We adopted the anthropogenic emissions from the Multi-resolution Emission Inventory for China (MEIC, version 1.3) developed by Tsinghua University (Li et al., 2017; Zheng et al., 2018). Furthermore, biogenic emissions were calculated online using the Model of Emissions of Gases and Aerosols from Nature (MEGAN, version 2.0.4) (Guenther, 2006). Our numerical simulations also incorporated offline biomass burning emissions of various air pollutants, based on the wildfire model Fire Inventory

#### 2.2 Satellite retrievals and surface measurements

We obtained the total column density of NH<sub>3</sub> from the passive satellite remote-sensing product of the Infrared Atmospheric Sounding Interferometer (IASI) (version 3.0, https://iasi.aeris-data.fr/nh<sub>3</sub>/, last accessed on December 2020) as the observational constraint. The IASI is a Fourier transform spectrometer on board the Metop series of meteorological satellites, which circle the Earth in a polar Sun-synchronous orbit (Van Damme et al., 2014). Consequently, the satellite-based IASI instrument can cover the entire globe and provide measurements twice a day at 09:30 and 21:30 local solar time. The IASI instrument detects infrared radiation in the spectral range from 645 to 2760 cm<sup>-1</sup> emitted by Earth's surface and atmosphere with a 12 km circular footprint at nadir. This radiation absorption range includes the NH<sub>3</sub> signal near 950 cm<sup>-1</sup>.

The daily NH<sub>3</sub> column concentrations are categorized into level-2 satellite data and are developed based on the ANNI-NH<sub>3</sub> inversion algorithm without averaging kernels, as presented by Van Damme et al. (2017). Specifically, their retrieval algorithm derives hyperspectral radiation indexes (HRI) from the direct satellite spectrum detection, which is then converted into final NH<sub>3</sub> column concentrations using an artificial neural network technique (Whitburn et al., 2016). For better data quality, the present study removed NH<sub>3</sub> column concentrations associated with cloud cover of more than 10%. Furthermore, we preprocessed the IASI NH<sub>3</sub> column concentration data through averaging all daily values to obtain a monthly mean value. Spatially, we mapped the original satellite product data to the grid cells of the WRF-Chem model for further comparison with those simulated NH<sub>3</sub> columns.

In addition, surface in-situ NH<sub>3</sub> measurements reported by Pan et al. (2018a) were collected for model evaluation. These ground-based measurements were summarized into the seasonal mean concentrations of NH<sub>3</sub> at 53 sites in China from September 2015 to August 2016.

Additionally, surface meteorological data, including air temperature, relative

humidity and wind speed was obtained from China Meteorological Administration website (https://data.cma.cn/) to assess the meteorological simulations over the study region. Air pollutant concentrations associated with NH<sub>3</sub> (such as PM<sub>2.5</sub>, NO<sub>2</sub> and SO<sub>2</sub>) from public website of the Ministry of Ecology and Environment (MEE) of China (https://air.cnemc.cn:18007/) were also derived for evaluation. Furthermore, speciated inorganic aerosol data from a representative site in Beijing were collected to evaluate the model's capacity in characterizing the formation of secondary inorganic aerosols (Tan et al., 2018). The complete information of the in-situ measurements used in this study is available in Tables S2~S4.

## 3 NH<sub>3</sub> simulations with bottom-up emissions

We applied the bottom-up NH<sub>3</sub> emissions from MEIC (Li et al., 2017; Zheng et al., 2018) to drive the prior simulation. As shown in Figure 2, the prior NH<sub>3</sub> emission amounted to 3.3 Tg yr<sup>-1</sup> in Eastern China, among which 93.0% emission is from AGR sources and the other 7.0% emission is from non-AGR sources. The largest emissions are recorded in July at 366.8 Gg month <sup>-1</sup>, while the smallest emissions are recorded in January at 206.5 Gg month <sup>-1</sup> (Figure S1).

We compared the prior model results with IASI NH<sub>3</sub> column concentration and surface NH<sub>3</sub> volume concentration observations. The detailed method for calculating NH<sub>3</sub> total column concentrations and surface volume concentrations from WRF-Chem is provided in Text S1.

To quantitatively describe model performance, we adopted three statistical metrics, including root mean squared error (RMSE,  $0 \sim +\infty$ ), index of agreement (IOA,  $0 \sim 1$ ) and mean fractional bias (MFB,  $-2 \sim 2$ ) (Huang et al., 2021). The IOA quantifies the overall model skill, where a value of 1 indicates a perfect match and 0 denotes complete disagreement. The MFB diagnoses systematic model bias, where positive values indicate overestimation, negative values indicate underestimation, and 0 signifies no average bias. The RMSE represents the average model error in the same units as the variable under evaluation, with lower values indicating better performance. They were

calculated following Eq.  $1\sim3$ , where C represents the concentration of the target 189 pollutant (e.g., NH<sub>3</sub> total column or surface concentrations), and subscripts s, o and N 190 represent simulations, observations, and the number of samples, respectively.

$$RMSE = \sqrt{\frac{\sum_{i=1}^{N} (c_m - c_o)^2}{N}} \quad (1)$$
$$IOA = 1 - \frac{\sum_{i=1}^{N} (c_s - c_o)^2}{\sum_{i=1}^{N} (|c_s - \overline{c_o}| + |c_o - \overline{c_o}|)^2} \quad (2)$$

$$MFB = \frac{1}{N} \sum_{i=1}^{N} \frac{(C_o - C_m)}{(\frac{C_o + C_m}{2})}$$
 (3)

As shown in Table S5, the annual average of NH3 total column concentrations is simulated to be 17.4×10<sup>15</sup> molec cm<sup>-2</sup> for Eastern China, with a 61% underestimation of MFB compared to the observations from IASI satellite retrievals (29.0×10<sup>15</sup> molec cm<sup>-2</sup>). The IOA between observations versus simulations is 0.72. The seasonal simulations of NH<sub>3</sub> concentrations also exhibit significant discrepancies with observations, especially in spring. Specifically, the simulated NH<sub>3</sub> total column concentration in Eastern China is only 13.2×1015 molec cm-2 in spring, with concentration in 67.5% of the study region being underestimated by more than 50%. These discrepancies are evidently exhibited in Figure 3. Most simulated NH<sub>3</sub> total column concentrations are underestimated by more than 30% compared with the observed values by satellite with the associated RMSE exceeding 10×10<sup>15</sup> molec cm<sup>-2</sup>.

As illustrated in Figure 6, satellite-based observations reveal that the spatial highvalue areas of NH<sub>3</sub> column are located at the junction of Henan, Shandong, and Hebei provinces. In contrast, the prior modeling results show that NH<sub>3</sub> column densities are more concentrated in Henan. This indicates a clear discrepancy in the spatial distribution of NH<sub>3</sub> column densities between the prior simulations and the observations.

Additionally, the comparison between the simulated and observed surface NH<sub>3</sub> volume concentrations also indicates a notable underestimation (Figure S2). The mean simulated surface NH<sub>3</sub> volume concentration over the study region is 6.3 µg m<sup>-3</sup>, which is only half of the observation value (12.7 µg m<sup>-3</sup>), with an IOA of 0.57 and an MFB of -61%, respectively (Table S5).

### 4 Top-down estimates of NH<sub>3</sub> emissions

### 4.1 Iterative algorithm for NH<sub>3</sub> emission estimation

We utilized an iterative algorithm (Figure 4) to update the prior NH<sub>3</sub> emissions from different sources constrained by IASI observations. This process was carried out in January, April, July, and October in 2016 to represent four seasons. The posterior emission inventory derived for each representative month was then applied to all three months within its corresponding season to generate the full 12-month posterior inventory. This representative-month approach was adopted to allow for a robust validation against the full 12-month period, with the remaining eight months serving as an independent dataset, and to manage the substantial computational cost of the iterative process. We compared the prior simulation results with satellite retrievals and discussed the performance of prior emissions in detail in Section 3. Furthermore, we conducted a series of sensitivity simulations to obtain prior simulated NH<sub>3</sub> from disparate sources and which were then fed into the iterative algorithm along with satellite data for calculation. In each iterative calculation, the monthly average satellite-derived NH<sub>3</sub> column concentration served as the target, and multiple linear regression (MLR) was applied to calculate the corresponding regression factors for AGR and non-AGR emissions (Figure S3). This separation of sectors by MLR is effective because their respective spatial distributions are distinct and largely uncorrelated (r = 0.35). Here, we take the *i* iteration in *k* month, *j* region as an example to calculate the regression factors, and the formula is as follows:  $TA_{\text{satellite}}^{j,k} - SA_{\text{transport}}^{j,k} = \alpha_i^{j,k} * SA_{\text{agriculture}}_{i-1}^{j,k} + \beta_i^{j,k} * SA_{\text{non-agriculture}}_{i-1}^{j,k}$ (4) where, TA<sub>satellite</sub><sup>j,k</sup> denotes the monthly average of total NH<sub>3</sub> column density retrieved from the IASI satellite data, and  $SA_{transport}^{j,k}$ ,  $SA_{agriculture}^{j,k}_{i-1}$  and  $SA_{non-agriculture}_{i-1}^{j,k}$  stand for the simulated total column concentration of  $NH_3$ contributed by outside transport, AGR emissions, and non-AGR emissions, respectively.

We clarified this NH<sub>3</sub> concentrations contributed by different pathways by conducting

sensitivity experiments with the WRF-Chem model (Table 1).

In each experiment, we zeroed out AGR emissions, non-AGR emissions and regional external emissions to obtain the corresponding NH<sub>3</sub> column concentrations. The  $SA_{agriculture}_{i-1}^{j,k}$ ,  $SA_{non-agriculture}_{i-1}^{j,k}$ , and  $SA_{transport}^{j,k}$  are calculated by subtracting  $A_{blank}$  from  $A_{agr}$ ,  $A_{non-agr}$ , and  $A_{transport}$ , respectively. Here, symbols A represent the total simulated NH<sub>3</sub> column concentrations that result from each of the sensitivity simulations listed in Table 1. Specifically, the modeling case  $A_{blank}$  refers to a simulated NH<sub>3</sub> total column in which all anthropogenic emissions within the study domain were zeroed out. The purpose of this simulation was to establish background concentrations, which represents the influence of the chemical boundary conditions provided to our model domain.

Furthermore, the MLR approach provided regression coefficients  $\alpha_i^{j,k}$  and  $\beta_i^{j,k}$ , which function as scaling factors, respectively correspond to AGR and non-AGR NH<sub>3</sub> emissions in month j from region k, within the i iteration. To ensure the statistical robustness of the regression equation, we need to correct for this regression coefficient. The biases between the model simulation and the satellite retrievals were calculated as  $D_{i}^{j,k}$ . Specifically, it is the difference between the mean simulated column and the mean satellite retrieval, divided by the mean satellite retrieval. We considered the residuals of the MLR approach, the goodness of fit and  $D_i^{j,k}$ , and obtained the judgment coefficient  $K_i^{j,k}$ . The regression coefficients with excessive residuals, defined as cases where the 95% confidence interval of the residual does not contain zero, are removed to increase credibility. Concurrently, the goodness of fit of the regression is calculated as the coefficient of determination (R-square, R<sup>2</sup>). To maintain algorithm stability, regressions with an R<sup>2</sup> less than 0.3 are deemed invalid and excluded from the emission update, as they exhibit insufficient explanatory power (indicating >70% unexplained variance) and introduce destabilizing noise into the adjustments. We further use it to make a trade-off for the regression coefficient. If a regression is valid, the adjustment factors a and b are set to the new regression coefficients; if invalid, the factors are kept unchanged from the previous iteration. The updated emissions for the next iteration are then calculated by multiplying the emissions from the previous step by these adjustment factors. Finally, the entire process is iteratively repeated, a framework that captures the overall non-linear atmospheric response by combining the dynamic simulation of non-linear chemistry within each WRF-Chem step with the collective behavior of multiple iterations. The iteration concludes when the mean bias between the simulated values and observations is less than 30%, a criterion chosen to represent a significant improvement over the large prior bias while falling within the range of widely accepted model performance benchmarks.

## 4.2 posterior NH<sub>3</sub> emission estimates

The top-down constrained results (posterior) indicate that the annual NH<sub>3</sub> emission in Eastern China has been updated to 4.2 Tg yr<sup>-1</sup>, representing a 27.3% increase compared to the prior value (Figure 2). The posterior AGR emissions increased slightly, from 3.0 Tg yr<sup>-1</sup> to 3.1 Tg yr<sup>-1</sup>, but the high-emission regions shift from Henan to Shandong, Jiangsu and northern Anhui (Ren et al., 2023). The posterior non-AGR emissions show a significant increase, from 0.2 Tg yr<sup>-1</sup> to 1.1 Tg yr<sup>-1</sup>, particularly in urban regions along the Yangtze River, as well as in southern BTH, central Shandong and northern Henan (Figure S4). Analysis of emission inventories (An et al., 2021; Hoesly et al., 2018; Li et al., 2021, 2017; Ma, 2020; Wu et al., 2024) reveals that residential activities and waste disposal are dominant sources of non-AGR NH<sub>3</sub> emissions, particularly in densely populated regions (Figure S5). In multiple iterations, the framework optimizes the relative mix of the two sources to better match the observed spatial patterns. For instance, the spatial correlation between model and observation in Henan increased from 0.47–0.58 (prior simulations) to 0.64–0.90 (posterior simulations).

In terms of seasonality, as shown in Figure 5, the posterior NH<sub>3</sub> emissions are highest in summer, with a total of 463.1 Gg month<sup>-1</sup>, followed by spring (442.4 Gg month<sup>-1</sup>), largely due to fertilizer application (Li et al., 2021; Lu et al., 2025; Ren et al.,

2025), and lowest in winter (217.4 Gg month<sup>-1</sup>). The seasonal variations in the posterior emissions is the net result of complex adjustments in both the AGR and non-AGR sectors. At the specific-source scale (Figure S6), AGR NH<sub>3</sub> emissions show similar seasonal patterns with the total NH<sub>3</sub> emissions, higher in summer and spring. In contrast, non-AGR NH<sub>3</sub> are highest in winter and fall because fossil fuel combustion-related emissions are higher in cold season, while the lowest emissions occur in summer. In addition, the ratio of AGR and non-AGR NH<sub>3</sub> emissions significantly varies across different regions. The contribution of non-AGR NH<sub>3</sub> emissions range from 18.8% to 35.8%, which is higher than the proportion in the prior inventory (Figure 5a). This shift can be attributed to the increased relative importance of fossil fuel combustion-related emissions under high PM<sub>2.5</sub> loadings, which in turn promote higher NH<sub>3</sub> emissions from these sources (Pan et al., 2018b). Meanwhile, AGR NH<sub>3</sub> emissions are relatively inactive in winter due to unfavorable meteorological conditions. Similar high fractions of non-AGR emissions have also been reported in other studies (Feng et al., 2022; He et al., 2021).

Table 2 compares the results with related studies focused on NH<sub>3</sub> emission estimates. Overall, the estimated NH<sub>3</sub> emission in this study is comparable to the estimates of the other studies based on both "top-down" and "bottom-up" approaches. In similar years and regions, the discrepancy between the estimates of this study and other studies ranges from 1.0% to 19.6%. The slight discrepancy can be partially explained by our estimate being a conservative lower bound, a consequence of the residual gap remaining with satellite retrieval. Additionally, uncertainties from the model's chemical mechanisms and the influence of nearby grid transport also contribute to this gap, but the overall impact on the final estimate is limited. Furthermore, the seasonal distribution of NH<sub>3</sub> emissions in this study aligns with the findings of previous studies (Kong et al., 2019; Liu et al., 2024; Zhang et al., 2018; Zhao et al., 2020).

In terms of sectors, other studies have indicated that the contribution of NH<sub>3</sub> emissions from AGR sources is more than 80%, using the bottom-up approach (Chen et al., 2021; Huang et al., 2012; Kang et al., 2016; Li et al., 2021). The relatively small proportion of non-AGR emissions is likely due to overlooked industrial (e.g., NH<sub>3</sub> slip

and indirect emissions) (Chen and Wang, 2025; Chen et al., 2022; Wei et al., 2022) and residential sources (e.g., from waste) (Shao et al., 2020), combined with unrepresentative transportation emission factors (Sun et al., 2017; Zhang et al., 2021). This study, however, reveals a proportion of 74.4% for AGR emissions, thereby emphasizing the contribution of non-AGR emissions. Concurrently, the eastern developed industry is expected to exhibit an increase in the proportion of NH<sub>3</sub> emissions from non-AGR sources when compared to the national average. Our work attempts to quantitatively disentangle the emissions from AGR and non-AGR sectors directly within our top-down framework and facilitates a more comprehensive capture of neglected non-AGR sources.

It is important to note that discrepancies in results between studies may be attributable to methodological differences (e.g. the sensitivity of the top-down approach to target data selection) and uncertainty in the underlying data. For instance, the NH<sub>3</sub> emission estimated by Paulot et al. (2014) using the mass balance method based on ammonium wet deposition fluxes is significantly lower than that in other studies, which may be attributed to its fewer observation sites in China. These discrepancies underscore the necessity to enhance the reliability of NH<sub>3</sub> observations in forthcoming studies, with the objective of enhancing the precision of the estimates.

#### 4.3 Simulated NH<sub>3</sub> with top-down emissions

Figure 6 compares the spatial distributions of NH<sub>3</sub> total column density from satellite retrievals, prior simulations and posterior simulations. The annual mean simulated NH<sub>3</sub> total column density improved from the prior result of 17.4×10<sup>15</sup> molec cm<sup>-2</sup> to a posterior value of 23.7×10<sup>15</sup> molec cm<sup>-2</sup>, with an increase of 35.9%, and is closer to the observed value of 29.0×10<sup>15</sup> molec cm<sup>-2</sup>. IOA and MFB between the posterior simulations versus measurements are 0.9 and -30.0%, respectively. Figure 3 also shows the improvement in model performance. More than 80% of the points fall in the range where the simulation-to-observation ratio is between 0.7 and 1.3 and the RMSE is less than 10×10<sup>15</sup> molec cm<sup>-2</sup>. A more consistent seasonal distribution can be obtained in a posterior simulation, with associated temporal MFB of NH<sub>3</sub> column

density on the seasonal scale is reduced from -53% (prior) to -24% (posterior). Simultaneously, the spatial distribution pattern of posterior simulation is more identical to the characteristics revealed by satellite-based observations (Figure 6). The spatial MFB is also decreased from -52% (prior) to -20% (posterior), with an increase in spatial correlation coefficient from 0.79 to 0.92. The improvement is especially notable in the BTH region, where the simulated NH<sub>3</sub> column densities are doubled. In summary, the posterior simulation improves the agreement between the simulated NH<sub>3</sub> column concentrations and satellite observations in both overall magnitude and spatial distribution, although some deviations remain, particularly in the colder seasons. These can likely be attributed to methodological limitations, such as the inherent tolerance of our 30% iterative stopping criterion and potential inconsistencies from aggregating monthly optimizations to a seasonal scale.

A similar improvement is also witnessed in the modeling of surface NH<sub>3</sub> concentrations, which were evaluated against in-situ measurements from 13 sites reported by Pan et al. (2018a) for the 2015-2016 period (Table S2). The posterior simulation significantly improves the annual mean, increasing the surface concentration from 6.3 µg m<sup>-3</sup> (prior) to 9.4 µg m<sup>-3</sup> (posterior), much closer to the observed average of 12.7 µg m<sup>-3</sup>. As shown in the scatter plot in Figure S7, the posterior simulation alleviates the underestimation at most sites, which is quantified by a 42% reduction in the overall underestimation bias and a clear improvement in the IOA. On a seasonal basis, the posterior emissions also alleviate the large underestimation of the prior simulation across all seasons, though the degree of improvement varies (Table S6). The prior simulation showed significant underestimation in all seasons, with the MFB ranging from -0.37 in winter to -0.79 in spring. The posterior simulation demonstrates a particularly evident improvement in spring, where the MFB reduced from -0.79 to -0.24. While some underestimation remains in summer, the posterior results still show improved performance metrics (e.g., lower RMSE and higher IOA) for all seasons, confirming a better capture of the seasonal characteristics overall. The remaining discrepancy between the posterior simulation and surface observations can be attributed

to several factors, such as the spatial representativeness of the surface sites and the accuracy of the secondary inorganic aerosol simulation.

Furthermore, improving the NH<sub>3</sub> simulation results in the other simulated air pollutants being closer to observed levels (Table 3). Specifically, we compare the annual mean concentrations of PM<sub>2.5</sub>, SO<sub>2</sub>, and NO<sub>2</sub> from the prior and posterior simulations against surface observations averaged from 80 monitoring sites across 9 major cities (Table S4). It was found that posterior NH<sub>3</sub> emissions effectively bridge the gap between simulated and observed PM<sub>2.5</sub>. The average PM<sub>2.5</sub> concentration increased from 65.7 µg m<sup>-3</sup> to 67.3 µg m<sup>-3</sup>, which is closer to the observed value of 67.1 µg m<sup>-3</sup>. To further characterize the model's chemical performance beyond total PM<sub>2.5</sub>, we also evaluated the simulation of secondary inorganic aerosol (SIA) components against insitu measurements from a representative site in Beijing (Table S7). The evaluation shows that the posterior NH<sub>3</sub> emissions improved the simulation of ammonium and nitrate, reducing the bias between simulated and observed concentrations. Although the model underestimates sulfate, likely due to missing formation mechanisms (Cai et al., 2024; Wang et al., 2021, 2020), the total SIA concentration is well reproduced with an overall bias of only -11.0%. A similar improvement is also observed for SO<sub>2</sub>, where the posterior simulated concentration (6.8 ppbv) better matches the observed value (6.5 ppbv), reducing the model's previous overestimation by 27%. This improvement is most significant in autumn. The successful capture of air pollutants highlights a significant improvement in the NH<sub>3</sub> emission inventory for Eastern China. The evaluation of routine air pollutants in each city is detailed in Figures S8~S10. The statistics of evaluation metrics for each city's meteorological simulations can also be found in Table S8.

### 5 PM<sub>2.5</sub> and its health burden response to NH<sub>3</sub> reduction

To investigate the response of PM<sub>2.5</sub> to various NH<sub>3</sub> emission reduction scenarios, we conducted sensitivity experiments as outlined in Table S9. We formulated emission reduction scenarios of 30%–60% for January and July of 2016, considering the severe particulate pollution in winter and the higher NH<sub>3</sub> concentrations in summer. Emission

reductions from both the AGR and non-AGR sectors were considered separately.

Figure 7 illustrates that reducing NH<sub>3</sub> emissions by 30%–60% can decrease the seasonal PM<sub>2.5</sub> concentrations by 1.5–5.7 μg m<sup>-3</sup> (2.0%–7.2%) averaged for Eastern China in winter, mainly due to the reduction in SIA. Specifically, nitrate, ammonium and sulfate are reduced by 0.9–3.3 μg m<sup>-3</sup>, 0.4–1.3 μg m<sup>-3</sup> and 0.3–1.0 μg m<sup>-3</sup>, respectively. It is worth noting that the reduction in sulfate is smaller than that in nitrate because NH<sub>3</sub> preferentially reacts with sulfuric acid during aerosol formation (Figure S11). When ambient NH<sub>3</sub> concentrations are limited, nitrate concentrations decrease more significantly than sulfate concentrations. In summer, although aerosol pollution is relatively lower, NH<sub>3</sub> emissions and atmospheric reactivity are higher. Consequently, reducing emissions by the same percent results in a decrease in PM<sub>2.5</sub> concentration by 5.5–8.8 μg m<sup>-3</sup>.

In terms of special sources, reducing non-AGR NH<sub>3</sub> emissions is just as crucial as reducing AGR NH<sub>3</sub> emissions in mitigating PM<sub>2.5</sub>. A 30% to 60% reduction in non-AGR NH<sub>3</sub> emissions during winter can lead to a decrease in PM<sub>2.5</sub> by 0.9–1.5 μg m<sup>-3</sup>, which is comparable to the effect of reducing AGR NH<sub>3</sub> emissions (0.9–2.0 μg m<sup>-3</sup>). It should be noted that the reduction in PM<sub>2.5</sub> resulting from both AGR and non-AGR NH<sub>3</sub> emissions is not proportional to the emission reduction across all sectors. This is due to the non-linear relationship between NH<sub>3</sub> emissions and PM<sub>2.5</sub> concentrations.

This study utilized the integrated exposure–response (IER) model to estimate premature mortality resulting from PM<sub>2.5</sub> exposure. Detailed methods and data can be found in our previous work (Li et al., 2023a). In the base case, PM<sub>2.5</sub> exposure exhibits a significant impact on premature mortality, leading to 698.4 thousand deaths in the study region. Specifically, premature deaths attributable to ischemic heart disease (IHD), stroke, lung cancer (LC), and chronic obstructive pulmonary disease (COPD) are 202.3, 347.9, 61.5, and 86.7 thousand, respectively. In other scenarios, the overall premature mortality burden decreases by 45.6–72.0 thousand instances (6.5%–10.3%) in Eastern China. Notably, the decline in premature deaths, especially those related to stroke, plays a significant role in the overall reduction.

### **6 Conclusions**

An accurate NH<sub>3</sub> emission inventory is essential for developing effective air quality improvement policies. Numerous studies have demonstrated that the current bottom-up NH<sub>3</sub> emission inventories in China often underestimate the total NH<sub>3</sub> emissions, with significant uncertainties in the estimation of emissions from various sources. In this study, we used IASI satellite products and an iterative algorithm with the WRF-Chem model to optimize the bottom-up NH<sub>3</sub> emission inventory for Eastern China and further assessed the impacts of NH<sub>3</sub> emission reductions from different sources on PM<sub>2.5</sub> concentrations.

The posterior results indicate that the NH<sub>3</sub> emission in Eastern China for 2016 amounted to 4.2 Tg. The highest emissions occurred in summer (463.1 Gg month<sup>-1</sup>), with AGR sources contributing 86.5% and non-AGR sources contributing 13.5%. In contrast, emissions were lowest in winter (217.4 Gg month<sup>-1</sup>), and the proportion of emissions from non-AGR sources were higher than that from AGR sources. Spatially, the region with the highest NH<sub>3</sub> emissions was located at the intersection of Henan, Hebei, and Shandong provinces. This is attributed to a combination of high emission intensity from dense agricultural and industrial activities and topographical effects that hinder the dispersal of pollutants. The optimization of the NH<sub>3</sub> inventory further improved the simulation underestimation of the NH<sub>3</sub> total column (MFB from -61% to -30%) and surface concentration (MFB from -61% to -19%). It also indirectly improved the simulation of other air pollutants, such as PM<sub>2.5</sub>, NO<sub>2</sub> and SO<sub>2</sub>.

Based on the posterior emission inventory, we conducted a series of sensitivity simulations to investigate the response of PM<sub>2.5</sub> concentrations to NH<sub>3</sub> emission reductions. A 30%–60% reduction in NH<sub>3</sub> emissions resulted in a 1.5–8.8 μg m<sup>-3</sup> decrease in PM<sub>2.5</sub> concentrations. In terms of sectoral contributions, reductions in AGR emissions led to a decrease in PM<sub>2.5</sub> ranging from 0.9 μg m<sup>-3</sup> to 7.4 μg m<sup>-3</sup>, while the response to reductions in non-AGR NH<sub>3</sub> emissions ranged from 0.9 μg m<sup>-3</sup> to 5.3 μg m<sup>-3</sup>. Furthermore, the reduction in NH<sub>3</sub> emissions had a beneficial impact on public health, with a 6.5%–10.3% decrease in premature deaths attributed to PM<sub>2.5</sub> exposure.

| 480               | This study obtained a high-resolution NH3 emission inventory for Eastern China                                                                                                                                                             |
|-------------------|--------------------------------------------------------------------------------------------------------------------------------------------------------------------------------------------------------------------------------------------|
| 481               | and highlights the significant role of non-AGR NH3 emission reductions in further                                                                                                                                                          |
| 482               | decreasing PM <sub>2.5</sub> levels. The findings provide robust data support for air quality                                                                                                                                              |
| 483               | research and offer scientific insights for exploring the potential air quality and public                                                                                                                                                  |
| 484               | health benefits of NH <sub>3</sub> emission reduction.                                                                                                                                                                                     |
| 485               |                                                                                                                                                                                                                                            |
| 486               | Conflicts of Interest: The authors declare that the research was conducted in the                                                                                                                                                          |
| 487               | absence of any commercial or financial relationships that could be construed as a                                                                                                                                                          |
| 488               | potential conflict of interest.                                                                                                                                                                                                            |
| 489               |                                                                                                                                                                                                                                            |
| 490               | Author Contributions: Data curation, model simulation, visualization, and writing-                                                                                                                                                         |
| 491               | original draft preparation, KQT, HRZ, GX, FYC, JM, XC and YX; Supervision,                                                                                                                                                                 |
| 492               | funding acquisition, writing-review and editing, NL, HL, JBJ, BJL and KL. All authors                                                                                                                                                      |
| 493               | have read and agreed to the published version of the manuscript.                                                                                                                                                                           |
| 494               |                                                                                                                                                                                                                                            |
| 495               | Funding: This work was supported by the National Key Research and Development                                                                                                                                                              |
| 496               | Program of China (2022YFC3701005), the special found of State Environmental                                                                                                                                                                |
| 497               | Protection Key Laboratory of Formation and Prevention of Urban Air Pollution                                                                                                                                                               |
| 498               | Complex (SEPAir-2024080216).                                                                                                                                                                                                               |
| 499               |                                                                                                                                                                                                                                            |
| 500               | Acknowledgements: The numerical calculations in this paper have been done on the                                                                                                                                                           |
| 501               | supercomputing system in the Supercomputing Center of Nanjing University of                                                                                                                                                                |
| 502               | Information Science & Technology.                                                                                                                                                                                                          |
| 503               |                                                                                                                                                                                                                                            |
| 504               | Reference                                                                                                                                                                                                                                  |
| 505<br>506<br>507 | Bai L, Lu X, Yin S, Zhang H, Ma S, Wang C, et al. 2020. A recent emission inventory of multiple air pollutant, pm2.5 chemical species and its spatial-temporal characteristics in central china. Journal of Cleaner Production 269:122114. |
| 508<br>509        | An, J., Huang, Y., Huang, C., Wang, X., Yan, R., Wang, Q., Wang, H., Jing, S., Zhang, Y., Liu, Y., Chen, Y., Xu, C., Qiao, L., Zhou, M., Zhu, S., Hu, Q., Lu, J., and Chen, C.: Emission inventory of                                      |

- air pollutants and chemical speciation for specific anthropogenic sources based on local
- measurements in the Yangtze River Delta region, China, Atmos. Chem. Phys., 21, 2003–2025,
- https://doi.org/10.5194/acp-21-2003-2021, 2021.
- Bessagnet, B., Beauchamp, M., Guerreiro, C., de Leeuw, F., Tsyro, S., Colette, A., Meleux, F., Rouïl,
- L., Ruyssenaars, P., Sauter, F., Velders, G. J. M., Foltescu, V. L., and van Aardenne, J.: Can further
- mitigation of ammonia emissions reduce exceedances of particulate matter air quality standards?,
- Environmental Science & Policy, 44, 149–163, https://doi.org/10.1016/j.envsci.2014.07.011, 2014.
- Cai, S., Liu, T., Huang, X., Song, Y., Wang, T., Sun, Z., Gao, J., and Ding, A.: Important Role of Low
- Cloud and Fog in Sulfate Aerosol Formation During Winter Haze Over the North China Plain,
- Geophysical Research Letters, 51, e2023GL106597, https://doi.org/10.1029/2023GL106597, 2024.
- Chang, Y., Deng, C., Dore, A. J., and Zhuang, G.: Human Excreta as a Stable and Important Source of
- Atmospheric Ammonia in the Megacity of Shanghai, PLoS One, 10, e0144661,
- https://doi.org/10.1371/journal.pone.0144661, 2015.
- Chang, Y., Zou, Z., Deng, C., Huang, K., Collett, J. L., Lin, J., and Zhuang, G.: The importance of
- vehicle emissions as a source of atmospheric ammonia in the megacity of Shanghai, Atmospheric
- Chemistry and Physics, 16, 3577–3594, https://doi.org/10.5194/acp-16-3577-2016, 2016.
- Chen, J., Du, X., Liu, X., Xu, W., and Krol, M.: Estimation of Ammonia Emissions over China Using
- IASI Satellite-Derived Surface Observations, Environ. Sci. Technol., 59, 9991–10000,
- https://doi.org/10.1021/acs.est.4c10878, 2025.
- Chen, P. and Wang, O.: Underestimated industrial ammonia emission in China uncovered by material
- flow analysis, Environmental Pollution, 368, 125740, https://doi.org/10.1016/j.envpol.2025.125740,
- 2025.
- Chen, Y., Shen, H., Kaiser, J., Hu, Y., Capps, S. L., Zhao, S., Hakami, A., Shih, J.-S., Pavur, G. K.,
- Turner, M. D., Henze, D. K., Resler, J., Nenes, A., Napelenok, S. L., Bash, J. O., Fahey, K. M.,
- Carmichael, G. R., Chai, T., Clarisse, L., Coheur, P.-F., Van Damme, M., and Russell, A. G.: High-
- resolution hybrid inversion of IASI ammonia columns to constrain US ammonia emissions using
- the CMAQ adjoint model, Atmospheric Chemistry and Physics, 21, 2067–2082,
- https://doi.org/10.5194/acp-21-2067-2021, 2021.
- Chen, Y., Zhang, Q., Cai, X., Zhang, H., Lin, H., Zheng, C., Guo, Z., Hu, S., Chen, L., Tao, S., Liu,
- 539 M., and Wang, X.: Rapid Increase in China's Industrial Ammonia Emissions: Evidence from Unit-
- Based Mapping, Environ Sci Technol, https://doi.org/10.1021/acs.est.1c08369, 2022.
- Cheng, L., Ye, Z., Cheng, S., and Guo, X.: Agricultural ammonia emissions and its impact on PM2.5
- concentrations in the Beijing-Tianjin-Hebei region from 2000 to 2018, Environ Pollut, 291, 118162,
- https://doi.org/10.1016/j.envpol.2021.118162, 2021.
- Du, P., Du, H., Zhang, W., Lu, K., Zhang, C., Ban, J., Wang, Y., Liu, T., Hu, J., and Li, T.: Unequal
- Health Risks and Attributable Mortality Burden of Source-Specific PM<sub>2.5</sub> in China, Environ. Sci.
- Technol., 58, 10897–10909, https://doi.org/10.1021/acs.est.3c08789, 2024.

- Feng, S., Gao, D., Liao, F., Zhou, F., and Wang, X.: The health effects of ambient PM2.5 and potential
- mechanisms, Ecotoxicology and Environmental Safety, 128, 67-74,
- https://doi.org/10.1016/j.ecoenv.2016.01.030, 2016.
- Feng, S., Xu, W., Cheng, M., Ma, Y., Wu, L., Kang, J., Wang, K., Tang, A., Collett, J. L., Fang, Y.,
- Goulding, K., Liu, X., and Zhang, F.: Overlooked Nonagricultural and Wintertime Agricultural NH3
- Emissions in Quzhou County, North China Plain: Evidence from 15N-Stable Isotopes,
- Environmental Science & Technology Letters, 9, 127–133,
- https://doi.org/10.1021/acs.estlett.1c00935, 2022.
- Fu, X., Wang, S. X., Ran, L. M., Pleim, J. E., Cooter, E., Bash, J. O., Benson, V., and Hao, J. M.:
- Estimating NH<sub&gt;3&lt;/sub&gt; emissions from agricultural fertilizer application in China
- using the bi-directional CMAQ model coupled to an agro-ecosystem model, Atmos. Chem. Phys.,
- 15, 6637–6649, https://doi.org/10.5194/acp-15-6637-2015, 2015.
- Gao, D., Zhao, B., Wang, S., Wang, Y., Gaudet, B., Zhu, Y., Wang, X., Shen, J., Li, S., He, Y., Yin, D.,
- and Dong, Z.: Increased importance of aerosol-cloud interactions for surface PM<sub>2.5</sub> pollution
- relative to aerosol-radiation interactions in China with the anthropogenic emission reductions,
- Atmos. Chem. Phys., 23, 14359–14373, https://doi.org/10.5194/acp-23-14359-2023, 2023.
- Geng, G., Liu, Y., Liu, Y., Liu, S., Cheng, J., Yan, L., Wu, N., Hu, H., Tong, D., Zheng, B., Yin, Z., He,
- 564 K., and Zhang, Q.: Efficacy of China's clean air actions to tackle PM2.5 pollution between 2013
- and 2020, Nat. Geosci., 17, 987–994, https://doi.org/10.1038/s41561-024-01540-z, 2024.
- Guo, H., Otjes, R., Schlag, P., Kiendler-Scharr, A., Nenes, A., and Weber, R. J.: Effectiveness of
- ammonia reduction on control of fine particle nitrate, Atmospheric Chemistry and Physics, 18,
- 12241–12256, https://doi.org/10.5194/acp-18-12241-2018, 2018.
- Guo, X., Ye, Z., Chen, D., Wu, H., Shen, Y., Liu, J., and Cheng, S.: Prediction and mitigation potential
- of anthropogenic ammonia emissions within the Beijing-Tianjin-Hebei region, China, Environ
- Pollut, 259, 113863, https://doi.org/10.1016/j.envpol.2019.113863, 2020.
- Guo, Y., Zhang, L., Winiwarter, W., Van Grinsven, H. J. M., Wang, X., Li, K., Pan, D., Liu, Z., and
- Gu, B.: Ambitious nitrogen abatement is required to mitigate future global PM2.5 air pollution
- toward the World Health Organization targets, One Earth, 7, 1600-1613,
- https://doi.org/10.1016/j.oneear.2024.08.007, 2024.
- Hoesly, R. M., Smith, S. J., Feng, L., Klimont, Z., Janssens-Maenhout, G., Pitkanen, T., Seibert, J. J.,
- Vu, L., Andres, R. J., Bolt, R. M., Bond, T. C., Dawidowski, L., Kholod, N., Kurokawa, J., Li, M.,
- Liu, L., Lu, Z., Moura, M. C. P., O'Rourke, P. R., and Zhang, Q.: Historical (1750–2014)
- anthropogenic emissions of reactive gases and aerosols from the Community Emissions Data
- System (CEDS), Geosci. Model Dev., 11, 369–408, https://doi.org/10.5194/gmd-11-369-2018, 2018.
- Hu, S., Zhao, G., Tan, T., Li, C., Zong, T., Xu, N., Zhu, W., and Hu, M.: Current challenges of
- improving visibility due to increasing nitrate fraction in PM2.5 during the haze days in Beijing,
- China, Environmental Pollution, 290, 118032, https://doi.org/10.1016/j.envpol.2021.118032, 2021.

- Huang, L., Zhu, Y., Zhai, H., Xue, S., Zhu, T., Shao, Y., Liu, Z., Emery, C., Yarwood, G., Wang, Y.,
- Fu, J., Zhang, K., and Li, L.: Recommendations on benchmarks for numerical air quality model
- applications in China Part 1: PM<sub>2.5</sub> and chemical species, Atmos. Chem. Phys., 21, 2725–2743,
- https://doi.org/10.5194/acp-21-2725-2021, 2021.
- Huang, R.-J., Zhang, Y., Bozzetti, C., Ho, K.-F., Cao, J.-J., Han, Y., Daellenbach, K. R., Slowik, J. G.,
- Platt, S. M., Canonaco, F., Zotter, P., Wolf, R., Pieber, S. M., Bruns, E. A., Crippa, M., Ciarelli, G.,
- Piazzalunga, A., Schwikowski, M., Abbaszade, G., Schnelle-Kreis, J., Zimmermann, R., An, Z.,
- Szidat, S., Baltensperger, U., Haddad, I. E., and Prévôt, A. S. H.: High secondary aerosol
- contribution to particulate pollution during haze events in China, Nature, 514, 218-222,
- https://doi.org/10.1038/nature13774, 2014.
- Huang, X., Song, Y., Li, M., Li, J., Huo, Q., Cai, X., Zhu, T., Hu, M., and Zhang, H.: A high-resolution
- ammonia emission inventory in China, Global Biogeochemical Cycles, 26, n/a-n/a,
- https://doi.org/10.1029/2011gb004161, 2012.
- Jin, J., Fang, L., Li, B., Liao, H., Wang, Y., Han, W., Li, K., Pang, M., Wu, X., and Xiang Lin, H.:
- 4DEnVar-based inversion system for ammonia emission estimation in China through assimilating
- IASI ammonia retrievals, Environ. Res. Lett., 18, 034005, https://doi.org/10.1088/1748-
- 9326/acb835, 2023.
- Kang, Y., Liu, M., Song, Y., Huang, X., Yao, H., Cai, X., Zhang, H., Kang, L., Liu, X., Yan, X., He,
- H., Zhang, Q., Shao, M., and Zhu, T.: High-resolution ammonia emissions inventories in China from
- 1980 to 2012, Atmospheric Chemistry and Physics, 16, 2043–2058, https://doi.org/10.5194/acp-16-
- 2043-2016, 2016.
- Kong, L., Tang, X., Zhu, J., Wang, Z., Pan, Y., Wu, H., Wu, L., Wu, Q., He, Y., Tian, S., Xie, Y., Liu,
- Z., Sui, W., Han, L., and Carmichael, G.: Improved Inversion of Monthly Ammonia Emissions in
- China Based on the Chinese Ammonia Monitoring Network and Ensemble Kalman Filter, Environ
- Sci Technol, 53, 12529–12538, https://doi.org/10.1021/acs.est.9b02701, 2019.
- Lei, R., Nie, D., Zhang, S., Yu, W., Ge, X., and Song, N.: Spatial and temporal characteristics of air
- pollutants and their health effects in China during 2019-2020, Journal of Environmental
- Management, 317, 115460, https://doi.org/10.1016/j.jenvman.2022.115460, 2022.
- Lei, Y., Yin, Z., Lu, X., Zhang, Q., Gong, J., Cai, B., Cai, C., Chai, Q., Chen, H., Chen, R., Chen, S.,
- Chen, W., Cheng, J., Chi, X., Dai, H., Feng, X., Geng, G., Hu, J., Hu, S., Huang, C., Li, T., Li, W.,
- Li, X., Liu, J., Liu, X., Liu, Z., Ma, J., Qin, Y., Tong, D., Wang, X., Wang, X., Wu, R., Xiao, Q., Xie,
- Y., Xu, X., Xue, T., Yu, H., Zhang, D., Zhang, N., Zhang, S., Zhang, S., Zhang, X., Zhang, X., Zhang,
- Z., Zheng, B., Zheng, Y., Zhou, J., Zhu, T., Wang, J., and He, K.: The 2022 report of synergetic
- roadmap on carbon neutrality and clean air for China: Accelerating transition in key sectors,
- Environmental Science and Ecotechnology, 19, 100335, https://doi.org/10.1016/j.ese.2023.100335,
- 2024.
- Li, B., Chen, L., Shen, W., Jin, J., Wang, T., Wang, P., Yang, Y., and Liao, H.: Improved gridded
- ammonia emission inventory in China, Atmospheric Chemistry and Physics, 21, 15883–15900,

- https://doi.org/10.5194/acp-21-15883-2021, 2021.
- Li, B., Liao, H., Li, K., Wang, Y., Zhang, L., Guo, Y., Liu, L., Li, J., Jin, J., Yang, Y., Gong, C., Wang,
- T., Shen, W., Wang, P., Dang, R., Liao, K., Zhu, Q., and Jacob, D. J.: Unlocking nitrogen
- management potential via large-scale farming for air quality and substantial co-benefits, National
- Science Review, 11, nwae324, https://doi.org/10.1093/nsr/nwae324, 2024.
- Li, M., Liu, H., Geng, G., Hong, C., Liu, F., Song, Y., Tong, D., Zheng, B., Cui, H., Man, H., Zhang,
- Q., and He, K.: Anthropogenic emission inventories in China: a review, National Science Review,
- 4, 834–866, https://doi.org/10.1093/nsr/nwx150, 2017.
- Li, N., Zhang, H., Zhu, S., Liao, H., Hu, J., Tang, K., Feng, W., Zhang, R., Shi, C., Xu, H., Chen, L.,
- and Li, J.: Secondary PM2.5 dominates aerosol pollution in the Yangtze River Delta region:
- Environmental and health effects of the Clean air Plan, Environment International, 171, 107725,
- https://doi.org/10.1016/j.envint.2022.107725, 2023a.
- Li, R., Gao, Y., Xu, J., Cui, L., and Wang, G.: Impact of Clean Air Policy on Criteria Air Pollutants
- and Health Risks Across China During 2013–2021, JGR Atmospheres, 128, e2023JD038939,
- https://doi.org/10.1029/2023JD038939, 2023b.
- Liu, H., Lei, J., Liu, Y., Zhu, T., Chan, K., Chen, X., Wei, J., Deng, F., Li, G., Jiang, Y., Bai, L., Wang,
- 638 K., Chen, J., Lan, Y., Xia, X., Wang, J., Wei, C., Li, Y., Chen, R., Gong, J., Duan, X., Zhang, K.,
- Kan, H., Shi, X., Guo, X., and Wu, S.: Hospital admissions attributable to reduced air pollution due
- to clean-air policies in China, Nat Med, https://doi.org/10.1038/s41591-025-03515-y, 2025.
- Liu, M., Huang, X., Song, Y., Tang, J., Cao, J., Zhang, X., Zhang, Q., Wang, S., Xu, T., Kang, L., Cai,
- X., Zhang, H., Yang, F., Wang, H., Yu, J. Z., Lau, A. K. H., He, L., Huang, X., Duan, L., Ding, A.,
- Xue, L., Gao, J., Liu, B., and Zhu, T.: Ammonia emission control in China would mitigate haze
- pollution and nitrogen deposition, but worsen acid rain, Proc Natl Acad Sci U S A, 116, 7760–7765,
- https://doi.org/10.1073/pnas.1814880116, 2019.
- Liu, P., Ding, J., Liu, L., Xu, W., and Liu, X.: Estimation of surface ammonia concentrations and
- emissions in China from the polar-orbiting Infrared Atmospheric Sounding Interferometer and the
- FY-4A Geostationary Interferometric Infrared Sounder, Atmos. Chem. Phys., 22, 9099–9110,
- https://doi.org/10.5194/acp-22-9099-2022, 2022a.
- Liu, R., Liu, T., Huang, X., Ren, C., Wang, L., Niu, G., Yu, C., Zhang, Y., Wang, J., Qi, X., Nie, W.,
- Chi, X., and Ding, A.: Characteristics and sources of atmospheric ammonia at the SORPES station
- in the western Yangtze river delta of China, Atmospheric Environment, 318, 120234,
- https://doi.org/10.1016/j.atmosenv.2023.120234, 2024.
- Liu, S., Geng, G., Xiao, Q., Zheng, Y., Liu, X., Cheng, J., and Zhang, Q.: Tracking Daily
- Concentrations of PM2.5 Chemical Composition in China since 2000, Environ. Sci. Technol., 56,
- 16517–16527, https://doi.org/10.1021/acs.est.2c06510, 2022b.
- Liu, Z., Zhou, M., Chen, Y., Chen, D., Pan, Y., Song, T., Ji, D., Chen, Q., and Zhang, L.: The nonlinear
- response of fine particulate matter pollution to ammonia emission reductions in North China,

- Environmental Research Letters, https://doi.org/10.1088/1748-9326/abdf86, 2021.
- Liu, Z., Rieder, H. E., Schmidt, C., Mayer, M., Guo, Y., Winiwarter, W., and Zhang, L.: Optimal
- reactive nitrogen control pathways identified for cost-effective PM2.5 mitigation in Europe, Nat
- Commun, 14, 4246, https://doi.org/10.1038/s41467-023-39900-9, 2023.
- Lu, L., Yuan, L., Cai, Z., Fu, J., and Wu, G.: Emission inventory and distribution characteristics of
- NH3 from agricultural fertilizers in Hunan, China, from 2012 to 2021, Atmospheric Pollution
- Research, 16, 102479, https://doi.org/10.1016/j.apr.2025.102479, 2025.
- 666 Ma, S.: High-resolution assessment of ammonia emissions in China: Inventories, driving forces and
- mitigation, Atmospheric Environment, 229, 117458,
- https://doi.org/10.1016/j.atmosenv.2020.117458, 2020.
- Pan, D., Mauzerall, D. L., Wang, R., Guo, X., Puchalski, M., Guo, Y., Song, S., Tong, D., Sullivan, A.
- P., Schichtel, B. A., Collett, J. L., and Zondlo, M. A.: Regime shift in secondary inorganic aerosol
- formation and nitrogen deposition in the rural United States, Nat. Geosci., 17, 617-623,
- https://doi.org/10.1038/s41561-024-01455-9, 2024.
- Pan, Y., Tian, S., Liu, D., Fang, Y., Zhu, X., Zhang, Q., Zheng, B., Michalski, G., and Wang, Y.: Fossil
- Fuel Combustion-Related Emissions Dominate Atmospheric Ammonia Sources during Severe Haze
- Episodes: Evidence from (15)N-Stable Isotope in Size-Resolved Aerosol Ammonium, Environ Sci
- Technol, 50, 8049–56, https://doi.org/10.1021/acs.est.6b00634, 2016.
- Pan, Y., Tian, S., Zhao, Y., Zhang, L., Zhu, X., Gao, J., Huang, W., Zhou, Y., Song, Y., Zhang, Q., and
- Wang, Y.: Identifying Ammonia Hotspots in China Using a National Observation Network, Environ
- Sci Technol, 52, 3926–3934, https://doi.org/10.1021/acs.est.7b05235, 2018a.
- Pan, Y., Tian, S., Liu, D., Fang, Y., Zhu, X., Gao, M., Gao, J., Michalski, G., and Wang, Y.: Isotopic
- evidence for enhanced fossil fuel sources of aerosol ammonium in the urban atmosphere, Environ
- Pollut, 238, 942–947, https://doi.org/10.1016/j.envpol.2018.03.038, 2018b.
- Paulot, F., Jacob, D. J., Pinder, R. W., Bash, J. O., Travis, K., and Henze, D. K.: Ammonia emissions
- in the United States, European Union, and China derived by high-resolution inversion of ammonium
- wet deposition data: Interpretation with a new agricultural emissions inventory (MASAGE NH3),
- Journal of Geophysical Research: Atmospheres, 119, 4343-4364,
- https://doi.org/10.1002/2013jd021130, 2014.
- Pendergrass, D. C., Jacob, D. J., Oak, Y. J., Lee, J., Kim, M., Kim, J., Lee, S., Zhai, S., Irie, H., and
- Liao, H.: A continuous 2011–2022 record of fine particulate matter (PM2.5) in East Asia at daily 2-
- km resolution from geostationary satellite observations: Population exposure and long-term trends,
- Atmospheric Environment, 346, 121068, https://doi.org/10.1016/j.atmosenv.2025.121068, 2025.
- Peng, Z., Wang, H., Zhang, M., Zhang, Y., Li, L., Li, Y., and Ao, Z.: Analysis of aerosol chemical
- components and source apportionment during a long-lasting haze event in the Yangtze River Delta,
- China, Journal of Environmental Sciences, 156, 14–29, https://doi.org/10.1016/j.jes.2024.06.023,
- 2025.

- Pu, W., Ma, Z., Collett, J. L., Guo, H., Lin, W., Cheng, Y., Quan, W., Li, Y., Dong, F., and He, D.:
- Regional transport and urban emissions are important ammonia contributors in Beijing, China,
- Environ Pollut, 265, 115062, https://doi.org/10.1016/j.envpol.2020.115062, 2020.
- Qi, J., Zheng, B., Li, M., Yu, F., Chen, C., Liu, F., Zhou, X., Yuan, J., Zhang, Q., and He, K.: A high-
- resolution air pollutants emission inventory in 2013 for the Beijing-Tianjin-Hebei region, China,
- Atmospheric Environment, 170, 156–168, https://doi.org/10.1016/j.atmosenv.2017.09.039, 2017.
- Ren, C., Huang, X., Liu, T., Song, Y., Wen, Z., Liu, X., Ding, A., and Zhu, T.: A dynamic ammonia
- emission model and the online coupling with WRF-Chem (WRF-SoilN-Chem v1.0): development
- and regional evaluation in China, Geoscientific Model Development, 16, 1641–1659,
- https://doi.org/10.5194/gmd-16-1641-2023, 2023.
- Ren, C., Huang, X., Wang, Y., Zhang, L., Zhou, X., Sun, W., Zhang, H., Liu, T., Ding, A., and Wang,
- 707 T.: Enhanced Soil Emissions of Reactive Nitrogen Gases by Fertilization and Their Impacts on
- Secondary Air Pollution in Eastern China, Environ. Sci. Technol., 59, 5119-5130,
- https://doi.org/10.1021/acs.est.4c12324, 2025.
- Shao, S.-C., Zhang, Y.-L., Chang, Y.-H., Cao, F., Lin, Y.-C., Mozaffar, A., and Hong, Y.-H.: Online
- characterization of a large but overlooked human excreta source of ammonia in China's urban
- atmosphere, Atmospheric Environment, 230, 117459,
- https://doi.org/10.1016/j.atmosenv.2020.117459, 2020.
- Song, L., Walters, W. W., Pan, Y., Li, Z., Gu, M., Duan, Y., Lü, X., and Fang, Y.: 15N natural abundance
- of vehicular exhaust ammonia, quantified by active sampling techniques, Atmospheric Environment,
- 255, 118430, https://doi.org/10.1016/j.atmosenv.2021.118430, 2021.
- Song, Q., Huang, L., Zhang, Y., Li, Z., Wang, S., Zhao, B., Yin, D., Ma, M., Li, S., Liu, B., Zhu, L.,
- Chang, X., Gao, D., Jiang, Y., Dong, Z., Shi, H., and Hao, J.: Driving Factors of PM<sub>2.5</sub> Pollution
- Rebound in North China Plain in Early 2023, Environ. Sci. Technol. Lett., 12, 305–312,
- https://doi.org/10.1021/acs.estlett.4c01153, 2025.
- Sun, K., Tao, L., Miller, D. J., Pan, D., Golston, L. M., Zondlo, M. A., Griffin, R. J., Wallace, H. W.,
- Leong, Y. J., Yang, M. M., Zhang, Y., Mauzerall, D. L., and Zhu, T.: Vehicle Emissions as an
- Important Urban Ammonia Source in the United States and China, Environ Sci Technol, 51, 2472–
- 2481, https://doi.org/10.1021/acs.est.6b02805, 2017.
- Tan, T., Hu, M., Li, M., Guo, Q., Wu, Y., Fang, X., Gu, F., Wang, Y., and Wu, Z.: New insight into
- PM2.5 pollution patterns in Beijing based on one-year measurement of chemical compositions,
- Science of The Total Environment, 621, 734–743, https://doi.org/10.1016/j.scitotenv.2017.11.208,
- 2018.
- Tang, C., Shi, C., Letu, H., Yin, S., Nakajima, T., Sekiguchi, M., Xu, J., Zhao, M., Ma, R., and Wang,
- 730 W.: Development of a hybrid algorithm for the simultaneous retrieval of aerosol optical thickness
- and fine-mode fraction from multispectral satellite observation combining radiative transfer and
- transfer learning approaches, Remote Sensing of Environment, 319, 114619,

- https://doi.org/10.1016/j.rse.2025.114619, 2025.
- Ti, C., Han, X., Chang, S. X., Peng, L., Xia, L., and Yan, X.: Mitigation of agricultural NH3 emissions
- reduces PM2.5 pollution in China: A finer scale analysis, Journal of Cleaner Production, 350,
- 131507, https://doi.org/10.1016/j.jclepro.2022.131507, 2022.
- Van Damme, M., Clarisse, L., Whitburn, S., Hadji-Lazaro, J., Hurtmans, D., Clerbaux, C., and Coheur,
- P. F.: Industrial and agricultural ammonia point sources exposed, Nature, 564, 99–103,
- https://doi.org/10.1038/s41586-018-0747-1, 2018.
- Wang, G., Zhang, R., Gomez, M. E., Yang, L., Levy Zamora, M., Hu, M., Lin, Y., Peng, J., Guo, S.,
- Meng, J., Li, J., Cheng, C., Hu, T., Ren, Y., Wang, Y., Gao, J., Cao, J., An, Z., Zhou, W., Li, G.,
- Wang, J., Tian, P., Marrero-Ortiz, W., Secrest, J., Du, Z., Zheng, J., Shang, D., Zeng, L., Shao, M.,
- Wang, W., Huang, Y., Wang, Y., Zhu, Y., Li, Y., Hu, J., Pan, B., Cai, L., Cheng, Y., Ji, Y., Zhang, F.,
- Rosenfeld, D., Liss, P. S., Duce, R. A., Kolb, C. E., and Molina, M. J.: Persistent sulfate formation
- from London Fog to Chinese haze, Proc Natl Acad Sci U S A, 113, 13630-13635,
- https://doi.org/10.1073/pnas.1616540113, 2016.
- Wang, J., Zhao, B., Wang, S., Yang, F., Xing, J., Morawska, L., Ding, A., Kulmala, M., Kerminen, V.-
- 748 M., Kujansuu, J., Wang, Z., Ding, D., Zhang, X., Wang, H., Tian, M., Petäjä, T., Jiang, J., and Hao,
- J.: Particulate matter pollution over China and the effects of control policies, Science of The Total
- Environment, 584–585, 426–447, https://doi.org/10.1016/j.scitotenv.2017.01.027, 2017.
- Wang, W., Liu, M., Wang, T., Song, Y., Zhou, L., Cao, J., Hu, J., Tang, G., Chen, Z., Li, Z., Xu, Z.,
- Peng, C., Lian, C., Chen, Y., Pan, Y., Zhang, Y., Sun, Y., Li, W., Zhu, T., Tian, H., and Ge, M.: Sulfate
- formation is dominated by manganese-catalyzed oxidation of SO2 on aerosol surfaces during haze
- events, Nat Commun, 12, https://doi.org/10.1038/s41467-021-22091-6, 2021.
- Wang, X., Gemayel, R., Hayeck, N., Perrier, S., Charbonnel, N., Xu, C., Chen, H., Zhu, C., Zhang, L.,
- Wang, L., Nizkorodov, S. A., Wang, X., Wang, Z., Wang, T., Mellouki, A., Riva, M., Chen, J., and
- George, C.: Atmospheric Photosensitization: A New Pathway for Sulfate Formation, Environ. Sci.
- Technol., 54, 3114–3120, https://doi.org/10.1021/acs.est.9b06347, 2020.
- Wei, J., Li, Z., Chen, X., Li, C., Sun, Y., Wang, J., Lyapustin, A., Brasseur, G. P., Jiang, M., Sun, L.,
- Wang, T., Jung, C. H., Qiu, B., Fang, C., Liu, X., Hao, J., Wang, Y., Zhan, M., Song, X., and Liu,
- Y.: Separating Daily 1 km PM2.5 Inorganic Chemical Composition in China since 2000 via Deep
- Learning Integrating Ground, Satellite, and Model Data, Environ. Sci. Technol., 57, 18282–18295,
- https://doi.org/10.1021/acs.est.3c00272, 2023.
- Wei, L., Zhang, H., Sun, C., and Yan, F.: Simultaneous estimation of ammonia injection rate and state
- of diesel urea-SCR system based on high gain observer, ISA Transactions, 126, 679-690,
- https://doi.org/10.1016/j.isatra.2021.08.002, 2022.
- Wu, C., Wang, G., Li, J., Li, J., Cao, C., Ge, S., Xie, Y., Chen, J., Liu, S., Du, W., Zhao, Z., and Cao,
- F.: Non-agricultural sources dominate the atmospheric NH3 in Xi'an, a megacity in the semi-arid
- region of China, Sci Total Environ, 722, 137756, https://doi.org/10.1016/j.scitotenv.2020.137756,

- 2020.
- Wu, N., Geng, G., Xu, R., Liu, S., Liu, X., Shi, Q., Zhou, Y., Zhao, Y., Liu, H., Song, Y., Zheng, J.,
- Zhang, Q., and He, K.: Development of a high-resolution integrated emission inventory of air
- pollutants for China, Earth Syst. Sci. Data, 16, 2893–2915, https://doi.org/10.5194/essd-16-2893-
- 2024, 2024.
- Xia, J., Zhou, Y., Fang, L., Qi, Y., Li, D., Liao, H., and Jin, J.: South Asia ammonia emission inversion
- through assimilating IASI observations, https://doi.org/10.5194/egusphere-2024-3938, 20 January
- 2025.
- Xiao, Q., Geng, G., Xue, T., Liu, S., Cai, C., He, K., and Zhang, Q.: Tracking PM<sub>2.5</sub> and O<sub>3</sub> Pollution
- and the Related Health Burden in China 2013-2020, Environ. Sci. Technol., 56, 6922-6932,
- https://doi.org/10.1021/acs.est.1c04548, 2022.
- Xu, W., Zhao, Y., Wen, Z., Chang, Y., Pan, Y., Sun, Y., Ma, X., Sha, Z., Li, Z., Kang, J., Liu, L., Tang,
- A., Wang, K., Zhang, Y., Guo, Y., Zhang, L., Sheng, L., Zhang, X., Gu, B., Song, Y., Van Damme,
- 783 M., Clarisse, L., Coheur, P.-F., Collett, J. L., Goulding, K., Zhang, F., He, K., and Liu, X.: Increasing
- importance of ammonia emission abatement in PM2.5 pollution control, Science Bulletin, 67, 1745–
- 1749, https://doi.org/10.1016/j.scib.2022.07.021, 2022.
- Yang, G., Ren, G., Zhang, P., Xue, X., Tysa, S. K., Jia, W., Qin, Y., Zheng, X., and Zhang, S.: PM<sub>2.5</sub>
- Influence on Urban Heat Island (UHI) Effect in Beijing and the Possible Mechanisms, JGR
- Atmospheres, 126, e2021JD035227, https://doi.org/10.1029/2021JD035227, 2021.
- Yang, Z., Wang, Y., Xu, X.-H., Yang, J., and Ou, C.-O.: Quantifying and characterizing the impacts of
- PM2.5 and humidity on atmospheric visibility in 182 Chinese cities: A nationwide time-series study,
- Journal of Cleaner Production, 368, 133182, https://doi.org/10.1016/j.jclepro.2022.133182, 2022.
- Zhang, H., Zhou, X., Ren, C., Li, M., Liu, T., and Huang, X.: A systematic review of reactive nitrogen
- simulations with chemical transport models in China, Atmospheric Research, 309, 107586,
- https://doi.org/10.1016/j.atmosres.2024.107586, 2024.
- Zhang, L., Chen, Y., Zhao, Y., Henze, D. K., Zhu, L., Song, Y., Paulot, F., Liu, X., Pan, Y., Lin, Y., and
- Huang, B.: Agricultural ammonia emissions in China: reconciling bottom-up and top-down
- estimates, Atmospheric Chemistry and Physics, 18, 339-355, https://doi.org/10.5194/acp-18-339-
- 2018, 2018.
- Zhang, Q., Zheng, Y., Tong, D., Shao, M., Wang, S., Zhang, Y., Xu, X., Wang, J., He, H., Liu, W., Ding,
- Y., Lei, Y., Li, J., Wang, Z., Zhang, X., Wang, Y., Cheng, J., Liu, Y., Shi, Q., Yan, L., Geng, G., Hong,
- C., Li, M., Liu, F., Zheng, B., Cao, J., Ding, A., Gao, J., Fu, Q., Huo, J., Liu, B., Liu, Z., Yang, F.,
- He, K., and Hao, J.: Drivers of improved PM2.5 air quality in China from 2013 to 2017, Proc Natl
- Acad Sci U S A, 116, 24463–24469, https://doi.org/10.1073/pnas.1907956116, 2019.
- Zhang, Q., Wei, N., Zou, C., and Mao, H.: Evaluating the ammonia emission from in-use vehicles
- using on-road remote sensing test, Environmental Pollution, 271, 116384,
- https://doi.org/10.1016/j.envpol.2020.116384, 2021.

- Zhang, X., Wu, Y., Liu, X., Reis, S., Jin, J., Dragosits, U., Van Damme, M., Clarisse, L., Whitburn, S.,
- Coheur, P. F., and Gu, B.: Ammonia Emissions May Be Substantially Underestimated in China,
- Environ Sci Technol, 51, 12089–12096, https://doi.org/10.1021/acs.est.7b02171, 2017.
- Zhang, Z., Yan, Y., Kong, S., Deng, O., Oin, S., Yao, L., Zhao, T., and Oi, S.: Benefits of refined NH3
- emission controls on PM2.5 mitigation in Central China, Sci Total Environ, 814, 151957,
- https://doi.org/10.1016/j.scitotenv.2021.151957, 2022.
- Zhao, Y., Yuan, M., Huang, X., Chen, F., and Zhang, J.: Quantification and evaluation of atmospheric
- ammonia emissions with different methods: a case study for the Yangtze River Delta region, China,
- Atmospheric Chemistry and Physics, 20, 4275–4294, https://doi.org/10.5194/acp-20-4275-2020,
- 2020.
- Zheng, B., Zhang, Q., Zhang, Y., He, K. B., Wang, K., Zheng, G. J., Duan, F. K., Ma, Y. L., and Kimoto,
- 818 T.: Heterogeneous chemistry: a mechanism missing in current models to explain secondary
- inorganic aerosol formation during the January 2013 haze episode in North China, Atmospheric
- Chemistry and Physics, 15, 2031–2049, https://doi.org/10.5194/acp-15-2031-2015, 2015.
- Zheng, B., Tong, D., Li, M., Liu, F., Hong, C., Geng, G., Li, H., Li, X., Peng, L., Qi, J., Yan, L., Zhang,
- Y., Zhao, H., Zheng, Y., He, K., and Zhang, Q.: Trends in China's anthropogenic emissions since
- 2010 as the consequence of clean air actions, Atmospheric Chemistry and Physics, 18, 14095–14111,
- https://doi.org/10.5194/acp-18-14095-2018, 2018.
- Zhou, M., Jiang, W., Gao, W., Gao, X., Ma, M., and Ma, X.: Anthropogenic emission inventory of
- multiple air pollutants and their spatiotemporal variations in 2017 for the Shandong Province, China,
- Environ Pollut, 288, 117666, https://doi.org/10.1016/j.envpol.2021.117666, 2021.
- Zhou, M., Nie, W., Qiao, L., Huang, D. D., Zhu, S., Lou, S., Wang, H., Wang, Q., Tao, S., Sun, P., Liu,
- Y., Xu, Z., An, J., Yan, R., Su, H., Huang, C., Ding, A., and Chen, C.: Elevated Formation of
- Particulate Nitrate From N<sub>2</sub> O<sub>5</sub> Hydrolysis in the Yangtze River Delta Region From 2011 to 2019,
- Geophysical Research Letters, 49, e2021GL097393, https://doi.org/10.1029/2021GL097393, 2022.
- Zhou, Y., Zhao, Y., Mao, P., Zhang, Q., Zhang, J., Qiu, L., and Yang, Y.: Development of a high-
- resolution emission inventory and its evaluation and application through air quality modeling for
- Jiangsu Province, China, Atmospheric Chemistry and Physics, 17, 211-233,
- https://doi.org/10.5194/acp-17-211-2017, 2017.
- Zhu, Q., Deng, Y.-L., Liu, Y., and Steenland, K.: Associations between Ultrafine Particles and Incident
- Dementia in Older Adults, Environ. Sci. Technol., acs.est.4c10574,
- https://doi.org/10.1021/acs.est.4c10574, 2025.

Table 1 List of sensitivity tests for optimized iterative algorithm.

| Case name            | AGR emission | Non-AGR<br>emission | Emission outside the domain |
|----------------------|--------------|---------------------|-----------------------------|
| $A_{total}$          | 4            | √                   | √                           |
| $A_{ m agr}$         | ✓            | ×                   | ×                           |
| $A_{\text{non-agr}}$ | ×            | √                   | ×                           |
| $A_{transport}$      | ×            | ×                   | √                           |
| $A_{blank}$          | ×            | ×                   | ×                           |

Table 2. NH<sub>3</sub> emission estimates in recent studies

| Region           | Sector              | Emission                         | Period    | Method      | Reference            |  |
|------------------|---------------------|----------------------------------|-----------|-------------|----------------------|--|
|                  |                     | 12.4 Tg yr <sup>-1</sup>         | 2016      | Bottom-up   | Ma (2020)            |  |
|                  |                     | 12.1 Tg yr <sup>-1</sup>         | 2016      | Bottom-up   | Li et al. (2021)     |  |
|                  |                     | 11.9~12.0<br>Tg yr <sup>-1</sup> | 2005~2015 | Bottom-up   | Chen et al (2021)    |  |
|                  |                     | 11.7 Tg yr <sup>-1</sup>         | 2008      | Top-down    | Zhang et a. (2018)   |  |
| China            |                     | 8.4 Tg N yr <sup>-1</sup>        | 2005-2008 | Top-down    | Paulot et al (2014)  |  |
|                  |                     | 0.74 Tg mon <sup>-1</sup>        | 2008 Apr  | Top-down    | Xu et al. (2013)     |  |
|                  |                     | 13.0 Tg yr <sup>-1</sup>         | 2016      | Top-down    | Kong et al<br>(2019) |  |
|                  |                     | 18.9 Tg yr <sup>-1</sup>         | 2015      | Top-down    | Zhang et a (2017)    |  |
| Eastern<br>China | Industry            | 274.5 Gg yr <sup>-1</sup>        | 2016      | Bottom-up   | Chen et al (2022)    |  |
|                  | /                   | 966.1 Gg yr <sup>-1</sup>        | 2016      | Bottom-up   | (Guo et al. 2020)    |  |
|                  |                     | 28.8 Gg mon <sup>-1</sup>        | 2015 Jan  |             | Huang et al. (2021)  |  |
|                  |                     | 82.5 Gg mon <sup>-1</sup>        | 2015 Apr  |             |                      |  |
| ВТН              | /                   | 102.9 Gg<br>mon <sup>-1</sup>    | 2015 Jul  | Top-down    |                      |  |
|                  |                     | 50.2 Gg mon <sup>-1</sup>        | 2015 Oct  |             |                      |  |
|                  | Agriculture         | 505.85 Gg yr                     | 2016      | To a dossum | This study           |  |
|                  | Non-<br>Agriculture | 282.53 Gg yr                     | 2016      | Top-down    |                      |  |
| VDD              | Agriculture         | 848.8 Gg yr <sup>-1</sup>        | 2014      | D. #        | Yu et al.            |  |
| YRD              | Non-                | 137.2 Gg yr <sup>-1</sup>        | 2014      | Bottom-up   | (2020)               |  |

|          | Agriculture                   |                               |                                   |             |                       |
|----------|-------------------------------|-------------------------------|-----------------------------------|-------------|-----------------------|
|          |                               | 77 Gg mon <sup>-1</sup>       | 2014 Jan                          | -<br>-      | Zhao et al.<br>(2020) |
|          |                               | 133 Gg mon <sup>-1</sup>      | 2014 Apr                          |             |                       |
|          | Agriculture                   | 169 Gg mon <sup>-1</sup>      | 2014 Jul                          | - Bottom-up |                       |
|          |                               | 108 Gg mon <sup>-1</sup>      | 108 Gg mon <sup>-1</sup> 2014 Oct |             |                       |
|          | 24.42 Gg<br>mon <sup>-1</sup> | 2015 Jan                      |                                   |             |                       |
|          | /                             | 88.0 Gg mon <sup>-1</sup>     | 2015 Apr                          | Top-down    | Huang et al. (2021)   |
|          |                               | 111.7 Gg<br>mon <sup>-1</sup> | 2015 Jul                          |             |                       |
|          |                               | 51.0 Gg mon <sup>-1</sup>     | 2015 Oct                          |             |                       |
|          | Agriculture                   | 1280.41 Gg                    |                                   |             |                       |
|          | Non-<br>Agriculture           | 297.86 Gg                     | 2016                              | Top-down    | this study            |
|          | /                             | 1035Gg yr <sup>-1</sup>       | 2013                              | Top-down    | Wang et al. (2018)    |
| **       |                               | 982 Gg yr <sup>-1</sup>       | 2016                              | Bottom-up   | Bai et al. (2020)     |
| Henan    | Agriculture                   | 647.73 Gg yr <sup>-</sup>     | 2016                              | Top-down    |                       |
|          | Non-                          | 206.20 Gg yr                  |                                   |             | this study            |
|          | Agriculture                   | 1                             |                                   |             |                       |
|          | /                             | 1210 Gg yr <sup>-1</sup>      | 2017                              | Bottom-up   | Zhou et al. (2021)    |
| Shandong | Agriculture                   | 715.29 Gg yr <sup>-</sup>     | 2016                              | Top-down    | this study            |
|          | Non- Agriculture              | 296.98 Gg yr <sup>-</sup>     |                                   |             |                       |

Table 3 Simulated and observed air pollutant concentrations

|                                         | Prior simulation | Posterior simulation | Observation |
|-----------------------------------------|------------------|----------------------|-------------|
| PM <sub>2.5</sub> (μg m <sup>-3</sup> ) | 65.7             | 67.3                 | 67.1        |
| NO <sub>2</sub> (ppb)                   | 22.3             | 22.1                 | 23.0        |
| SO <sub>2</sub> (ppb)                   | 8.2              | 6.8                  | 6.5         |

**Figure 1.** Simulation domains of the WRF-Chem model used in this study (left). Right panel illustrates the four research regions in Eastern China. Names and locations are labeled with different colors in this panel.

**Figure 2.** Prior and posterior NH<sub>3</sub> emissions from agricultural and non-agricultural sectors in the study region. The red numbers show the total emissions.

**Figure 3.** Scatter plots of the prior and posterior NH<sub>3</sub> total column data versus IASI retrievals. Each point represents prior (or posterior) data for a specific season and a specific region. Circles, triangles, rhombuses, and rectangles correspond to the BTH, Henan, Shandong, and YRD regions, respectively. Orange and blue markers represent a prior and a posterior data, respectively. The red box indicates the performance area, with a model error within  $\pm 30\%$  and an RMSE below  $10(\times 10^{15} \text{ molec cm}^{-2})$ .

Figure 4. Visualization of the workflow in this study.

**Figure 5.** Posterior emission characteristics. (a) Contribution from regional emission sectors. (b) Comparison of the posteriori and prior emissions (unit: Mg) in study region.

**Figure 6.** Distributions of NH<sub>3</sub> total column from prior simulation, posterior simulation and satellite retrieval in different seasons.

**Figure 7.** Response of NH<sub>3</sub> emission reduction in 30-60% in (a)-(b) concentration of PM<sub>2.5</sub> and (c) premature death caused by different diseases. The IHD, Stroke, LC and COPD represent the premature death caused by ischemic heart disease, stroke, lung cancer, chronic obstructive pulmonary disease.