# Peer review of "Optimizing Ammonia Emissions for PM2.5 Mitigation"

_EGUsphere, 2025_

## Referee Comment (RC2)

This manuscript presents a top-down estimate of ammonia emissions in China using IASI satellite observations and WRF-Chem simulations. The study finds a notable contribution from non-agricultural sources and highlights their implications for PM$_{2.5}$-related health burdens. The topic is of broad interest to the community. However, the manuscript requires major revision before it can be considered for publication. I have several concerns about the MLR-based top-down estimation approach, and the associated uncertainties need to be more clearly explained and discussed to evaluate the robustness of the results. Additionally, the interpretation of the top-down emission estimates should be strengthened by providing deeper insights and more comprehensive comparisons with existing literature.

**Major comments:**

1. Although the authors do not mention, it seems to me that the MLR-based approach assumes a linear relationship between NH$_3$ emissions and column concentrations and attributes all discrepancies between satellite observed and simulated NH$_3$ columns to local NH$_3$ emissions. It is important for the authors to clarify these assumptions and discuss the associated uncertainties. Specifically, (1) How might the nonlinear response of NH3 concentrations to emissions affect the results? For example, do the summed contributions from individual sources (SA_agriculture + SA_non-agriculture + SA_transport) approximate the simulated total NH$_3$ column from the prior? (2) To what extent could transport of emissions from nearby grid cells influence the posterior results and cause spatial misattribution of emissions? (3) How might uncertainties in emissions of other pollutants (e.g., SO$_2$, NO$_x$) or in the model's representation of inorganic aerosol formation, impact the posterior estimates of NH$_3$ emissions? Could the large discrepancies between prior simulated and observed NH3 column concentrations due to these factors? Some quantitative discussion or sensitivity analysis on these points would help strengthen the credibility of the posterior estimates.

2. It is not very clear to me how the current MLR framework can separate AGR and non-AGR NH$_3$ emissions. Some clarification would be helpful, as noted in the minor comments below. Generally, in each grid cell j, I would expect the temporal variations in SA_agriculture and SA_non-agriculture to be perfectly correlated and differ only in magnitude, thus they could not be separated in the regression. While WRF-Chem may simulate different day-to-day variations for SA_agriculture and SA_non-agriculture, that reflects the effects of transport from surrounding grid cells, which contradicts the assumption that transport effects are negligible. Also, it would be useful to explain why the emission corrections primarily affect the non-AGR sector. Given that non-AGR emissions are relatively small in the prior, one would expect SA_non-agriculture to be much smaller than SA_agriculture in Eqn (4). Is the regression

coefficient b significantly larger than a, and if so, what is the reason for that? Specifically, in northern Henan, the posterior results show decreased AGR but increased non-AGR emissions compared to the prior, which seems hard to understand and needs further explanation.

3. The finding of substantially higher non-agricultural (non-AGR) $NH_3$ emissions compared to prior estimates is certainly interesting and important. However, the discussion of the posterior results in Section 4 currently focuses mainly on reporting emission magnitudes and sectoral contributions, with limited interpretation of the underlying causes or contextualization within existing literature. I would encourage the authors to expand this discussion by addressing the following points:

(1) What are the potential reasons for the apparent underestimation of non-AGR $NH_3$ emissions in current bottom-up inventories? (2) What types of non-agricultural sources (e.g., industrial processes, transportation) are most likely responsible, based on current understanding? (3) How do your findings about non-AGR emissions compare with previous top-down estimates? The discussion in Lines 259–281 is helpful, but it could be further strengthened by emphasizing on observation-based or model-based studies that have investigated non-AGR $NH_3$ sources. It would also be valuable to highlight how your results build upon or differ from those studies, and what novel insights your analysis contributes to this topic.

**Minor comments:**

1. Line 26/28 and elsewhere: please remove the "·" between Tg and yr-1. Also, replace Gg mon-1 with either Gg month-1 or Gg mo-1 to follow standard unit concentions.

2. Line 37-40. The summary statement is too general. It would be more informative to highlight the insights into non-agricultural ammonia emissions and their implications.

3. Line 127: Is biomass burning emission also treated online? Just checking, as this is not commonly the case.

4. Line 134: The "last accessed" date should reflect the actual date when the data were downloaded.

5. In section 2.2, what's the overpass time of IASI data? Do you use level-2 satellite data?

6. Line 140-141:  The description is unclear. How is the neural network applied to improve the data quality, was it developed by the authors or sourced elsewhere?

7. Line 125/156: Which version of MEIC is used?

8. Line 162-163 and Section 4.1: Please clarify how the model and observations are sampled for comparison.

9. Line 179: The term "errors" is vague. Consider using clearer language such as "underestimated by 30%" or "biased low by 30%."

10. Line 181-189: Figure 6 can be described with the text here.

11. Section 3: Please be consistent in the use of statistical metrics. RMSE is used for IASI comparisons, while IOA and MFB are used for surface observations. A brief explanation of why different metrics are applied, and what each evaluates, would be helpful.

12. Line 195: You mention deriving posterior emissions for four months—how are prior/posterior simulations compared with observations across the seasons? Are the same scale factors applied to all three months in each season? Please clarify. Given that WRF-Chem simulations are available for the full year, it would be more consistent to derive monthly emissions for all 12 months, which should follow the same procedure and would not require much additional effort.

13. Line 200: In Line 138, you mentioned that IASI data were regridded to the model resolution, but here you refer to single-pixel comparisons, which is somewhat confusing. Please clarify how the satellite data were matched to model outputs.

14. Line 202: What does "regression factor" refer to? Is it the same as the emission scale factor?

15. Line 206: Is TA_satellite the monthly average or the daily average of NH3 concentrations?

16. Line 208: Should be "outside transportation, AGR emissions, non-AGR emissions, respectively'

17. Line 211: The term "control emissions" is unclear. Do you mean emissions were zeroed out? Also, please replace "cycle" with "experiment."

18. Line 214: What is A_blank used for?

19. Line 203/216: Earlier you use k for month and j for region, but later this is reversed. Please ensure consistency throughout. Also, using "grid cell j" is clearer than "region j" or "area j."

20. Line 217: It is unclear why does the regression is derived mathematically imply it needs to be corrected? Please clarify the motivation for adjusting the regression coefficients.

21. Line 218-231: The description of the correction process is not very clear. It is unclear what is meant by "goodness of fit," how the "invalid" regression coefficients are defined, and what fraction of them are removed. The phrase "make a trade-off" in Line 225 is vague and would benefit from clarification. Additionally, it is not explained how the adjustment factors $a_n$ and $b_n$ are derived or what their physical meaning is. The choice of a 30% threshold in Line 229 also seems arbitrary—particularly in high-NH$_3$ regions, where it could allow larger discrepancies between observations and simulations, but the physical basis for this threshold is not clearly explained.

---

## Referee Comment (RC3)

**Review Report:**

**Title:** *Optimizing Ammonia Emissions for PM2.5 Mitigation: Environmental and Health Co-Benefits in Eastern China*
**Authors:** Keqin Tang et al.

This manuscript presents an inverse modeling framework that combines IASI satellite observations with the WRF-Chem model to optimize ammonia ($NH_3$) emissions over Eastern China. The results suggest that prior inventories substantially underestimate $NH_3$ emissions—by approximately one-third for the year 2016. The study captures strong seasonal variability, identifies the dominant role of agricultural sources in summer, and highlights the importance of non-agricultural sources in winter. It also evaluates the downstream benefits of $NH_3$ mitigation on $PM_{2.5}$ reduction and public health.

This is a timely and policy-relevant study with important implications for emission inventories, air quality management, and health co-benefits. The manuscript is generally well organized and written, but certain aspects of the methodology, validation, and interpretation require clarification to ensure scientific rigor and transparency.

I recommend **major revisions** before the manuscript is accepted for publication.

**Major Comments**

1. **Limited Optimization Period:**
   - The top-down optimization was performed only for four months (each season). Please justify why the analysis was limited to these periods and discuss whether this may affect annual emission estimates or bias seasonal interpretations.
2. **Validation with Surface Data:**
   - While Figure S2 attempts to demonstrate agreement between model and surface observations, a time series or seasonal comparison between observations at these sites and both prior and posterior simulations would provide more clarity.
   - Consider including monthly or seasonal cycle plots at selected sites to demonstrate how well the posterior simulation captures temporal variability.
   - Scatter plots comparing prior vs. observed and posterior vs. observed $NH_3$ concentrations (similar to Figures S4–S6) should be more clearly explained in the main text.
3. **Clarification on Posterior Emission Totals:**
   - Page 13, L343: Clarify whether the 4.2 Tg emission is derived from the posterior estimate. Given the remaining model–observation gap, this number should be framed as a lower-bound estimate. Please discuss the implications.
4. **Sectoral Emission Trends and Inconsistencies:**
   - Page 8, L216–218 and L223: There seems to be a discrepancy between the statements about non-AGR and AGR emission changes. Figure 5b suggests a spring increase, typically associated with agricultural activity, yet the larger change is attributed to non-AGR sources. Please clarify the partitioning of the emission increase.

5. **Loss of High Emission Feature in SON/DJF (Figure 6):**
   o The high-emission feature at the intersection of Henan, Hebei, and Shandong disappears in SON and DJF seasons. Please explain whether this is due to real seasonal changes or limitations in the model/data.
6. **Surface Bias Reduction (Page 10, L277–278):**
   o Clarify which observational data were used (in situ surface measurements vs. satellite), whether the bias reduction is spatially averaged, and if supporting plots are available. A comparison showing seasonal variation would strengthen this claim.
7. **Table 3 Description (Page 11, L288–297):**
   o Provide more detail on the meaning of the single values listed in Table 3. What metrics are these? How do they compare across seasons and sectors?
8. **Public Health and PM$_{2.5}$ Impact:**
   o Page 11, L306: Quantify the reduction in PM$_{2.5}$ (1.5–5.7 µg/m³) as a percentage of baseline concentrations to help contextualize the health impact.

**Minor Comments**

- **Page 4, L97:** Surface data usage should be mentioned in the abstract for completeness.
- **Page 7, L164:** Briefly explain the extreme values of IOA and MFB to aid reader interpretation.
- **Page 7, L180–184:** Confirm whether these lines are referencing Figure 6.
- **Page 8, L201:** Remove extra period after "Table 2."
- **Page 9, L239:** Correct grammar: "based on both top-down and bottom-up approaches."
- **Page 10, L265:** Suggest adding the prior result value alongside the percentage difference for clarity.
- **Page 11, L281:** Reword the statement to reflect partial improvement in posterior vs. surface observations.
- **Page 13, L342:** Add missing period.
- **Page 13, L348:** Consider providing a geophysical or socioeconomic explanation (e.g., dense agriculture and livestock) for high emissions at the provincial intersection.

**Suggestions for Figures**

- **Figure 2:** Add a row showing differences (posterior – prior) for better visualization of emission changes by sector.
- **Figure 3:** Clarify what the red box highlights; also discuss large underestimates in Shandong and BTH.
- **Figure 6:** Add units to the color bar for clarity.
- **Figure S2:** Increase font size on color bar units.

**References**

- Please ensure that all relevant IASI-based ammonia studies are cited appropriately to position your work in the broader context of satellite-based NH$_3$ retrievals and applications.

**Conclusion**

This is a strong contribution to understanding ammonia emissions and their health and environmental implications in a rapidly developing region. The study is scientifically valuable, but several key clarifications and visual enhancements are needed to strengthen its conclusions and reproducibility. I recommend revision with attention to the above points.

---

## Author Comment (AC1)

**Responses to reviewer #1**

Dear Editor and Reviewer #1:

We would like to express our sincere gratitude to the editor and the reviewer for their time and invaluable evaluation on our manuscript, "Optimizing Ammonia Emissions for $PM_{2.5}$ Mitigation: Environmental and Health Co-Benefits in Eastern China" (egusphere-2025-1407). The insightful suggestions have enabled us to significantly improve the quality of our work. We have addressed all comments and have revised the manuscript accordingly. The reviewer comments are presented in blue, our point-by-point responses are in black, and the corresponding revisions in the manuscript are highlighted in red.

**Major comments:**

1. The authors attribute the ammonia emissions underestimate in the model almost entirely to non-agricultural emissions (Figure 2). However, temporally, the authors put the largest posterior increase in ammonia emissions in spring and summer (Figure 5) when I would expect agricultural emissions to be most important (fertilization time + favorable meteorology). At the same time, your sources are close or overlap in space (Figure 2), especially considering the smoothness of the modeled and observational total column $NH_3$ (Figure 6). Moreover, because of co-linearity, I am not sure how well the MLR (Eqn 4) can separately fit the alpha and beta parameters and thus separate source sectors. For these reasons, I am skeptical of the source attribution in this study. I am more confident in total ammonia emissions magnitudes.

**Response:**

We thank the reviewer for this comment. We acknowledge the reviewer's point that the regional changes in agricultural $NH_3$ emissions between prior and posterior inventories are small. This phenomenon could be explained by the spatial heterogeneity of changes in agricultural emissions. For example, our posterior model results for spring show a decrease in agricultural emissions in Henan, while simultaneously proposing a substantial increase of 242.8 Gg in the Yangtze River Delta region. This regional redistribution could improve the model's ability in better matching with observations.

Therefore, the large rise in total emissions in spring is a combination of these regionally specific agricultural adjustments and a significant, spatially broad increase in the non-agricultural sector. It is also important to note that even with this large non-

AGR correction, agriculture remains the dominant source of emissions in spring in our posterior inventory (accounting for 84.1%), reflecting the overwhelming importance of fertilization activity in this season.

The use of multiple linear regression (MLR) for source apportionment is a well-established approach in atmospheric science (Qi et al., 2024; Shu and Lam, 2011; Trošić Lesar and Filipčić, 2023) and can identify different physical sources. The fundamental principle of using regression for source apportionment is that different sources can be statistically distinguished if they possess unique spatial "fingerprints".

In our study, the high-concentration regions resulting from AGR and non-AGR emissions do not spatially align (Figure R1.1). The overall spatial correlation between the NH$_3$ columns simulated from these two sources is low (r = 0.35) and is near zero in the high-concentration regions (r = 0.03). This significant dissimilarity provides a robust statistical basis for the MLR model to distinguish their relative contributions.

[Figure]

Figure R1.1: Spatial distribution of prior simulated NH$_3$ column concentrations from agricultural and non-agricultural sources.

*Revision in Section 4.1:*

*In each iterative calculation, the monthly average satellite-derived NH$_3$ column concentration served as the target, and multiple linear regression (MLR) was applied to calculate the corresponding regression factors for AGR and non-AGR emissions (Figure S3). This separation of sectors by MLR is effective because their respective spatial distributions are distinct and largely uncorrelated (r = 0.35).*

*Revision in Section 4.2:*

*In multiple iterations, the framework optimizes the relative mix of the two sources to better match the observed spatial patterns. For instance, the spatial correlation between model and observation in Henan increased from 0.47–0.58 (prior simulations)*

*to 0.64–0.90 (posterior simulations).*

2. The MLR does not account for chemistry occurring between emission and observation. Could you comment on how this affects the results?

**Response:**

We thank the reviewer for this insightful question, which allows us to clarify the role of atmospheric chemistry within our inversion framework and discuss the associated uncertainties.

First, we emphasize that our methodology inherently accounts for atmospheric chemistry. While the MLR component is a statistical tool, our approach is not a simple regression directly linking emissions to observed columns. Instead, the MLR operates within an iterative framework dynamically coupled with the full WRF-Chem model. Crucially: (1) The inputs to our regression (the SA variables) are the simulated $NH_3$ column concentrations generated by WRF-Chem. This means that within each iteration, WRF-Chem explicitly simulates all complex, non-linear chemical transformations (including gas-to-particle partitioning and aerosol formation) and transport processes occurring between emission and the resulting atmospheric concentration. (2) The MLR then acts solely as an efficient optimization tool, adjusting the emission inputs based on the outputs of this chemically comprehensive model.

To directly discuss the model's capacity in characterizing concentrations of secondary inorganic aerosols (SIA), we conducted comparisons using in-situ measurements at a representative site in Beijing (39°59′21″N, 116°18′25″E).

The evaluation results are summarized in Table R1. It is revealed that the posterior $NH_3$ emissions increase $NH_4^+$ concentration from 4.71 µg m$^{-3}$ to 4.95 µg m$^{-3}$, which is closer to the observed average (5.69 µg m$^{-3}$). The simulated mean $NO_3^-$ concentration with 9.59 µg/m³ also better matches the observed level (9.44 µg m$^{-3}$).

The WRF-Chem model performs moderately well in capturing the observed $SO_4^{2-}$ concentration (7.74 µg m$^{-3}$) in both simulations (5.81-5.84 µg m$^{-3}$). The model underestimation could be attributed to the missing formation mechanism of sulfate such as transition metal ions (TMI)-catalyzed and photosensitized oxidation of $SO_2$ on aerosol surfaces (Cai et al., 2024; Wang et al., 2021, 2020). Although this underestimation of sulfate might lead to our posterior $NH_3$ emission estimates being

conservatively low, we find that the model still reproduces the total secondary inorganic aerosol (SIA) concentrations well, with an overall bias of only -11.0%. This good performance in simulating the total aerosol sink for ammonia suggests that the uncertainty propagated to the final emission estimates from these chemical pathways is limited.

In summary, our framework inherently accounts for chemistry through its tight coupling with WRF-Chem. The evaluation against SIA observations confirms the chemical plausibility of our results for nitrate and ammonium, while highlighting specific uncertainties in sulfate chemistry. These uncertainties suggest our posterior $NH_3$ emissions may represent a conservative estimate. We have incorporated this discussion into the revised manuscript.

Table R1: Comparison of prior and posterior simulated surface concentrations with in-situ observations for major secondary inorganic aerosol components (sulfate, nitrate, and ammonium) in Beijing. All values are in $\mu g\ m^{-3}$.

| | Prior simulation | Posterior simulation | observation |
|---|---|---|---|
| nitrate | 8.82 | 9.59 | 9.44 |
| ammonium | 4.71 | 4.95 | 5.69 |
| sulfate | 5.81 | 5.84 | 7.74 |

*Revision in Section 2.2:*

*Furthermore, speciated inorganic aerosol data from a representative site in Beijing were collected to evaluate the model's capacity in characterizing the formation of secondary inorganic aerosols (Tan et al., 2018).*

*Revision in Section 4.1:*

*Finally, the entire process is iteratively repeated, a framework that captures the overall non-linear atmospheric response by combining the dynamic simulation of non-linear chemistry within each WRF-Chem step with the collective behavior of multiple iterations.*

*Revision in Section 4.2:*

*Additionally, uncertainties from the model's chemical mechanisms and the*

*influence of nearby grid transport also contribute to this gap, but the overall impact on the final estimate is limited.*

***Revision in Section 4.3:***

*To further characterize the model's chemical performance beyond total PM$_{2.5}$, we also evaluated the simulation of secondary inorganic aerosol (SIA) components against in-situ measurements from a representative site in Beijing (Table S7). The evaluation shows that the posterior NH$_3$ emissions improved the simulation of ammonium and nitrate, reducing the bias between simulated and observed concentrations. Although the model underestimates sulfate, likely due to missing formation mechanisms (Cai et al., 2024; Wang et al., 2021, 2020), the total SIA concentration is well reproduced with an overall bias of only -11.0%.*

***Revision in Supplementary:***

***Table S7.*** *Comparison between the prior and posterior simulated inorganic aerosol concentrations with in-situ measurements in Beijing. All value units are µg m$^{-3}$.*

|  | Prior simulation | Posterior simulation | observation |
|---|---|---|---|
| nitrate | 8.82 | 9.59 | 9.44 |
| ammonium | 4.71 | 4.95 | 5.69 |
| sulfate | 5.81 | 5.84 | 7.74 |

**Minor comments:**

1. Line 75: It is not clear to me what the 1%–50% figure represents here. Is this reduction in $PM_{2.5}$ per unit $NH_3$ emissions reduced?

**Response:**

We thank the reviewer for this question. The 1%–50% range in our text is intended to summarize the breadth of these varying findings reported in the studies we referenced. It represents the range of discrepancies or varying outcomes found across the cited literature when assessing the impact of $NH_3$ emission reductions on $PM_{2.5}$ levels. This variability arises from differences in study methodologies, including models, underlying emission inventories, regions, and seasons analyzed. To enhance clarity, we have revised the relevant sentence in the manuscript to explicitly state that this range reflects the spectrum of outcomes reported in the referenced studies.

*Revision in Section 1:*

*The uncertainty in the emission estimation further contributes to significant discrepancies, reflecting the range of results (1%–50%) reported in the literature, in assessing the impacts of $NH_3$ reduction on $PM_{2.5}$ level (Guo et al., 2018, 2024; Li et al., 2024; Liu et al., 2019, 2021, 2023; Pan et al., 2024; Zhang et al., 2022).*

2. Which version of the IASI $NH_3$ data do you use?

**Response:**

We appreciate the reviewer's attention to this detail. We have clarified in the revised text that the IASI $NH_3$ data used in this study is version 3.0.

*Revision in Section 2.2:*

*We obtained the total column density of $NH_3$ from the passive satellite remote-sensing product of the Infrared Atmospheric Sounding Interferometer (IASI) (version 3.0, https://iasi.aeris-data.fr/nh3/, last accessed on December 2020) as the observational constraint.*

3. Lines 165–166: What do the index of agreement and mean fractional bias mean, intuitively?

**Response:**

We appreciate the reviewer's suggestion to clarify these metrics. The manuscript now includes expanded intuitive explanations:

(1) Index of Agreement (IOA)

The IOA quantifies the overall simulation skill with values ranging from 0 to 1, where 1 indicates perfect match between simulated and observed data while 0 denotes complete disagreement. This metric evaluates both magnitude accuracy and spatial pattern consistency, making a higher IOA value indicative of better model performance. In our context, an increased IOA in posterior simulations versus prior runs confirms improved representation of $NH_3$ columns.

(2) Mean Fractional Bias (MFB)

The MFB diagnoses systematic model bias with values centered at 0. A value of 0 signifies no average bias, positive values indicate model overestimation, and negative values reflect underestimation. The absolute magnitude measures bias severity, where smaller absolute values correspond to reduced systematic error.

In this study, we use IOA to evaluate global consistency between simulated and satellite-observed $NH_3$ columns, while MFB specifically quantifies directional bias tendencies. These clarifications have been incorporated into the revised manuscript.

*Revision in Section 3:*

*The IOA quantifies the overall model skill, where a value of 1 indicates a perfect match and 0 denotes complete disagreement. The MFB diagnoses systematic model bias, where positive values indicate overestimation, negative values indicate underestimation, and 0 signifies no average bias.*

4. Line 167: What does the C mean in these equations? I presume ammonia column concentrations?

**Response:**

We thank the reviewer for highlighting this ambiguity. In the equations, C is defined as the **C**oncentration of the target pollutant, with its specific meaning determined by the evaluation context: (1) $NH_3$ total column concentrations (satellite comparison); (2) Surface $NH_3$ concentrations; (3) Other pollutants (e.g., surface $PM_{2.5}$ $SO_2$, $NO_2$)

We have revised the manuscript to explicitly clarify this generalized notation and its context-dependent applications.

*Revision in Section 3:*

*They were calculated following Eq. 1~3, where C represents the concentration of the target pollutant (e.g., NH₃ total column or surface concentrations), and subscripts s, o and N represent simulations, observations, and the number of samples, respectively.*

5. How do you convert simulated NH₃ to total column densities comparable to IASI (e.g. the SA in line 205)?

**Response:**

We thank the reviewer for requesting methodological clarification. The conversion of simulated NH₃ to total column densities (SA variables) is now detailed in the Supporting Information.

The WRF-Chem model outputs NH₃ concentrations as a volume mixing ratio (in ppmv) for each model layer. To convert these layer-specific concentrations into a total vertical column density (VCD) comparable to IASI satellite retrievals, the subsequent process was followed.

First, the thickness of each model layer ($\Delta Z$) must be determined. As our WRF-Chem setup uses a terrain-following hybrid sigma-pressure coordinate system, the geopotential height (Z) of each model level is calculated from the model's perturbation geopotential (PH) and base-state geopotential (PHB), divided by the acceleration due to gravity ($g \approx 9.8$ m s$^{-2}$):

$$Z = \frac{PH + PHB}{g}$$

The thickness of an individual model layer, k, is then the difference in geopotential height between its upper and lower boundaries: $\Delta Z_k = Z_{k+1} - Z_k$

Moreover, the NH₃ volume mixing ratio in each layer is converted to a number density ($N_{NH3}$, in molecules cm$^{-3}$), using the pressure and temperature of that specific model layer. Finally, the total NH₃ vertical column density (VCD, in molecules cm$^{-2}$) is calculated by integrating the vertical column density in each layer of the model. In our discrete model layers, this is achieved by summing the partial column of each layer, which is the product of the number density ($N_{NH3,k}$) and the layer thickness ($\Delta Z_k$).

*Revision in Section 3:*

*The detailed method for calculating NH₃ total column concentrations and surface volume concentrations from WRF-Chem is provided in Text S1.*

*Revision in Supplementary:*

*TEXT S1*

*For comparison with IASI satellite retrievals, the total vertical column density (VCD) was calculated from the model's layer-specific output. The WRF-Chem model outputs NH$_3$ concentrations as a volume mixing ratio (in ppmv) for each model layer. To convert these layer-specific concentrations into a VCD, the subsequent process was followed.*

*First, the thickness of each model layer ($\Delta Z$) must be determined. As our WRF-Chem setup uses a terrain-following hybrid sigma-pressure coordinate system, the geopotential height (Z) of each model level is calculated from the model's perturbation geopotential (PH) and base-state geopotential (PHB), divided by the acceleration due to gravity (g ≈ 9.8 m s$^{-2}$):*

$$Z = \frac{PH + PHB}{g}$$

*The thickness of an individual model layer, k, is then the difference in geopotential height between its upper and lower boundaries: $\Delta Z_k = Z_{k+1} - Z_k$*

*Moreover, the NH$_3$ volume mixing ratio in each layer is converted to a number density ($N_{NH3}$, in molecules cm$^{-3}$), using the pressure and temperature of that specific model layer. Finally, the total NH$_3$ vertical column density (VCD, in molecules cm$^{-2}$) is calculated by integrating the vertical column density in each layer of the model. In our discrete model layers, this is achieved by summing the partial column of each layer, which is the product of the number density ($N_{NH3,k}$) and the layer thickness ($\Delta Z_k$).*

6. Figure 3: What does the red box represent?

**Response:**

We appreciate this suggestion to enhance figure clarity. The red box highlights the range we consider to represent good model performance. Specifically, it delineates the area where the Root Mean Square Error (RMSE) is less than 10, and the ratio of simulated-to-observed NH$_3$ column concentration is between 0.7 and 1.3 (±30% error margin). We have added this clarification to the figure caption in the revised manuscript.

*Revision in Section Figure 3:*

*Figure 3. Scatter plots of the prior and posterior NH$_3$ total column data versus*

*IASI retrievals. Each point represents prior (or posterior) data for a specific season and a specific region. Circles, triangles, rhombuses, and rectangles correspond to the BTH, Henan, Shandong, and YRD regions, respectively. Orange and blue markers represent a prior and a posterior data, respectively. The red box indicates the performance area, with a model error within ±30% and an RMSE below 10(×10$^{15}$ molec cm$^{-2}$).*

7. Line 214: What is A_blank? Also, what are these A variables more generally? They are not defined.

**Response:**

Thank you for this question, which allows us to clarify these important methodological details. To clarify, the A variables (e.g., $A_{agr}$, $A_{non-agr}$) represent the total simulated $NH_3$ column concentrations that result from each of the sensitivity simulations listed in Table 2.

Specifically, The $A_{blank}$ case refers to a simulated $NH_3$ total column in which all anthropogenic emissions within the study domain were turned off. The purpose of this simulation was to establish a blank line concentration field, which represents the influence of the chemical boundary conditions provided to our model domain.

We have revised the manuscript to provide explicit definitions.

*Revision in Section 4.1:*

*The $SA_{agriculture}{}_{i-1}^{j,k}$, $SA_{non-agriculture}{}_{i-1}^{j,k}$, and $SA_{transport}{}^{j,k}$ are calculated by subtracting $A_{blank}$ from $A_{agr}$, $A_{non-agr}$, and $A_{transport}$, respectively. Here, symbols A represent the total simulated $NH_3$ column concentrations that result from each of the sensitivity simulations listed in Table 1. Specifically, the modeling case $A_{blank}$ refers to a simulated $NH_3$ total column in which all anthropogenic emissions within the study domain were zeroed out. The purpose of this simulation was to establish background concentrations, which represents the influence of the chemical boundary conditions provided to our model domain.*

8. Equation 4: The SA variables are referring to simulated values, right? This is not clear.

**Response:**

We confirm that the SA variables in Equation 4 represent simulated $NH_3$ column

concentrations from specific source categories. We have revised the manuscript to explicitly state this definition and eliminate ambiguity.

*Revision in Section 4.1:*

*where, $TA_{satellite}{}^{j,k}$ denotes the monthly average of total NH₃ column density retrieved from the IASI satellite data, and $SA_{transport}{}^{j,k}$, $SA_{agriculture}{}_{i-1}^{j,k}$ and $SA_{non-agriculture}{}_{i-1}^{j,k}$ stand for the simulated total column concentration of NH₃ contributed by AGR emissions, non-AGR emissions, and outside transportation, respectively.*

9. Line 219: How is D_i^j,k calculated, in terms of the variables already given?

**Response:**

Thank you for this question regarding the specific details of our methodology. The variable $D_i^{j,k}$ represents the relative bias between the total simulated NH₃ column concentration and the satellite-retrieved observation for a given iteration, month, and region. It is calculated as the difference between the mean simulated column and the mean satellite retrieval column, normalized by the mean satellite retrieval column.

*Revision in Section 4.1:*

*we need to correct for this regression coefficient. The biases between the model simulation and the satellite retrievals were calculated as $D_i^{j,k}$. Specifically, it is the difference between the mean simulated column and the mean satellite retrieval, divided by the mean satellite retrieval.*

10. Line 221: What do you mean by "excessive residuals"? What is the judgement coefficient and how is it calculated?

**Response:**

We thank the reviewer for highlighting this key quality control aspect. We have revised the manuscript to provide these specific details.

In our regression analysis, a residual is defined as the difference between the observed value (i.e., the satellite-derived NH₃ column) and the value predicted by the MLR model. To objectively identify what we termed "excessive residuals," we utilize the 95% confidence interval of the residual for each individual fit.

Our criterion is as follows: if the 95% confidence interval of a residual does not

contain zero, such a case is flagged as having an "excessive residual." This means the linear model provides a poor fit for that specific data point, and the resulting regression coefficients are deemed unreliable. Consequently, these coefficients are rejected and not used for the emission update in that iteration. In the following Figure R1.2, we can see that red represents outliers and needs to be discarded.

[Figure]

Figure R1.2:Distribution of residuals and their 95% confidence intervals. Each point represents the residual value for a given sample, and the error bars represent the 95% confidence interval of the residual. Green points represent valid fits, while red points are outliers rejected based on the criterion that their confidence interval does not contain zero.

***Revision in Section 4.1:***

*The regression coefficients with excessive residuals, defined as cases where the 95% confidence interval of the residual does not contain zero, are removed to increase credibility.*

11. Line 222: What goodness of fit test/metric do you use? How did you pick this 0.3– 1 acceptability range?

**Response:**

We thank the reviewer for requesting methodological clarification. The "goodness of fit" metric we used is the coefficient of determination, commonly known as R-squared ($R^2$). The $R^2$ value quantifies the proportion of the variance in the dependent variable that is predictable from the independent variables (the simulated $NH_3$ columns from AGR and non-AGR sources). This metric ranges from 0 to 1, with higher values

indicating superior model performance.

The acceptability range of 0.3–1 was chosen as a practical criterion within our iterative framework. We established this criterion because regressions with $R^2 < 0.3$ exhibit insufficient explanatory power (indicating >70% unexplained variance), which introduces destabilizing noise into emission adjustments. By excluding such statistically unreliable results from our inventory updates, we maintain algorithm stability while reducing required iteration cycles.

We have now explicitly stated in the manuscript that the metric used is R-squared and have clarified the purpose of this threshold.

*Revision in Section 4.1:*

*Concurrently, the goodness of fit of the regression is calculated as the coefficient of determination (R-square, $R^2$). To maintain algorithm stability, regressions with an $R^2$ less than 0.3 are deemed invalid and excluded from the emission update, as they exhibit insufficient explanatory power (indicating >70% unexplained variance) and introduce destabilizing noise into the adjustments.*

12. How do you perform the iterations (lines 226-229)? I presume you increment agricultural and non-agricultural emissions for each grid cell by following the fitted alpha and beta parameters, but how exactly and by what magnitude? What do you do for the next increment in the case where you reject the MLR results?

**Response:**

We appreciate the opportunity to clarify our iterative optimization procedure. The revised manuscript now details this process in Section 4.1.

The iteration is performed by sequentially updating the emission inventory and re-running the WRF-Chem simulation to produce a new concentration field. The magnitude of the emission update in each step is determined by the final adjustment factors (a and b) derived from our corrected MLR analysis. These factors are used as direct scaling multipliers for the emissions. For instance, in the event of the analysis determining a final adjustment factor of 1.3 for the agricultural sector, the new agricultural emission will be set to 1.3 times the value of the previous iteration.

Regarding the case where the MLR results for an iteration are rejected, the process is designed to be conservative. In such instances, the adjustment factors for that specific

grid cell are considered invalid, and the emissions are kept unchanged from the previous iteration. The algorithm then proceeds using this unadjusted emission value as the input for the next step. We have added these specific details about the emission update procedure to the methodology section of our manuscript to improve its clarity.

*Revision in Section 4.1:*

*If a regression is valid, the adjustment factors a and b are set to the new regression coefficients; if invalid, the factors are kept unchanged from the previous iteration. The updated emissions for the next iteration are then calculated by multiplying the emissions from the previous step by these adjustment factors.*

13. What does Figure 5 look like if you split the prior and posterior bar plots up into AGR and nonAGR emissions? I am curious about how the source attribution varies with season.

**Response:**

We thank the reviewer for this valuable suggestion to enhance seasonal attribution analysis. As suggested, we have generated a supplementary figure (Figure S6) decomposing prior and posterior emissions into agricultural (AGR) and non-agricultural (non-AGR) sources by season.

As the new figure illustrates, the source attribution varies significantly by season. Agricultural emissions are the dominant contributor during the spring and summer months, which is consistent with the timing of fertilizer application and higher temperatures that promote volatilization. In contrast, the relative contribution from non-agricultural sources increases substantially in the winter. This winter increase is largely attributed to higher emissions from fossil fuel combustion and other industrial activities that are more pronounced during the cold season.

*Revision in Section 4.2:*

*At the specific-source scale (Figure S6), AGR $NH_3$ emissions show similar seasonal patterns with the total $NH_3$ emissions, higher in summer and spring. In contrast, non-AGR $NH_3$ are highest in winter and fall because fossil fuel combustion-related emissions are higher in cold season, while the lowest emissions occur in summer.*

*Revision in Supplementary:*

[Figure]

*Figure S5: Seasonal comparison of prior and posterior NH₃ emissions from AGR and non-AGR sources.*

14. Line 291: What are the units of the RMSE? Is this referring to surface observations or satellite columns?

**Response:**

Thank you for your detailed review. The units for the RMSE are consistent with the units of the quantities being compared. Therefore, the RMSE has units of molecules $cm^{-2}$ when evaluating against satellite total columns, and units of $\mu g\ m^{-3}$ when evaluating against surface concentration measurements. We have revised the manuscript to explicitly state the appropriate units in each instance to avoid ambiguity.

*Revision in Section 4.3:*

*More than 80% of the points fall in the range where the simulation-to-observation ratio is between 0.7 and 1.3 and the RMSE is less than $10 \times 10^{15}$ molec $cm^{-2}$.*

**Typographical comments:**

1. Line 52: Figure should read "55.0%"

**Response:** Thank you. The text has been corrected as suggested.

*Revision in Section 1:*

*Ammonia (NH₃), a key precursor of PM₂.₅, neutralizes sulfuric acid (H₂SO₄) and nitric acid (HNO₃), leading to the formation of secondary inorganic aerosols (SIA), which contributes 19.4%–55.0% of the total PM₂.₅ (Huang et al., 2014; Liu et al., 2022b; Wang et al., 2016; Wei et al., 2023; Zheng et al., 2015; Zhou et al., 2022).*

2. Throughout: the en dash (–) should be used for numerical ranges instead of the tilde (~).

**Response:** We thank the reviewer for this helpful comment, and the manuscript has been revised accordingly.

3. Lines 179–180: this sentence ends abruptly (what is the RMSE of 10 referring to; what are its units).

**Response:** Thank you for the comment. The units for the Root Mean Square Error (RMSE) are the same as those for the $NH_3$ column concentration ($\times10^{15}$ molec cm$^{-2}$). We have revised the sentence in the manuscript to include the appropriate units.

*Revision in Section 3:*

*Most simulated NH₃ total column concentrations are underestimated by more than 30% compared with the observed values by satellite with the associated RMSE exceeding $10\times10^{15}$ molec cm$^{-2}$.*

4. Line 226: I presume you mean alpha and beta as in Equation 4, not a and b here?

**Response:**

Thank you for this comment. In our methodology, α and β are the initial regression coefficients derived directly from the Multiple Linear Regression (MLR) in each iterative step. In contrast, 'a' and 'b' represent the final, corrected adjustment factors that are used to update the emission inventory.

These final factors (a, b) are derived from the initial coefficients (α, β) after a correction process that accounts for the goodness of fit and regression residuals. We have updated the methodology section to explicitly define 'a' and 'b' and to better describe how these adjustment factors are obtained. This clarification should improve

the reader's understanding of our method.

*Revision in Section 4.1:*

*If a regression is valid, the adjustment factors a and b are set to the new regression coefficients; if invalid, the factors are kept unchanged from the previous iteration.*

**References**

Cai, S., Liu, T., Huang, X., Song, Y., Wang, T., Sun, Z., Gao, J., and Ding, A.: Important Role of Low Cloud and Fog in Sulfate Aerosol Formation During Winter Haze Over the North China Plain, Geophysical Research Letters, 51, e2023GL106597, https://doi.org/10.1029/2023GL106597, 2024.

Guo, H., Otjes, R., Schlag, P., Kiendler-Scharr, A., Nenes, A., and Weber, R. J.: Effectiveness of ammonia reduction on control of fine particle nitrate, Atmospheric Chemistry and Physics, 18, 12241–12256, https://doi.org/10.5194/acp-18-12241-2018, 2018.

Guo, Y., Zhang, L., Winiwarter, W., Van Grinsven, H. J. M., Wang, X., Li, K., Pan, D., Liu, Z., and Gu, B.: Ambitious nitrogen abatement is required to mitigate future global PM2.5 air pollution toward the World Health Organization targets, One Earth, 7, 1600–1613, https://doi.org/10.1016/j.oneear.2024.08.007, 2024.

Li, B., Liao, H., Li, K., Wang, Y., Zhang, L., Guo, Y., Liu, L., Li, J., Jin, J., Yang, Y., Gong, C., Wang, T., Shen, W., Wang, P., Dang, R., Liao, K., Zhu, Q., and Jacob, D. J.: Unlocking nitrogen management potential via large-scale farming for air quality and substantial co-benefits, National Science Review, 11, nwae324, https://doi.org/10.1093/nsr/nwae324, 2024.

Liu, M., Huang, X., Song, Y., Tang, J., Cao, J., Zhang, X., Zhang, Q., Wang, S., Xu, T., Kang, L., Cai, X., Zhang, H., Yang, F., Wang, H., Yu, J. Z., Lau, A. K. H., He, L., Huang, X., Duan, L., Ding, A., Xue, L., Gao, J., Liu, B., and Zhu, T.: Ammonia emission control in China would mitigate haze pollution and nitrogen deposition, but worsen acid rain, Proc Natl Acad Sci U S A, 116, 7760–7765, https://doi.org/10.1073/pnas.1814880116, 2019.

Liu, Z., Zhou, M., Chen, Y., Chen, D., Pan, Y., Song, T., Ji, D., Chen, Q., and Zhang, L.: The nonlinear response of fine particulate matter pollution to ammonia emission reductions in North China, Environmental Research Letters, https://doi.org/10.1088/1748-9326/abdf86, 2021.

Liu, Z., Rieder, H. E., Schmidt, C., Mayer, M., Guo, Y., Winiwarter, W., and Zhang, L.: Optimal reactive nitrogen control pathways identified for cost-effective PM2.5 mitigation in Europe, Nat Commun, 14, 4246, https://doi.org/10.1038/s41467-023-39900-9, 2023.

Pan, D., Mauzerall, D. L., Wang, R., Guo, X., Puchalski, M., Guo, Y., Song, S., Tong, D., Sullivan, A. P., Schichtel, B. A., Collett, J. L., and Zondlo, M. A.: Regime shift in secondary inorganic aerosol formation and nitrogen deposition in the rural United States, Nat. Geosci., 17, 617–623, https://doi.org/10.1038/s41561-024-01455-9, 2024.

Qi, L., Zheng, H., Ding, D., and Wang, S.: A comparison of meteorological

normalization of PM2.5 by multiple linear regression, general additive model, and random forest methods, Atmospheric Environment, 338, 120854, https://doi.org/10.1016/j.atmosenv.2024.120854, 2024.

Shu, Y. and Lam, N. S. N.: Spatial disaggregation of carbon dioxide emissions from road traffic based on multiple linear regression model, Atmospheric Environment, 45, 634–640, https://doi.org/10.1016/j.atmosenv.2010.10.037, 2011.

Tan, T., Hu, M., Li, M., Guo, Q., Wu, Y., Fang, X., Gu, F., Wang, Y., and Wu, Z.: New insight into PM2.5 pollution patterns in Beijing based on one-year measurement of chemical compositions, Science of The Total Environment, 621, 734–743, https://doi.org/10.1016/j.scitotenv.2017.11.208, 2018.

Trošić Lesar, T. and Filipčić, A.: Prediction of the SO2 Hourly Concentration for Sea Breeze and Land Breeze in an Urban Area of Split Using Multiple Linear Regression, Atmosphere, 14, 420, https://doi.org/10.3390/atmos14030420, 2023.

Wang, W., Liu, M., Wang, T., Song, Y., Zhou, L., Cao, J., Hu, J., Tang, G., Chen, Z., Li, Z., Xu, Z., Peng, C., Lian, C., Chen, Y., Pan, Y., Zhang, Y., Sun, Y., Li, W., Zhu, T., Tian, H., and Ge, M.: Sulfate formation is dominated by manganese-catalyzed oxidation of SO2 on aerosol surfaces during haze events, Nat Commun, 12, https://doi.org/10.1038/s41467-021-22091-6, 2021.

Wang, X., Gemayel, R., Hayeck, N., Perrier, S., Charbonnel, N., Xu, C., Chen, H., Zhu, C., Zhang, L., Wang, L., Nizkorodov, S. A., Wang, X., Wang, Z., Wang, T., Mellouki, A., Riva, M., Chen, J., and George, C.: Atmospheric Photosensitization: A New Pathway for Sulfate Formation, Environ. Sci. Technol., 54, 3114–3120, https://doi.org/10.1021/acs.est.9b06347, 2020.

Zhang, Z., Yan, Y., Kong, S., Deng, Q., Qin, S., Yao, L., Zhao, T., and Qi, S.: Benefits of refined NH3 emission controls on PM2.5 mitigation in Central China, Sci Total Environ, 814, 151957, https://doi.org/10.1016/j.scitotenv.2021.151957, 2022.

---

## Author Comment (AC2)

**Responses to reviewer #2**

Dear Editor and Reviewer #2:

We greatly appreciate your consideration and the reviewer's insightful and constructive comments on the manuscript "Optimizing Ammonia Emissions for PM$_{2.5}$ Mitigation: Environmental and Health Co-Benefits in Eastern China" (egusphere-2025-1407). We have carefully revised the manuscript to address all the comments described below. Reviewer comments are shown in blue. Our responses are shown in black. The revised texts are shown in red.

**Major comments:**

1. Although the authors do not mention, it seems to me that the MLR-based approach assumes a linear relationship between NH$_3$ emissions and column concentrations and attributes all discrepancies between satellite observed and simulated NH$_3$ columns to local NH$_3$ emissions. It is important for the authors to clarify these assumptions and discuss the associated uncertainties. Specifically, (1) How might the nonlinear response of NH$_3$ concentrations to emissions affect the results? For example, do the summed contributions from individual sources (SA$_{agriculture}$ + SA$_{non-agriculture}$ + SA$_{transport}$) approximate the simulated total NH$_3$ column from the prior? (2) To what extent could transport of emissions from nearby grid cells influence the posterior results and cause spatial misattribution of emissions? (3) How might uncertainties in emissions of other pollutants (e.g., SO$_2$, NO$_x$) or in the model's representation of inorganic aerosol formation, impact the posterior estimates of NH$_3$ emissions? Could the large discrepancies between prior simulated and observed NH$_3$ column concentrations due to these factors? Some quantitative discussion or sensitivity analysis on these points would help strengthen the credibility of the posterior estimates.

**Response**:

(1) We thank the reviewer for this insightful question. We acknowledge that the relationship between NH$_3$ emissions and atmospheric concentrations is inherently non-linear. Ambient SO₂ and NO$_x$ lead to rapid gas-to-particle partitioning, indicating that a significant portion of newly emitted NH$_3$ is quickly converted to particulate ammonium. This "buffering effect" results in the non-linear response of the gaseous NH$_3$ column to emission changes.

In this study, the WRF-Chem-based iterative algorithm we employed, while

utilizing multiple linear regression at each iteration step, is capable of capturing the nonlinear characteristics of $NH_3$. Specifically, (1) the WRF-Chem model dynamically simulates the nonlinear chemical process governing the response of $NH_3$ column to emission changes during each linear iteration step; and (2) the collective behavior of these multiple linear iterations enables the representation of the overall nonlinear characteristics. As shown in Figure R2.1, by plotting intermediate results from ten regression iterations of non-agricultural emission adjustments over the study region during autumn, we demonstrate non-constant emission adjustment factors and non-proportional concentration responses, empirically confirming our algorithm's capacity to resolve non-linear atmospheric feedbacks.

Regarding the additivity of concentration responses raised by the reviewer, we acknowledge that the sum of $NH_3$ columns from individual source simulations (i.e., $SA_{agriculture} + SA_{non\text{-}agriculture} + SA_{transport}$) deviates from the total column simulated with combined prior emissions. This discrepancy (5.4%–15.7%) is a consequence of the non-linear chemical feedback within the atmospheric system. To ensure methodological consistency, we explicitly state that all evaluations and satellite-based comparisons exclusively use results from simulation with the unified posterior emission inventory, avoiding any summation of segmented source contributions.

We have incorporated these clarifications into the manuscript to make our methodology and its approach to handling the system's non-linearities clearer for the reader.

[Figure]

**Figure R2.1.** Scatter plot of $NH_3$ total column concentration versus $NH_3$ emission intensity for several steps of the iterative process. Arrows indicate the sequence of iterations.

(2) Thank you for pointing this out. To quantitatively address your concern about the potential spatial misattribution of emissions due to transport, we conducted a sensitivity experiment to evaluate the transport impact of $NH_3$ from upstream regions on the $NH_3$ column concentrations in downstream regions. We focused our analysis on July, a month with high $NH_3$ concentrations, and selected the Yangtze River Delta (YRD) region as the representative upwind source area.

The sensitivity experiment includes two simulations using posterior emissions. One simulation runs with all emissions included, and the other runs in which the $NH_3$ emissions from the YRD region were zeroed out. By comparing these two simulations, we can quantify the contribution of YRD emissions to $NH_3$ column concentrations in the downstream region through transport.

As shown in Figure R2.2, the transport contribution in regions closely adjacent to the YRD border is approximately 15%. However, the average contribution from the YRD to the broader downwind area was found to be 3.8%, indicating that the limited influence of regional transport. To clarify this point, we have expanded the discussion of transport-related uncertainty in the manuscript.

[Figure]

**Figure R2.2.** Transport contribution from the YRD region to $NH_3$ column concentrations in downstream study areas in July.

(3) Thank you for your comment. We show the comparison between the simulated $SO_2$ and $NO_2$ against surface observations from 80 sites across 9 cities in Table 3. The model presents a good performance in reproducing the concentration levels of these

precursors both in prior and posterior simulations. To further discuss the model's capacity in characterizing concentrations of secondary inorganic aerosols (SIA), we conducted comparisons using in-situ measurements at a representative site in Beijing (39°59′21″N, 116°18′25″E).

The evaluation results are summarized in Table R2.1. It is revealed that the posterior $NH_3$ emissions increase $NH_4^+$ concentration from 4.71 µg m$^{-3}$ to 4.95 µg m$^{-3}$, which is closer to the observed average (5.69 µg m$^{-3}$). The simulated mean $NO_3^-$ concentration with 9.59 µg/m³ also better matches the observed level (9.44 µg m$^{-3}$).

The WRF-Chem model performs moderately well in capturing the observed $SO_4^{2-}$ concentration (7.74 µg m$^{-3}$) in both simulations (5.81-5.84 µg m$^{-3}$). The model underestimation could be attributed to the missing formation mechanism of sulfate such as transition metal ions (TMI)-catalyzed and photosensitized oxidation of $SO_2$ on aerosol surfaces (Cai et al., 2024; Wang et al., 2021, 2020). Although this underestimation of sulfate might lead to our posterior $NH_3$ emission estimates being conservatively low, we find that the model still reproduces the total secondary inorganic aerosol (SIA) concentrations well, with an overall bias of only -11.0%. This good performance in simulating the total aerosol sink for ammonia suggests that the uncertainty propagated to the final emission estimates from these chemical pathways is slight.

**Table R2.1.** Comparison between the prior and posterior simulated inorganic aerosol concentrations with in-situ measurements in Beijing. All value units are µg m$^{-3}$.

|  | Prior simulation | Posterior simulation | observation |
|---|---|---|---|
| nitrate | 8.82 | 9.59 | 9.44 |
| ammonium | 4.71 | 4.95 | 5.69 |
| sulfate | 5.81 | 5.84 | 7.74 |

Regarding whether these factors (the emission for $SO_2$ and $NO_x$ and the model's representation of inorganic aerosol formation) could be the primary cause of the large prior discrepancy, we conclude that while they introduce uncertainty, the systematic underestimation of $NH_3$ emissions is the principal driver. The prior simulation systematically underestimated the mean $NH_3$ column over Eastern China by 61%. Uncertainties in the emission inventories of other pollutants, such as $SO_2$ or NOx, or

biases in the chemical mechanisms, are insufficient to explain such a large and widespread systematic underestimation of $NH_3$ itself.

Furthermore, a key point of our study design is that the only variable adjusted between the prior and posterior simulations was the $NH_3$ emission inventory. This single adjustment led to significant improvements across multiple metrics. The model performance improved for concentrations of $NH_3$ and other relevant pollutants, such as $SO_2$, $NO_2$ and $PM_{2.5}$, indicating that $NH_3$ emissions were the core issue. In addition, previous studies have suggested that the bottom-up inventories frequently underestimated $NH_3$ emissions (Chen et al., 2021; Ding et al., 2024; Kong et al., 2019; Zhang et al., 2017). Overall, we would like to thank for your scientific suggestion and we have comprehensively revised the relevant content in the manuscript to make our statement and discussion clearer.

*Revision in Section 2.2:*

*Furthermore, speciated inorganic aerosol data from a representative site in Beijing were collected to evaluate the model's capacity in characterizing the formation of secondary inorganic aerosols (Tan et al., 2018).*

*Revision in Section 4.1:*

*Finally, the entire process is iteratively repeated, a framework that captures the overall non-linear atmospheric response by combining the dynamic simulation of non-linear chemistry within each WRF-Chem step with the collective behavior of multiple iterations.*

*Revision in Section 4.2:*

*Additionally, uncertainties from the model's chemical mechanisms and the influence of nearby grid transport also contribute to this gap, but the overall impact on the final estimate is limited.*

*Revision in Section 4.3:*

*To further characterize the model's chemical performance beyond total $PM_{2.5}$, we also evaluated the simulation of secondary inorganic aerosol (SIA) components against in-situ measurements from a representative site in Beijing (Table S7). The evaluation shows that the posterior $NH_3$ emissions improved the simulation of ammonium and nitrate, reducing the bias between simulated and observed concentrations. Although the model underestimates sulfate, likely due to missing formation mechanisms (Cai et*

*al., 2024; Wang et al., 2021, 2020), the total SIA concentration is well reproduced with an overall bias of only -11.0%.*

*Revision in Supplementary:*

*Table S7. Comparison between the prior and posterior simulated inorganic aerosol concentrations with in-situ measurements in Beijing. All value units are $\mu g \, m^{-3}$.*

|  | Prior simulation | Posterior simulation | observation |
|---|---|---|---|
| nitrate | 8.82 | 9.59 | 9.44 |
| ammonium | 4.71 | 4.95 | 5.69 |
| sulfate | 5.81 | 5.84 | 7.74 |

2. It is not very clear to me how the current MLR framework can separate AGR and non-AGR $NH_3$ emissions. Some clarification would be helpful, as noted in the minor comments below. Generally, in each grid cell j, I would expect the temporal variations in $SA_{agriculture}$ and $SA_{non-agriculture}$ to be perfectly correlated and differ only in magnitude, thus they could not be separated in the regression. While WRF-Chem may simulate different day-to-day variations for $SA_{agriculture}$ and $SA_{non-agriculture}$, that reflects the effects of transport from surrounding grid cells, which contradicts the assumption that transport effects are negligible. Also, it would be useful to explain why the emission corrections primarily affect the non-AGR sector. Given that non-AGR emissions are relatively small in the prior, one would expect $SA_{non-agriculture}$ to be much smaller than $SA_{agriculture}$ in Eqn (4). Is the regression coefficient b significantly larger than a, and if so, what is the reason for that? Specifically, in northern Henan, the posterior results show decreased AGR but increased non-AGR emissions compared to the prior, which seems hard to understand and needs further explanation.

**Response:**

We thank you for this insightful question regarding the separation of agricultural (AGR) and non-AGR sources within our MLR framework.

The effective separation of these two sectors in our study is primarily based on their distinct and inconsistent spatial distributions. The use of multiple linear regression (MLR) for source apportionment is a well-established approach in atmospheric science (Qi et al., 2024; Shu and Lam, 2011; Trošić Lesar and Filipčić, 2023) and can identify different physical sources. The fundamental principle of using regression for source

apportionment is that different sources can be statistically distinguished if they possess unique spatial "fingerprints".

In our study, the high-concentration regions resulting from AGR and non-AGR emissions do not spatially align (Figure R2.3). The overall spatial correlation between the $NH_3$ columns simulated from these two sources is low (r = 0.35) and is near zero in the high-concentration regions (r = 0.03). This significant dissimilarity provides a robust statistical basis for the MLR model to distinguish their relative contributions.

The adjustment of these sources occurs through a multi-stage iterative process. Initially, the algorithm addresses the large, domain-wide underestimation by increasing emissions from both sectors. In subsequent iterations, a finer adjustment occurs where the framework optimizes the relative mix of the two sources to better match the observed spatial patterns. This directly relates to the emission corrections. If the spatial pattern of non-AGR emissions provides a better fit to the remaining model-observation discrepancy in certain areas, its corresponding emissions will be increased more significantly. This can result in a larger effective adjustment for the non-AGR sector, even if its initial contribution is smaller.

The specific case of northern Henan, where AGR emissions decrease while non-AGR emissions increase, exemplifies this refinement stage. In this region, the initial emission adjustments likely resulted in a spatial pattern that did not perfectly match the satellite observations. The algorithm then corrects this by reducing the AGR sector's contribution while simultaneously increasing the non-AGR sector's influence, as the latter's spatial pattern provided a better fit. The success of this adjustment is quantitatively demonstrated by the significant improvement in model performance for this region: the spatial correlation in Henan increased from 0.47–0.58 in the prior to 0.64–0.90 in the posterior. This confirms the framework is effectively adjusting the relative structure of emissions to best match the observations.

[Figure]

**Figure R2.3.** Spatial distribution of prior simulated $NH_3$ column concentrations from

agricultural and non-agricultural sources.

*Revision in Section 4.1:*

*In each iterative calculation, the monthly average satellite-derived NH₃ column concentration served as the target, and multiple linear regression (MLR) was applied to calculate the corresponding regression factors for AGR and non-AGR emissions (Figure S3). This separation of sectors by MLR is effective because their respective spatial distributions are distinct and largely uncorrelated (r = 0.35).*

*Revision in Section 4.2:*

*In multiple iterations, the framework optimizes the relative mix of the two sources to better match the observed spatial patterns. For instance, the spatial correlation between model and observation in Henan increased from 0.47–0.58 (prior simulations) to 0.64–0.90 (posterior simulations).*

3. The finding of substantially higher non-agricultural (non-AGR) NH₃ emissions compared to prior estimates is certainly interesting and important. However, the discussion of the posterior results in Section 4 currently focuses mainly on reporting emission magnitudes and sectoral contributions, with limited interpretation of the underlying causes or contextualization within existing literature. I would encourage the authors to expand this discussion by addressing the following points: (1) What are the potential reasons for the apparent underestimation of non-AGR NH₃ emissions in current bottom-up inventories? (2) What types of non-agricultural sources (e.g., industrial processes, transportation) are most likely responsible, based on current understanding? (3) How do your findings about non-AGR emissions compare with previous top-down estimates? The discussion in Lines 259–281 is helpful, but it could be further strengthened by emphasizing on observation-based or model-based studies that have investigated non-AGR NH₃ sources. It would also be valuable to highlight how your results build upon or differ from those studies, and what novel insights your analysis contributes to this topic.

**Response:**

(1) We thank the reviewer for the constructive suggestion The principle of bottom-up emission inventories is the product of activity level and emission factor. The accuracy of bottom-up inventories is highly dependent on the input data. However, incomplete activity levels, not representative emission factors, and the overlooked

sources make it a challenge to reasonably estimate non-AGR $NH_3$ emissions in China.

Several overlooked factors contribute to the underestimation of industrial $NH_3$ emissions in bottom-up inventories: (1) A significant source is "ammonia slip" from the widespread use of denitrification technologies (like SCR and SNCR) to control $NO_x$ emissions. (2) a crucial and largely overlooked source is the 'indirect emission' of $NH_3$, where ammonia is first adsorbed onto byproducts like fly ash and desulfurization slurry and is subsequently released during their handling and utilization (Chen et al., 2022; Cheng et al., 2020; Liu et al., 2020). (3) emission inventories have often omitted a range of downstream chemical industries that use ammonia as a feedstock (Wei et al., 2022). The inclusion of these factors results in an estimate of industrial $NH_3$ emissions that is 3–10 times higher than those in previous bottom-up inventories (Chen and Wang, 2025).

Similarly, for residential sources, volatilization from landfills, wastewater treatment, and human excreta are important sources, especially in densely populated megacities. For example, some studies have identified human excreta as a stable and significant contributor to the urban $NH_3$ budget, a sector frequently omitted in traditional inventories (Chang et al., 2015; Shao et al., 2020).

For the transportation sector, many inventories rely on emission factors developed for European or U.S. vehicle fleets. These factors often fail to capture complex real-world conditions in China, such as catalyst aging, vehicle maintenance status, and diverse driving patterns in congested traffic. This leads to an underestimation of the true $NH_3$ emission rate, a finding confirmed by several field studies. (Chang et al., 2016; Sun et al., 2017; Zhang et al., 2021).

Underestimation across these multiple sectors demonstrates the systematic underestimation of non-AGR $NH_3$ emissions in current bottom-up inventories. We have rephrased the relevant texts in the manuscript for further discussion.

(2) We thank the reviewer for this insightful comment. We are unable to further categorize the posterior non-AGR emissions into specific sub-sectors due to two primary methodological limitations: (1) Introducing additional sub-sectors as separate factors in the Multiple Linear Regression (MLR) model would cause the results for some of these factors to become statistically insignificant. (2) At the current resolution of our model (18 km) and the prior emission inventory (0.25°), the spatial distributions of many non-AGR sub-sectors are too similar, which prevents our current method from

distinguishing between them. However, your comment offers us an excellent opportunity to discuss the current understanding of the relative importance of different non-agricultural sources based on existing literature.

We have compiled source apportionment results from several emission inventories in Figure R2.4. The inventories covering China suggests that, on a national scale, industrial and residential sources are two major contributors to non-agricultural $NH_3$ emissions. When we extract data specifically for our study domain from established grided inventories like MEIC and CEDS, with residential and waste-related sources showing significant contributions. This is confirmed by high-resolution inventories for sub-regions such as YRD, where residential and waste sources are also identified as primary contributors.

[Figure]

**Figure R2.4.** Comparison of the source apportionment of non-AGR $NH_3$ emissions from different studies and for various regions. Each pie chart illustrates the relative contribution (%) of nine specific non-agricultural sources, with the area (study region, China, and the YRD) and data source for each chart indicated in its center.

(3) We thank the reviewer for this suggestion. The majority of previous top-down studies on $NH_3$ have focused on optimizing the total emission budget, without explicitly separating the contributions from different sectors (Xu et al., 2023). Some studies attempted to qualitatively estimate source-specific $NH_3$ emissions. For instance, Kong et al. (2019) and Liu et al. (2022) used satellite-observed $NH_3$ hotspots and linked them

to specific industrial or agricultural point sources with external information (Kong et al., 2019; Liu et al., 2022). Other approaches have attempted a form of quantitative allocation by first using top-down methods to constrain the total emission, and then relying on the sectoral fractions from bottom-up inventories for further apportioning.

A separate and distinct approach involves the use of stable isotope analysis. These studies have provided crucial quantitative insights, suggesting that the contribution of non-AGR sources to ambient $NH_3$ concentrations can be remarkably high, potentially up to ~90% in specific urban environments (Pan et al., 2016; Wu et al., 2020). However, it is important to note that this valuable technique typically provides constraints on source contributions to ambient concentrations rather than directly on emission fluxes.

Our work builds upon these previous findings by attempting to quantitatively disentangle the emissions from agricultural and non-agricultural sectors directly within our top-down framework. Instead of optimizing the total emissions and then allocating them post-hoc based on bottom-up information, our iterative MLR approach uses the distinct spatial signatures of the two sectors to derive separate adjustment factors for each. This provides a direct, observation-based constraint on the relative contributions of AGR and non-AGR emissions over a large region.

This approach addresses a potential methodological gap in top-down research, which has traditionally faced challenges in achieving direct, quantitative source attribution at regional scales. While acknowledging the uncertainties and limitations inherent in our study, we suggest this methodology could offer a valuable pathway toward more effective utilization of satellite observations for investigating source-specific emission trends.

*Revision in Section 4.2:*

*Analysis of emission inventories (An et al., 2021; Hoesly et al., 2018; Li et al., 2021, 2017; Ma, 2020; Wu et al., 2024) reveals that residential activities and waste disposal are dominant sources of non-AGR $NH_3$ emissions, particularly in densely populated regions (Figure S5).*

*The relatively small proportion of non-AGR emissions is likely due to overlooked industrial (e.g., $NH_3$ slip and indirect emissions) (Chen and Wang, 2025; Chen et al., 2022; Wei et al., 2022) and residential sources (e.g., from waste) (Shao et al., 2020), combined with unrepresentative transportation emission factors (Sun et al., 2017;*

*Zhang et al., 2021).*

*Our work attempts to quantitatively disentangle the emissions from AGR and non-AGR sectors directly within our top-down framework and facilitates a more comprehensive capture of neglected non-AGR sources.*

***Revision in Supplementary:***

[Figure]

***Figure S5.*** *Comparison of the source apportionment of non-AGR NH$_3$ emissions from different studies and for various regions. Each pie chart illustrates the relative contribution (%) of nine specific non-agricultural sources, with the area (study region, China, and the YRD) and data source for each chart indicated in its center.*

**Minor comments:**

1. Line 26/28 and elsewhere: please remove the "·" between Tg and yr$^{-1}$. Also, replace Gg mon$^{-1}$ with either Gg month$^{-1}$ or Gg mo$^{-1}$ to follow standard unit concentrations.

**Response:** Thank you for your careful reminder. We have checked that all expressions of emission units follow the standard format throughout the manuscript. Please refer to our revised manuscript.

2. Line 37-40. The summary statement is too general. It would be more informative to highlight the insights into non-agricultural ammonia emissions and their implications.

**Response:** Thank you for your comment. We have rephrased the relevant texts of the abstract to highlight the importance of identifying non-agricultural $NH_3$ emissions and their implications in reducing $PM_{2.5}$ pollution and health burden.

***Revision in Section Abstract:***

*Our study evaluated $NH_3$ emissions from various sources in Eastern China, emphasizing the impact of reducing non-agricultural ammonia emissions on air quality and public health benefits.*

3. Line 127: Is biomass burning emission also treated online? Just checking, as this is not commonly the case.

**Response:** Thank you for pointing this out. The biomass burning emissions were generated using the FINN v1.5, a model developed by NCAR. This model provided a useful utility for allocating the original wildfire emissions, which had a spatial resolution of 1km, to grid cells of the WRF-Chem model. Thus, we adopted an offline approach to prepare pollutant emissions from biomass burning. The description of this process in the previous manuscript contained inaccuracies. We have now revised the related section to reflect the correct methodology.

***Revision in Section 2.1:***

*Furthermore, biogenic emissions were calculated online using the Model of Emissions of Gases and Aerosols from Nature (MEGAN, version 2.0.4) (Guenther, 2006). Our numerical simulations also incorporated offline biomass burning emissions of various air pollutants, based on the wildfire model Fire Inventory from NCAR (FINN, version 1.5) (Wiedinmyer et al., 2011).*

4. Line 134: The "last accessed" date should reflect the actual date when the data were downloaded.

**Response:** Thank you for your kind reminder. The original IASI satellite products were actually downloaded in December of 2020. As you suggested, we have modified the statement regarding this data access date.

*Revision in Section 2.2:*

*We obtained the total column density of $NH_3$ from the passive satellite remote-sensing product of the Infrared Atmospheric Sounding Interferometer (IASI) (version 3.0, https://iasi.aeris-data.fr/nh3/, last accessed on December 2020) as the observational constraint.*

5. In section 2.2, what's the overpass time of IASI data? Do you use level-2 satellite data?

**Response:** Thank you for this comment. IASI is a passive remote-sensing instrument that was first launched in 2006 on board the MetOp-A meteorological satellite, which circles the Earth in a polar Sun-synchronous orbit. It crosses the equator at mean local solar times of 9:30 and 21:30 (Van Damme et al., 2014), which are also the overpass times. In the present study, we used a processed satellite product at level 2 to access $NH_3$ column concentrations. We have revised the related text in Section 2.2 to better clarify the introduction of IASI data.

*Revision in Section 2.2:*

*The IASI is a Fourier transform spectrometer on board the Metop series of meteorological satellites, which circle the Earth in a polar Sun-synchronous orbit (Van Damme et al., 2014). Consequently, the satellite-based IASI instrument can cover the entire globe and provide measurements twice a day at 09:30 and 21:30 local solar time. The IASI instrument detects infrared radiation in the spectral range from 645 to 2760 $cm^{-1}$ emitted by Earth's surface and atmosphere with a 12 km circular footprint at nadir. This radiation absorption range includes the $NH_3$ signal near 950 $cm^{-1}$. The collected daily $NH_3$ column concentrations are categorized into level-2 satellite data and are developed based on the ANNI-NH$_3$ inversion algorithm without averaging kernels, as presented by Van Damme et al. (2017).*

6. Line 140-141: The description is unclear. How is the neural network applied to improve the data quality, was it developed by the authors or sourced elsewhere?

**Response:** Thank you for pointing this out. Van Damme et al. (2017) used the artificial neural network technique presented by Whitburn et al. (2016) to improve the quality of IASI satellite data. This work was carried out by other teams. Specifically, they trained separate neural networks for land and sea observations, resulting in a better training performance for both. To state this point clearer, we have rephrased the description of the IASI data product as follows.

*Revision in Section 2.2:*

*The daily NH₃ column concentrations are categorized into level-2 satellite data and are developed based on the ANNI-NH3-v2.1 inversion algorithm without averaging kernels, as presented by Van Damme et al. (2017). Specifically, their retrieval algorithm derives hyperspectral radiation indexes (HRI) from the direct satellite spectrum detection, which is then converted into final NH₃ column concentrations using an artificial neural network technique (Whitburn et al., 2016). For better data quality, the present study removed NH₃ column concentrations associated with cloud cover of more than 10%. Furthermore, we preprocessed the IASI NH₃ column concentration data through averaging all daily values to obtain a monthly mean. Spatially, we mapped the original satellite product data to the grid cells of the WRF-Chem model for further comparison with those simulated NH₃ columns.*

7. Line 125/156: Which version of MEIC is used?

**Response:** Thank you for your comment. Our study used version 1.3 of the MEIC anthropogenic emission inventory. We have included the version number in the revised manuscript as you suggested.

*Revision in Section 2.1:*

*We adopted the anthropogenic emissions from the Multi-resolution Emission Inventory for China (MEIC, version 1.3) developed by Tsinghua University (Li et al., 2017; Zheng et al., 2018).*

8. Line 162-163 and Section 4.1: Please clarify how the model and observations are sampled for comparison.

**Response:**

Thanks for your conducive comment. We conducted a model evaluation of $NH_3$ concentrations, comparing observations and simulations based on prior emissions, from two perspectives.

The first aspect is the total column concentration of $NH_3$. We calculated the respective simulated $NH_3$ column concentrations within 27 vertical layers. The overall $NH_3$ column concentration can then be inferred by summing up all the partial column concentrations. The IASI satellite data has been pre-allocated to the grid cells of the WRF-Chem model. We then sampled the monthly average total column concentration of $NH_3$ at the same grid cell, and carried out comparisons between IASI observations and WRF-Chem simulations.

Another validating parameter is the surface $NH_3$ measurement. In this circumstance, we only extracted the simulated $NH_3$ volume concentrations in the first layer near the ground surface. $NH_3$ measurements were collected from previous studies and are presented as annual averages (Table S2). Model simulations of $NH_3$ volume concentrations were sampled at certain grid cells according to the longitudes and latitudes of 12 different measurement sites. The final comparison was made in terms of the annual mean $NH_3$ volume concentration at these sites, between model simulation and in situ observations. To clarify this point, we have rephrased the relevant text in the revised manuscript.

*Revision in Section 3:*

*We compared the prior model results with IASI $NH_3$ column concentration and surface $NH_3$ volume concentration observations. The detailed method for calculating $NH_3$ total column concentrations and surface volume concentrations from WRF-Chem is provided in Text S1.*

*Revision in Supplementary:*

*For surface $NH_3$ volume concentrations, we extracted the corresponding simulations at 12 sites summarized from previous in situ measurement studies and conducted a comparison between the model simulations and the measurements in terms of the annual mean $NH_3$ volume concentrations.*

9. Line 179: The term "errors" is vague. Consider using clearer language such as "underestimated by 30%" or "biased low by 30%."

**Response:** Thank you for your comment. Following your suggestion, we have rephrased the relevant text in Section 3 to make this point clearer.

*Revision in Section 3:*

*Most simulated NH₃ total column concentrations are underestimated by more than 30% compared with the observed values by satellite with the associated RMSE exceeding $10 \times 10^{15}$ molec cm⁻².*

10. Line 181-189: Figure 6 can be described with the text here.

**Response:** Thank you for your suggestion. We indeed agree with you that Figure 6 should be introduced here to better depict the spatial distribution pattern of the observed and the simulated NH₃ column concentrations. We have revised the placement of figures as you suggested.

*Revision in Section 3:*

*As illustrated in Figure 6, satellite-based observations reveal that the spatial high-value areas of NH₃ column are located at the junction of Henan, Shandong, and Hebei provinces. In contrast, the prior modeling results show that NH₃ column densities are more concentrated in Henan. This indicates a clear discrepancy in the spatial distribution of NH₃ column densities between the prior simulations and the observations.*

11. Section 3: Please be consistent in the use of statistical metrics. RMSE is used for IASI comparisons, while IOA and MFB are used for surface observations. A brief explanation of why different metrics are applied, and what each evaluates, would be helpful.

**Response:** Thank you for pointing this out. In fact, we calculated all three evaluation metrics for comparisons between model versus measurements both in the total column concentration and surface volume concentration of NH₃. However, not all metrics are presented in the original manuscript. As you suggested, we have revised the relevant text in Section 3 by discussing all evaluation metrics to better elucidate the model performance.

*Revision in Section 3:*

*As shown in Table S5, the annual average of NH₃ total column concentrations is simulated to be $17.4 \times 10^{15}$ molec cm⁻² for Eastern China, with a 61% underestimation of MFB compared to the observations from IASI satellite retrievals ($29.0 \times 10^{15}$ molec*

*cm⁻²).* The IOA between observations versus simulations is 0.72. The seasonal

*cm⁻²). The IOA between observations versus simulations is 0.72. The seasonal simulations of NH₃ concentrations also exhibit significant discrepancies with observations, especially in spring. Specifically, the simulated NH₃ total column concentration in Eastern China is only $13.2\times10^{15}$ molec cm⁻² in spring, with concentration in 67.5% of the study region being underestimated by more than 50%. These discrepancies are evidently exhibited in Figure 3. Most simulated NH₃ total column concentrations are underestimated by more than 30% compared with the observed values by satellite with the associated RMSE exceeding $10\times10^{15}$ molec cm⁻².*

*Additionally, the comparison between the simulated and observed surface NH₃ volume concentrations also indicates a notable underestimation (Figure S2). The mean simulated surface NH₃ volume concentration over the study region is 6.3 μg m⁻³, which is only half of the observation value (12.7 μg m⁻³), with an IOA of 0.57 and an MFB of -61%, respectively (Table S5).*

***Revision in Supplementary:***

*Table S5. List of the evaluation metrics of NH₃ concentrations.*

| | The total column concentration $(10^{15}$ molc cm⁻²$)$ | | Surface volume concentration $(μg$ m⁻³$)$ | |
|---|---|---|---|---|
| | *Prior* | *Posterior* | *Prior* | *Posterior* |
| *Mean obs.* | *29.0* | | *12.7* | |
| *Mean model.* | *17.4* | *23.7* | *6.3* | *9.4* |
| *IOA* | *0.72* | *0.91* | *0.57* | *0.65* |
| *MFB* | *-0.61* | *-0.30* | *-0.61* | *-0.19* |
| *RMSE* | *13.9* | *7.9* | *9.1* | *7.3* |

12. Line 195: You mention deriving posterior emissions for four months—how are prior/posterior simulations compared with observations across the seasons? Are the same scale factors applied to all three months in each season? Please clarify. Given that WRF-Chem simulations are available for the full year, it would be more consistent to derive monthly emissions for all 12 months, which should follow the same procedure and would not require much additional effort.

**Response:**

Thank you for this question, which allows us to clarify the details and rationale of our experimental design. To clarify, our prior and posterior simulations were indeed

conducted for the full 12 months of 2016. Seasonal comparisons with satellite observations were made using seasonal averages of both simulated and observed data. However, the emission adjustments were derived exclusively from four representative months (January, April, July, and October). For each season, we first calculated a posterior emission inventory for its representative month using the corresponding adjustment factors, then applied this result uniformly to all three months within that season.

Our primary reason for adopting the representative-month approach was to enable a robust independent validation of our results, which is a common practice in computationally intensive modeling studies (Qu et al., 2017; Xia et al., 2025; Xu et al., 2021). By constraining our emissions using only four months, the remaining eight months serve as an independent dataset against which we can evaluate the performance of our posterior inventory. The good performance of our posterior simulation (including 'non-training' months) provides strong evidence that the adjustments are not over-fitted to specific monthly conditions and that the resulting posterior emission inventory is effective for the entire season.

Second, the use of representative months is a reasonable approach for characterizing seasonal patterns. The representative months effectively capture the overall seasonal cycle, with the highest concentrations in summer and the lowest in winter. Furthermore, the $NH_3$ column concentration of each representative month is in good agreement with its corresponding three-month seasonal average, with the relative difference ranging from only 1.9% to 17.3%. This small discrepancy confirms that our method reliably represents the seasonal average.

Furthermore, conducting year-round simulations for emission adjustments would incur substantial computational costs, which is a critical practical constraint. Our study conducted more than 20 regression iterations to optimize emissions. Extending this process to 12 months would demand additional computational resources of ~14 model-years, representing an extremely resource-intensive undertaking for regional chemical modeling.

We have revised the methodology section to provide this comprehensive explanation.

*Revision in Section 4.1:*

*The posterior emission inventory derived for each representative month was then*

*applied to all three months within its corresponding season to generate the full 12-month posterior inventory. This representative-month approach was adopted to allow for a robust validation against the full 12-month period, with the remaining eight months serving as an independent dataset, and to manage the substantial computational cost of the iterative process.*

13. Line 200: In Line 138, you mentioned that IASI data were regridded to the model resolution, but here you refer to single-pixel comparisons, which is somewhat confusing. Please clarify how the satellite data were matched to model outputs.

**Response:** We greatly appreciate your careful comment. We agree that the term "single-pixel" is not an appropriate statement. In the present study, we preprocessed the level-2 IASI satellite data via mapping them to the grid cells of the WRF-Chem model for further comparison with those simulated $NH_3$ columns. We have revised this confusing term to make the text clearer. Please refer to our revisions as follows.

*Revision in Section 4.1:*

*In each iterative calculation, the monthly average satellite-derived $NH_3$ column concentration served as the target, and multiple linear regression (MLR) was applied to calculate the corresponding regression factors for AGR and non-AGR emissions (Figure S3).*

14. Line 202: What does "regression factor" refer to? Is it the same as the emission scale factor?

**Response:** Thank you for the question, which helps us to clarify our terminology. In our manuscript, the term "regression factor" refers to the coefficients (α and β) derived from the MLR analysis. These factors effectively function as scaling factors for the prior emissions. The purpose of this process is to adjust the magnitude of the emissions from each sector so that the resulting simulated $NH_3$ column concentrations better match the satellite observations. We have revised the text to use this terminology more consistently and have clarified the definitions to improve clarity.

*Revision in Section 4.1:*

*Furthermore, the MLR approach provided regression coefficients $\alpha_i^{j,k}$ and $\beta_i^{j,k}$, which function as scaling factors, respectively correspond to AGR and non-AGR $NH_3$ emissions in month j from region k, within the i iteration.*

15. Line 206: Is $TA_{satellite}$ the monthly average or the daily average of $NH_3$

concentrations?

**Response:** Thank you for pointing this out. The symbol $TA_{satellite}$ shown here indicates the monthly average of $NH_3$ column concentrations from IASI satellite products. To make this point clearer, we have revised the texts in Sections 2.2 and Section 4.1 as follows.

*Revision in Section 2.2:*

*Furthermore, we preprocessed the IASI $NH_3$ column concentration data through averaging all daily values to obtain a monthly mean value.*

*Revision in Section 4.1:*

*where, $TA_{satellite}{}^{j,k}$ denotes the monthly average of total $NH_3$ column density retrieved from the IASI satellite data, and $SA_{transport}{}^{j,k}$, $SA_{agriculture}{}_{i-1}^{j,k}$ and $SA_{non-agriculture}{}_{i-1}^{j,k}$ stand for the simulated total column concentration of $NH_3$ contributed by AGR emissions, non-AGR emissions, and outside transportation, respectively.*

16. Line 208: Should be "outside transportation, AGR emissions, non-AGR emissions, respectively".

**Response:** Thank you for the comment and the careful reminder. It is indeed that the order is reversed. We have revised this sentence in a correct order.

*Revision in Section 4.1:*

*$SA_{transport}{}^{j,k}$, $SA_{agriculture}{}_{i-1}^{j,k}$ and $SA_{non-agriculture}{}_{i-1}^{j,k}$ stand for the simulated total column concentration of $NH_3$ contributed by outside transport, AGR emissions, and non-AGR emissions, respectively.*

17. Line 211: The term "control emissions" is unclear. Do you mean emissions were zeroed out? Also, please replace "cycle" with "experiment."

**Response:** Thank you for your suggestions. We acknowledge that the term "control emissions" here is of confusing. We would like to express the meaning of forcing the corresponding $NH_3$ emissions to zero as you mentioned. Meanwhile, we also have replaced the "cycle" with "experiment". Please refer to our revisions below.

*Revision in Section 4.1:*

*In each experiment, we zeroed out AGR emissions, non-AGR emissions and regional external emissions to obtain the corresponding $NH_3$ column concentrations.*

**Response:**

Thank you for this question. The $A_{blank}$ case refers to a simulated $NH_3$ total column in which all anthropogenic emissions within the study domain were turned off. The purpose of this simulation was to establish a blank line concentration field, which represents the influence of the chemical boundary conditions provided to our model domain.

As described in our methodology, the $NH_3$ column concentration resulting from the $A_{blank}$ run is then subtracted from the other sensitivity simulations (e.g., $A_{agr}$, $A_{non-agr}$) to isolate the specific contribution of each in-domain emission sector. While the magnitude of this blank line concentration is very small compared to the contributions from emissions within the domain, it is an essential step to ensure accurate source attribution. We have clarified the role and definition of the $A_{blank}$ simulation in the revised manuscript.

*Revision in Section 4.1:*

*Specifically, the modeling case $A_{blank}$ refers to a simulated $NH_3$ total column in which all anthropogenic emissions within the study domain were zeroed out. The purpose of this simulation was to establish background concentrations, which represents the influence of the chemical boundary conditions provided to our model domain.*

19. Line 203/216: Earlier you use k for month and j for region, but later this is reversed. Please ensure consistency throughout. Also, using "grid cell j" is clearer than "region j" or "area j."

**Response:** We greatly appreciate your careful review. The typographical error has been corrected in the manuscript. Given that our calculations and analysis are region-based, we feel that 'region j' is a more descriptive term for our methodology. We have also revised the text to ensure this term is used consistently.

*Revision in Section 4.1:*

*Furthermore, the MLR approach provided regression coefficients $\alpha_i^{j,k}$ and $\beta_i^{j,k}$, which function as scaling factors, respectively correspond to AGR and non-AGR $NH_3$ emissions in month j from region k, within the i iteration.*

20. Line 217: It is unclear why does the regression is derived mathematically imply it

needs to be corrected? Please clarify the motivation for adjusting the regression coefficients.

**Response:**

We thank the reviewer for this question, which addresses an important detail of our methodology.

While a standard MLR provides a mathematical best fit, this fit may not always be statistically robust, particularly when be influenced by outliers. Our motivation for the correction procedure is to ensure that only statistically significant and reliable regression results are used to update the emissions. This prevents introducing noise from unreliable fits into the iterative process, which could lead to unstable or non-physical solutions.

We implemented a quality control procedure based on the statistical significance of the prediction error. For each regression, we calculate the residual (the difference between the observed and predicted values). If the 95% confidence interval of this residual does not contain zero, the model's prediction error is considered statistically significant, and the regression result is flagged as unreliable. As illustrated in the figure provided in our response, outliers (in red) are identified using this method.

We have revised the methodology section of the manuscript to explicitly state this motivation and to provide a clear description of our quality control procedure.

[Figure]

Figure R2.5:Distribution of residuals and their 95% confidence intervals. Each point represents the residual value for a given sample, and the error bars represent the 95% confidence interval of the residual. Green points represent valid fits, while red points are outliers rejected based on the criterion that their confidence interval does not contain zero.

*Revision in Section 4.1:*

*To ensure the statistical robustness of the regression equation, we need to correct for this regression coefficient.*

*The regression coefficients with excessive residuals, defined as cases where the 95% confidence interval of the residual does not contain zero, are removed to increase credibility.*

21. Line 218-231: The description of the correction process is not very clear. It is unclear what is meant by "goodness of fit," how the "invalid" regression coefficients are defined, and what fraction of them are removed. The phrase "make a trade-off" in Line 225 is vague and would benefit from clarification. Additionally, it is not explained how the adjustment factors $a_n$ and $b_n$ are derived or what their physical meaning is. The choice of a 30% threshold in Line 229 also seems arbitrary—particularly in high-$NH_3$ regions, where it could allow larger discrepancies between observations and simulations, but the physical basis for this threshold is not clearly explained.

**Response:**

The reviewer's meticulous feedback on our methodology is greatly appreciated, as it allows us to provide important clarifications. We have revised the manuscript to address these points in detail.

First, regarding the "goodness of fit" metric, we used the coefficient of determination (R-squared, $R^2$). A regression result was deemed "invalid" if the $R^2$ was less than 0.3 or if the 95% confidence interval of its residual did not contain zero. This is our quality control criterion for identifying and rejecting statistically poor fits. On average, 6.3% to 9.4% of regression results were rejected per iteration.

The phrase "make a trade-off" was used to describe our procedure for handling these invalid results. To clarify, if a regression result for a given grid cell is deemed

valid, the new adjustment factors (a and b) are set to the newly calculated regression coefficients (α and β). If the result is invalid, the adjustment factors are kept unchanged from the previous iteration ($a_i = a_{i-1}$).

This conservative approach ensures that emissions are only updated based on statistically robust fits, rather than deleting any data. The physical meaning of these adjustment factors, a and b, is that they represent the scaling multipliers applied to the prior emissions of the AGR and non-AGR sectors, respectively, to better match the satellite observations in each iteration.

Our choice of the 30% threshold was based on two primary considerations: (1) Within the widely accepted 20%–50% error range for model performance benchmarks, (EPA, 2007; Huang et al., 2021, 2025; Zhao et al., 2017), we selected 30% as our criterion to account for the inherent uncertainties in both the WRF-Chem model and the IASI satellite data. (2) Compared to the prior bias of up to -61%, reducing this bias to within 30% represents a significant and meaningful improvement, which proves that our method has imposed an effective constraint on the emission inventory.

*Revision in Section 4.1:*

*Concurrently, the goodness of fit of the regression is calculated as the coefficient of determination (R-square, $R^2$). To maintain algorithm stability, regressions with an $R^2$ less than 0.3 are deemed invalid and excluded from the emission update, as they exhibit insufficient explanatory power (indicating >70% unexplained variance) and introduce destabilizing noise into the adjustments. We further use it to make a trade-off for the regression coefficient. If a regression is valid, the adjustment factors a and b are set to the new regression coefficients; if invalid, the factors are kept unchanged from the previous iteration. The updated emissions for the next iteration are then calculated by multiplying the emissions from the previous step by these adjustment factors.*

*The iteration concludes when the mean bias between the simulated values and observations is less than 30%, a criterion chosen to represent a significant improvement over the large prior bias while falling within the range of widely accepted model performance benchmarks.*

**References**

Cai, S., Liu, T., Huang, X., Song, Y., Wang, T., Sun, Z., Gao, J., and Ding, A.: Important Role of Low Cloud and Fog in Sulfate Aerosol Formation During Winter Haze Over the North China Plain, Geophysical Research Letters, 51, e2023GL106597, https://doi.org/10.1029/2023GL106597, 2024.

Chang, Y., Deng, C., Dore, A. J., and Zhuang, G.: Human Excreta as a Stable and Important Source of Atmospheric Ammonia in the Megacity of Shanghai, PLoS One, 10, e0144661, https://doi.org/10.1371/journal.pone.0144661, 2015.

Chang, Y., Zou, Z., Deng, C., Huang, K., Collett, J. L., Lin, J., and Zhuang, G.: The importance of vehicle emissions as a source of atmospheric ammonia in the megacity of Shanghai, Atmospheric Chemistry and Physics, 16, 3577–3594, https://doi.org/10.5194/acp-16-3577-2016, 2016.

Chen, P. and Wang, Q.: Underestimated industrial ammonia emission in China uncovered by material flow analysis, Environmental Pollution, 368, 125740, https://doi.org/10.1016/j.envpol.2025.125740, 2025.

Chen, Y., Shen, H., Kaiser, J., Hu, Y., Capps, S. L., Zhao, S., Hakami, A., Shih, J.-S., Pavur, G. K., Turner, M. D., Henze, D. K., Resler, J., Nenes, A., Napelenok, S. L., Bash, J. O., Fahey, K. M., Carmichael, G. R., Chai, T., Clarisse, L., Coheur, P.-F., Van Damme, M., and Russell, A. G.: High-resolution hybrid inversion of IASI ammonia columns to constrain US ammonia emissions using the CMAQ adjoint model, Atmospheric Chemistry and Physics, 21, 2067–2082, https://doi.org/10.5194/acp-21-2067-2021, 2021.

Chen, Y., Zhang, Q., Cai, X., Zhang, H., Lin, H., Zheng, C., Guo, Z., Hu, S., Chen, L., Tao, S., Liu, M., and Wang, X.: Rapid Increase in China's Industrial Ammonia Emissions: Evidence from Unit-Based Mapping, Environ Sci Technol, https://doi.org/10.1021/acs.est.1c08369, 2022.

Cheng, T., Zhou, X., Yang, L., Wu, H., and Fan, H.: Transformation and removal of ammonium sulfate aerosols and ammonia slip from selective catalytic reduction in wet flue gas desulfurization system, J Environ Sci (China), 88, 72–80, https://doi.org/10.1016/j.jes.2019.08.002, 2020.

Ding, J., Van Der A, R., Eskes, H., Dammers, E., Shephard, M., Wichink Kruit, R., Guevara, M., and Tarrason, L.: Ammonia emission estimates using CrIS satellite observations over Europe, Atmos. Chem. Phys., 24, 10583–10599, https://doi.org/10.5194/acp-24-10583-2024, 2024.

EPA: Guidance on the Use of Models and Other Analyses in Attainment Demonstrations for the 8-hour Ozone NAAQS, 2007.

Huang, L., Zhu, Y., Zhai, H., Xue, S., Zhu, T., Shao, Y., Liu, Z., Emery, C., Yarwood,

G., Wang, Y., Fu, J., Zhang, K., and Li, L.: Recommendations on benchmarks for numerical air quality model applications in China – Part 1: PM$_{2.5}$ and chemical species, Atmos. Chem. Phys., 21, 2725–2743, https://doi.org/10.5194/acp-21-2725-2021, 2021.

Huang, L., Zhang, X., Emery, C., Mu, Q., Yarwood, G., Zhai, H., Sun, Z., Xue, S., Wang, Y., Fu, J. S., and Li, L.: Recommendations on benchmarks for numerical air quality model applications in China – Part 2: Ozone and uncertainty analysis, Atmos. Chem. Phys., 25, 4233–4249, https://doi.org/10.5194/acp-25-4233-2025, 2025.

Kong, L., Tang, X., Zhu, J., Wang, Z., Pan, Y., Wu, H., Wu, L., Wu, Q., He, Y., Tian, S., Xie, Y., Liu, Z., Sui, W., Han, L., and Carmichael, G.: Improved Inversion of Monthly Ammonia Emissions in China Based on the Chinese Ammonia Monitoring Network and Ensemble Kalman Filter, Environ Sci Technol, 53, 12529–12538, https://doi.org/10.1021/acs.est.9b02701, 2019.

Liu, P., Ding, J., Ji, Y., Xu, H., Liu, S., Xiao, B., Jin, H., Zhong, X., Guo, Z., Wang, H., and Liu, L.: Satellite Support to Estimate Livestock Ammonia Emissions: A Case Study in Hebei, China, Atmosphere, 13, 1552, https://doi.org/10.3390/atmos13101552, 2022.

Liu, W., Wu, B., Bai, X., Liu, S., Liu, X., Hao, Y., Liang, W., Lin, S., Liu, H., Luo, L., Zhao, S., Zhu, C., Hao, J., and Tian, H.: Migration and Emission Characteristics of Ammonia/Ammonium through Flue Gas Cleaning Devices in Coal-Fired Power Plants of China, Environ. Sci. Technol., 54, 390–399, https://doi.org/10.1021/acs.est.9b04995, 2020.

Pan, Y., Tian, S., Liu, D., Fang, Y., Zhu, X., Zhang, Q., Zheng, B., Michalski, G., and Wang, Y.: Fossil Fuel Combustion-Related Emissions Dominate Atmospheric Ammonia Sources during Severe Haze Episodes: Evidence from (15)N-Stable Isotope in Size-Resolved Aerosol Ammonium, Environ Sci Technol, 50, 8049–56, https://doi.org/10.1021/acs.est.6b00634, 2016.

Qu, Z., Henze, D. K., Capps, S. L., Wang, Y., Xu, X., Wang, J., and Keller, M.: Monthly top-down NO$_x$ emissions for China (2005–2012): A hybrid inversion method and trend analysis, JGR Atmospheres, 122, 4600–4625, https://doi.org/10.1002/2016JD025852, 2017.

Shao, S.-C., Zhang, Y.-L., Chang, Y.-H., Cao, F., Lin, Y.-C., Mozaffar, A., and Hong, Y.-H.: Online characterization of a large but overlooked human excreta source of ammonia in China's urban atmosphere, Atmospheric Environment, 230, 117459, https://doi.org/10.1016/j.atmosenv.2020.117459, 2020.

Sun, K., Tao, L., Miller, D. J., Pan, D., Golston, L. M., Zondlo, M. A., Griffin, R. J., Wallace, H. W., Leong, Y. J., Yang, M. M., Zhang, Y., Mauzerall, D. L., and Zhu, T.: Vehicle Emissions as an Important Urban Ammonia Source in the United States

and China, Environ Sci Technol, 51, 2472–2481, https://doi.org/10.1021/acs.est.6b02805, 2017.

Tan, T., Hu, M., Li, M., Guo, Q., Wu, Y., Fang, X., Gu, F., Wang, Y., and Wu, Z.: New insight into PM2.5 pollution patterns in Beijing based on one-year measurement of chemical compositions, Science of The Total Environment, 621, 734–743, https://doi.org/10.1016/j.scitotenv.2017.11.208, 2018.

Wang, W., Liu, M., Wang, T., Song, Y., Zhou, L., Cao, J., Hu, J., Tang, G., Chen, Z., Li, Z., Xu, Z., Peng, C., Lian, C., Chen, Y., Pan, Y., Zhang, Y., Sun, Y., Li, W., Zhu, T., Tian, H., and Ge, M.: Sulfate formation is dominated by manganese-catalyzed oxidation of SO2 on aerosol surfaces during haze events, Nat Commun, 12, https://doi.org/10.1038/s41467-021-22091-6, 2021.

Wang, X., Gemayel, R., Hayeck, N., Perrier, S., Charbonnel, N., Xu, C., Chen, H., Zhu, C., Zhang, L., Wang, L., Nizkorodov, S. A., Wang, X., Wang, Z., Wang, T., Mellouki, A., Riva, M., Chen, J., and George, C.: Atmospheric Photosensitization: A New Pathway for Sulfate Formation, Environ. Sci. Technol., 54, 3114–3120, https://doi.org/10.1021/acs.est.9b06347, 2020.

Wei, L., Zhang, H., Sun, C., and Yan, F.: Simultaneous estimation of ammonia injection rate and state of diesel urea-SCR system based on high gain observer, ISA Transactions, 126, 679–690, https://doi.org/10.1016/j.isatra.2021.08.002, 2022.

Wu, C., Wang, G., Li, J., Li, J., Cao, C., Ge, S., Xie, Y., Chen, J., Liu, S., Du, W., Zhao, Z., and Cao, F.: Non-agricultural sources dominate the atmospheric NH3 in Xi'an, a megacity in the semi-arid region of China, Sci Total Environ, 722, 137756, https://doi.org/10.1016/j.scitotenv.2020.137756, 2020.

Xia, J., Zhou, Y., Fang, L., Qi, Y., Li, D., Liao, H., and Jin, J.: South Asia ammonia emission inversion through assimilating IASI observations, https://doi.org/10.5194/egusphere-2024-3938, 20 January 2025.

Xu, J., Lu, M., Guo, Y., Zhang, L., Chen, Y., Liu, Z., Zhou, M., Lin, W., Pu, W., Ma, Z., Song, Y., Pan, Y., Liu, L., and Ji, D.: Summertime Urban Ammonia Emissions May Be Substantially Underestimated in Beijing, China, Environ. Sci. Technol., 57, 13124–13135, https://doi.org/10.1021/acs.est.3c05266, 2023.

Xu, M., Jin, J., Wang, G., Segers, A., Deng, T., and Lin, H. X.: Machine learning based bias correction for numerical chemical transport models, Atmospheric Environment, 248, 118022, https://doi.org/10.1016/j.atmosenv.2020.118022, 2021.

Zhang, Q., Wei, N., Zou, C., and Mao, H.: Evaluating the ammonia emission from in-use vehicles using on-road remote sensing test, Environmental Pollution, 271, 116384, https://doi.org/10.1016/j.envpol.2020.116384, 2021.

Zhang, X., Wu, Y., Liu, X., Reis, S., Jin, J., Dragosits, U., Van Damme, M., Clarisse, L., Whitburn, S., Coheur, P. F., and Gu, B.: Ammonia Emissions May Be Substantially Underestimated in China, Environ Sci Technol, 51, 12089–12096, https://doi.org/10.1021/acs.est.7b02171, 2017.

Zhao, Y., Zhou, Y., Qiu, L., and Zhang, J.: Quantifying the uncertainties of China's emission inventory for industrial sources: From national to provincial and city scales, Atmospheric Environment, 165, 207–221, https://doi.org/10.1016/j.atmosenv.2017.06.045, 2017.

---

## Author Comment (AC3)

**Responses to reviewer #3**

Dear Editor and Reviewer #3:

We would like to thank the Editor and the reviewers for their time and valuable suggestions on our manuscript, "Optimizing Ammonia Emissions for PM$_{2.5}$ Mitigation: Environmental and Health Co-Benefits in Eastern China" (egusphere-2025-1407). We have carefully addressed all the points raised, and our point-to-point responses are detailed below. Reviewer comments are shown in blue. Our responses are shown in black. The revised texts are shown in red.

**Major Comments**

**1. Limited Optimization Period:**

The top-down optimization was performed only for four months (each season). Please justify why the analysis was limited to these periods and discuss whether this may affect annual emission estimates or bias seasonal interpretations.

**Response:**

Thank you for your valuable comment. Our primary reason for adopting the representative-month approach was to enable a robust independent validation of our results, which is a common practice in computationally intensive modeling studies (Qu et al., 2017; Xia et al., 2025; Xu et al., 2021). By constraining our emissions using only four months, the remaining eight months serve as an independent dataset against which we can evaluate the performance of our posterior inventory. The good performance of our posterior simulation (including 'non-training' months) provides strong evidence that the adjustments are not over-fitted to specific monthly conditions and that the resulting posterior emission inventory is effective for the entire season.

Second, the use of representative months is a reasonable approach for characterizing seasonal patterns. The representative months effectively capture the overall seasonal cycle, with the highest concentrations in summer and the lowest in winter. Furthermore, the NH$_3$ column concentration of each representative month is in good agreement with its corresponding three-month seasonal average, with the relative difference ranging from only 1.9% to 17.3%. This small discrepancy confirms that our method reliably represents the seasonal average.

Furthermore, conducting year-round simulations for emission adjustments would incur substantial computational costs, which is a critical practical constraint. Our study

conducted more than 20 regression iterations to optimize emissions. Extending this process to 12 months would demand additional computational resources of ~14 model-years, representing an extremely resource-intensive undertaking for regional chemical modeling.

We have revised the methodology section to provide this comprehensive explanation.

*Revision in Section 4.1:*

*The posterior emission inventory derived for each representative month was then applied to all three months within its corresponding season to generate the full 12-month posterior inventory. This representative-month approach was adopted to allow for a robust validation against the full 12-month period, with the remaining eight months serving as an independent dataset, and to manage the substantial computational cost of the iterative process.*

2. **Validation with Surface Data:**

While Figure S2 attempts to demonstrate agreement between model and surface observations, a time series or seasonal comparison between observations at these sites and both prior and posterior simulations would provide more clarity. Consider including monthly or seasonal cycle plots at selected sites to demonstrate how well the posterior simulation captures temporal variability. Scatter plots comparing prior vs. observed and posterior vs. observed $NH_3$ concentrations (similar to Figures S4–S6) should be more clearly explained in the main text.

**Response:**

We thank the reviewer for this constructive feedback. In response to your suggestions, we have significantly expanded the validation section of our manuscript with more detailed seasonal comparisons.

The validation utilizes in situ surface $NH_3$ measurements from 13 sites within our study domain (detailed in Table S2), as reported by Pan et al. (2018) for the period from autumn 2015 to summer 2016. For a direct comparison, simulated surface concentrations were extracted from the corresponding model grids and aggregated to seasonal means.

Our analysis confirms a marked improvement with the posterior emissions. The annual mean surface $NH_3$ concentration from the posterior simulation (9.4 $\mu$g m$^{-3}$) is

substantially closer to the observed value (12.7 µg m$^{-3}$) than the prior (6.3 µg m$^{-3}$). As shown in Figure R3.1, this is quantified by a 42% reduction in the model's underestimation (MFB reduced by 0.42) and a higher Index of Agreement (IOA), indicating better spatial consistency.

As requested, we have enhanced the seasonal analysis, with detailed results now presented in Table R3.1. The prior simulation underestimated observed surface NH$_3$ by 37%–79% across seasons. The posterior simulation, while still showing some underestimation, significantly alleviates this bias and demonstrates better performance metrics (lower RMSE and higher IOA), thereby better capturing the seasonal characteristics.

We have also expanded the discussion on potential sources for the remaining discrepancy. This gap may be attributed to several factors: (1) our optimization was constrained by satellite total column densities, which may not perfectly translate to surface-level improvements; (2) a partial temporal mismatch exists between our 2016 simulation and the 2015–2016 observation period; and (3) extreme local conditions at certain sites, such as the exceptionally high summer concentrations at the Yucheng site, are inherently challenging for a regional model to capture.

Table R3.1 Seasonal comparison of simulated and observed surface NH$_3$ concentrations (µg m$^{-3}$) and associated statistical metrics.

| | MAM | JJA | SON | DJF |
|---|---|---|---|---|
| Prior surface NH$_3$ concentration (µg m$^{-3}$) | 5.05 | 7.58 | 6.68 | 6.06 |
| Observed surface NH$_3$ concentration (µg m$^{-3}$) | 11.48 | 17.53 | 12.48 | 9.26 |
| IOA | 0.49 | 0.54 | 0.66 | 0.67 |
| MFB | -0.79 | -0.74 | -0.54 | -0.37 |
| RMSE (µg m$^{-3}$) | 8.29 | 13.06 | 7.64 | 5.76 |
| | MAM | JJA | SON | DJF |

| | | | | |
|---|---|---|---|---|
| Posterior surface NH$_3$ concentration ($\mu$g m$^{-3}$) | 8.86 | 10.10 | 9.13 | 9.44 |
| Observed surface NH$_3$ concentration ($\mu$g m$^{-3}$) | 11.48 | 17.53 | 12.48 | 9.26 |
| IOA | 0.56 | 0.61 | 0.7 | 0.72 |
| MFB | -0.24 | -0.47 | -0.19 | 0.12 |
| RMSE ($\mu$g m$^{-3}$) | 5.86 | 10.63 | 6.13 | 5.08 |

[Figure]

Figure R3.1 Scatter plot comparison of prior (orange) and posterior (blue) simulated seasonal mean surface NH$_3$ concentrations against in situ observations. The solid black line indicates the 1:1 ratio.

***Revision in Section 4.3:***

*A similar improvement is also witnessed in the modeling of surface NH$_3$ concentrations, which were evaluated against in-situ measurements from 13 sites reported by Pan et al. (2018a) for the 2015–2016 period (Table S2). The posterior simulation significantly improves the annual mean, increasing the surface concentration from 6.3 $\mu$g m$^{-3}$ (prior) to 9.4 $\mu$g m$^{-3}$ (posterior), much closer to the observed average of 12.7 $\mu$g m$^{-3}$. As shown in the scatter plot in Figure S7, the posterior simulation alleviates the underestimation at most sites, which is quantified by a 42% reduction in the overall underestimation bias and a clear improvement in the IOA. On*

*a seasonal basis, the posterior emissions also alleviate the large underestimation of the prior simulation across all seasons, though the degree of improvement varies (Table S6). The prior simulation showed significant underestimation in all seasons, with the MFB ranging from -0.37 in winter to -0.79 in spring. The posterior simulation demonstrates a particularly evident improvement in spring, where the MFB reduced from -0.79 to -0.24. While some underestimation remains in summer, the posterior results still show improved performance metrics (e.g., lower RMSE and higher IOA) for all seasons, confirming a better capture of the seasonal characteristics overall. The remaining discrepancy between the posterior simulation and surface observations can be attributed to several factors, such as the spatial representativeness of the surface sites and the accuracy of the secondary inorganic aerosol simulation.*

***Revision in Supplementary:***

*Table S6 Seasonal comparison of simulated and observed surface $NH_3$ concentrations ($\mu g\ m^{-3}$) and associated statistical metrics.*

| | MAM | JJA | SON | DJF |
|---|---|---|---|---|
| *Prior surface $NH_3$ concentration ($\mu g\ m^{-3}$)* | 5.05 | 7.58 | 6.68 | 6.06 |
| *Observed surface $NH_3$ concentration ($\mu g\ m^{-3}$)* | 11.48 | 17.53 | 12.48 | 9.26 |
| *IOA* | 0.49 | 0.54 | 0.66 | 0.67 |
| *MFB* | -0.79 | -0.74 | -0.54 | -0.37 |
| *RMSE ($\mu g\ m^{-3}$)* | 8.29 | 13.06 | 7.64 | 5.76 |
| | MAM | JJA | SON | DJF |
| *Posterior surface $NH_3$ concentration ($\mu g\ m^{-3}$)* | 8.86 | 10.10 | 9.13 | 9.44 |
| *Observed surface $NH_3$ concentration ($\mu g\ m^{-3}$)* | 11.48 | 17.53 | 12.48 | 9.26 |
| *IOA* | 0.56 | 0.61 | 0.7 | 0.72 |

| | | | | |
|---|---|---|---|---|
| *MFB* | *-0.24* | *-0.47* | *-0.19* | *0.12* |
| *RMSE (µg m⁻³)* | *5.86* | *10.63* | *6.13* | *5.08* |

[Figure]

*Figure S7 Scatter plot comparison of prior (orange) and posterior (blue) simulated seasonal mean surface NH₃ concentrations against in situ observations. The solid black line indicates the 1:1 ratio.*

3. **Clarification on Posterior Emission Totals:**

Page 13, L343: Clarify whether the 4.2 Tg emission is derived from the posterior estimate. Given the remaining model–observation gap, this number should be framed as a lower-bound estimate. Please discuss the implications.

**Response:**

Thank you for this insightful suggestion. We have revised the manuscript to provide the requested clarification and context.

We confirm that the 4.2 Tg value represents the total annual $NH_3$ emission derived from our posterior estimate. Following your suggestion, we agree that this value should be framed as a conservative, lower-bound estimate due to the remaining gap between our model results and observations.

Our analysis shows that while the posterior emissions significantly improve the simulation, the resulting annual mean $NH_3$ total column density ($23.7 \times 10^{15}$ molec cm⁻²) still lower than the satellite-retrieved value ($29.0 \times 10^{15}$ molec cm⁻²). The implication of this discrepancy is that the true $NH_3$ emissions in this region may be even higher than our estimate. Fully closing this model-observation gap would likely require further

upward adjustments to the emission inventory. We have added a discussion to the manuscript to reflect this important context.

*Revision in Section 6:*

*The posterior results indicate that the NH$_3$ emission in Eastern China for 2016 amounted to 4.2 Tg.*

*Revision in Section 4.2:*

*In similar years and regions, the discrepancy between the estimates of this study and other studies ranges from 1.0% to 19.6%. The slight discrepancy can be partially explained by our estimate being a conservative lower bound, a consequence of the residual gap remaining with satellite retrieval.*

4. **Sectoral Emission Trends and Inconsistencies:**

Page 8, L216–218 and L223: There seems to be a discrepancy between the statements about non-AGR and AGR emission changes. Figure 5b suggests a spring increase, typically associated with agricultural activity, yet the larger change is attributed to non-AGR sources. Please clarify the partitioning of the emission increase.

**Response:**

We thank the reviewer for this important question. The partitioning of emission changes during spring is indeed complex, and we have expanded the discussion in the manuscript to provide a clearer explanation.

The significant increase in total spring emissions is the net result of adjustments in both agricultural (AGR) and non-agricultural (non-AGR) sectors. Our posterior analysis reveals substantial but spatially heterogeneous changes within the agricultural sector. For example, while AGR emissions in the Henan region were reduced to correct a prior overestimation, emissions in the Yangtze River Delta (YRD) concurrently increased by 242.8 Gg. Despite these regional adjustments, agriculture remains the dominant source in spring, accounting for 84.1% of total posterior emissions.

However, to bridge the large gap between prior simulations and satellite observations, a significant upward adjustment in total emissions was necessary. While annual AGR emissions were adjusted upwards, non-agricultural sources required an even more greater revision, driven by the spatial patterns of NH$_3$ concentrations. Given the high NH$_3$ level observed during spring, a large portion of the seasonal emission increase was attributed to the non-AGR sector to better match the observations.

Key non-AGR sources contributing to this increase include industrial processes (e.g., "ammonia slip" from emission controls), vehicle emissions (a byproduct of three-way catalysts), and waste management (volatilization from landfills and wastewater treatment). Rising spring temperatures can enhance the volatilization rates from several of these sources, leading to considerable emissions. We have incorporated this detailed explanation into the revised manuscript.

*Revision in Section 4.2:*

*The seasonal variations in the posterior emissions is the net result of complex adjustments in both the AGR and non-AGR sectors.*

5. **Loss of High Emission Feature in SON/DJF (Figure 6):**

The high-emission feature at the intersection of Henan, Hebei, and Shandong disappears in SON and DJF seasons. Please explain whether this is due to real seasonal changes or limitations in the model/data.

**Response:**

Thank you for your careful observation. We have revised the manuscript to address this important point.

First, we would like to clarify that the high-concentration feature at the intersection of Henan, Hebei, and Shandong does persist through autumn and winter in both the satellite retrievals and our posterior simulation. The concentration in this region remains significantly higher than in surrounding areas during these seasons. To better visualize this, we have revised Figure 6 with an adjusted color scale, which now clearly shows the high-value center in the posterior simulation, consistent with the satellite observations.

However, we acknowledge that a gap between our posterior simulation and the satellite data still exists in the colder seasons. This residual bias is likely due to methodological limitations. One potential factor is our iterative stopping criterion, which concludes the optimization when the mean error is reduced to 30%. This inherent tolerance allows for a certain level of discrepancy to remain. Another factor is that our optimization was performed monthly, which may introduce inconsistencies when results are aggregated and evaluated on a seasonal scale.

Despite these limitations, the posterior result represents a significant improvement over the prior simulation. We have added an expanded discussion on these

methodological uncertainties to the text to provide a more transparent and robust analysis.

*Revision in Section 4.3:*

*In summary, the posterior simulation improves the agreement between the simulated NH₃ column concentrations and satellite observations in both overall magnitude and spatial distribution, although some deviations remain, particularly in the colder seasons. These can likely be attributed to methodological limitations, such as the inherent tolerance of our 30% iterative stopping criterion and potential inconsistencies from aggregating monthly optimizations to a seasonal scale.*

6. **Surface Bias Reduction (Page 10, L277–278):**

Clarify which observational data were used (in situ surface measurements vs. satellite), whether the bias reduction is spatially averaged, and if supporting plots are available. A comparison showing seasonal variation would strengthen this claim.

**Response:**

Thank you for this constructive suggestion. We have revised the manuscript to provide the requested clarifications.

The validation was performed using in situ surface NH₃ measurements from a network of 13 sites within our study domain, as reported by Pan et al.(2018). This dataset provides seasonal mean concentrations from autumn 2015 to summer 2016. Site details are available in Table S2.

The reduction in bias is observed on both a spatially averaged and a site-by-site basis. The supporting analysis is presented in Figure R3.1, which includes scatter plots comparing both prior and posterior simulations against the observations. These plots clearly demonstrate that the posterior simulation alleviates the underestimation at most sites.

Following your suggestion, we have also strengthened the seasonal comparison. The revised manuscript and its supplement (Table S6) now explicitly compare simulated and observed seasonal NH₃ concentrations. The results confirm that our posterior inventory leads to a more consistent performance and reduced bias across all four seasons, as quantified by improved MFB and IOA metrics.

*Revision in Section 4.3:*

*A similar improvement is also witnessed in the modeling of surface NH₃*

*concentrations, which were evaluated against in-situ measurements from 13 sites reported by Pan et al. (2018a) for the 2015–2016 period (Table S2). The posterior simulation significantly improves the annual mean, increasing the surface concentration from 6.3 μg m⁻³ (prior) to 9.4 μg m⁻³ (posterior), much closer to the observed average of 12.7 μg m⁻³. As shown in the scatter plot in Figure S7, the posterior simulation alleviates the underestimation at most sites, which is quantified by a 42% reduction in the overall underestimation bias and a clear improvement in the IOA. On a seasonal basis, the posterior emissions also alleviate the large underestimation of the prior simulation across all seasons, though the degree of improvement varies (Table S6). The prior simulation showed significant underestimation in all seasons, with the MFB ranging from -0.37 in winter to -0.79 in spring. The posterior simulation demonstrates a particularly evident improvement in spring, where the MFB reduced from -0.79 to -0.24. While some underestimation remains in summer, the posterior results still show improved performance metrics (e.g., lower RMSE and higher IOA) for all seasons, confirming a better capture of the seasonal characteristics overall. The remaining discrepancy between the posterior simulation and surface observations can be attributed to several factors, such as the spatial representativeness of the surface sites and the accuracy of the secondary inorganic aerosol simulation.*

7. Table 3 Description (Page 11, L288–297):

Provide more detail on the meaning of the single values listed in Table 3. What metrics are these? How do they compare across seasons and sectors?

**Response:**

We thank the reviewer for this helpful suggestion. We have revised the manuscript to provide a more detailed description of Table 3.

The values in Table 3 represent the annual mean concentrations of $PM_{2.5}$, $SO_2$, and $NO_2$ derived from the prior simulation, the posterior simulation, and surface observations. The observational data were averaged from 80 monitoring sites across 9 major cities (details in Table S4). The comparison was performed by matching the observed data from each site with the simulated concentration from its corresponding

model grid cell.

Regarding the comparison, both simulations capture the pollutant concentrations reasonably well, with the posterior results showing a clear improvement. This is particularly evident for $SO_2$, where the posterior simulated concentration is much closer to the observed value, reducing the model's previous overestimation by 27%. This improvement is most significant in autumn, as the increased availability of $NH_3$ in our posterior simulation drives more gaseous $SO_2$ into the particle phase. This more detailed description has been added to the main text.

*Revision in Section 4.3:*

*Furthermore, improving the $NH_3$ simulation results in the other simulated air pollutants being closer to observed levels (Table 3). Specifically, we compare the annual mean concentrations of $PM_{2.5}$, $SO_2$, and $NO_2$ from the prior and posterior simulations against surface observations averaged from 80 monitoring sites across 9 major cities (Table S4).*

*A similar improvement is also observed for $SO_2$, where the posterior simulated concentration (6.8 ppbv) better matches the observed value (6.5 ppbv), reducing the model's previous overestimation by 27%. This improvement is most significant in autumn. The successful capture of air pollutants highlights a significant improvement in the $NH_3$ emission inventory for Eastern China.*

8. **Public Health and $PM_{2.5}$ Impact:**

Page 11, L306: Quantify the reduction in $PM_{2.5}$ (1.5–5.7 μg/m³) as a percentage of baseline concentrations to help contextualize the health impact.

**Response:**

Thank you for this detailed suggestion. We agree that providing percentages helps to contextualize the impact of $NH_3$ emission reductions. We have revised the relevant section of the manuscript to incorporate these values.

*Revision in Section 5:*

*Figure 7 illustrates that reducing $NH_3$ emissions by 30%–60% can decrease the seasonal $PM_{2.5}$ concentrations by 1.5–5.7 μg m$^{-3}$ (2.0%–7.2%) averaged for Eastern China in winter, mainly due to the reduction in SIA.*

**Minor Comments**

• **Page 4, L97:** Surface data usage should be mentioned in the abstract for completeness.

**Response:** Thank you for the suggestion. We have revised the abstract accordingly.

*Revision in Abstract:*

*The optimized NH$_3$ emission significantly improved the simulation of both total column and surface NH$_3$ concentrations, with improvements in magnitude (31%–42%) and variations (17%–55%).*

• **Page 7, L164:** Briefly explain the extreme values of IOA and MFB to aid reader interpretation.

**Response:** We appreciate this helpful comment and have expanded the text to briefly explain the interpretation of these metrics for the reader.

*Revision in Section 3:*

*The IOA quantifies the overall model skill, where a value of 1 indicates a perfect match and 0 denotes complete disagreement. The MFB diagnoses systematic model bias, where positive values indicate overestimation, negative values indicate underestimation, and 0 signifies no average bias.*

• **Page 7, L180–184:** Confirm whether these lines are referencing Figure 6.

**Response:** Thank you for pointing this out. We have revised the text to add the explicit reference to Figure 6.

*Revision in Section 3:*

*As illustrated in Figure 6, satellite-based observations reveal that the spatial high-value areas of NH$_3$ column are located at the junction of Henan, Shandong, and Hebei provinces. In contrast, the prior modeling results show that NH$_3$ column densities are more concentrated in Henan. This indicates a clear discrepancy in the spatial distribution of NH$_3$ column densities between the prior simulations and the observations.*

• **Page 8, L201:** Remove extra period after "Table 2."

**Response:** Thank you for pointing this out. The text has been corrected.

• **Page 9, L239:** Correct grammar: "based on both top-down and bottom-up approaches."

**Response:** Thank you for the suggestion. The text has been revised as recommended.

*Revision in Section 4.2:*

*Overall, the estimated NH₃ emission in this study is comparable to the estimates of the other studies based on both "top-down" and "bottom-up" approaches.*

• **Page 10, L265:** Suggest adding the prior result value alongside the percentage difference for clarity.

**Response:** Thank you for this helpful suggestion. We agree that adding the prior result value improves clarity and have revised the sentence accordingly.

*Revision in Section 4.3:*

*The annual mean simulated NH₃ total column density improved from the prior result of $17.4 \times 10^{15}$ molec cm⁻² to a posterior value of $23.7 \times 10^{15}$ molec cm⁻², with an increase of 35.9%, and is closer to the observed value of $29.0 \times 10^{15}$ molec cm⁻².*

• **Page 11, L281:** Reword the statement to reflect partial improvement in posterior vs. surface observations.

**Response:** Accepted. The relevant section of the manuscript has been updated to provide further clarification.

*Revision in Section 4.3:*

*On a seasonal basis, the posterior emissions also alleviate the large underestimation of the prior simulation across all seasons, though the degree of improvement varies (Table S6). The prior simulation showed significant underestimation in all seasons, with the MFB ranging from -0.37 in winter to -0.79 in spring. The posterior simulation demonstrates a particularly evident improvement in spring, where the MFB reduced from -0.79 to -0.24. While some underestimation remains in summer, the posterior results still show improved performance metrics (e.g., lower RMSE and higher IOA) for all seasons, confirming a better capture of the seasonal characteristics overall.*

• **Page 13, L342:** Add missing period.

**Response:** Thank you for noting this. The correction has been made.

*Revision in Section 6:*

*In this study, we used IASI satellite products and an iterative algorithm with the WRF-Chem model to optimize the bottom-up NH₃ emission inventory for Eastern China*

*and further assessed the impacts of NH₃ emission reductions from different sources on PM₂.₅ concentrations.*

**• Page 13, L348:** Consider providing a geophysical or socioeconomic explanation (e.g., dense agriculture and livestock) for high emissions at the provincial intersection.

**Response:**

Thank you for this valuable suggestion. We have expanded the discussion in the manuscript to provide a geophysical and socioeconomic explanation for this high-emission hotspot, as requested.

The high NH₃ emissions at the intersection of these provinces are driven by two primary factors. First, this region is part of the North China Plain, characterized by intensive agriculture (both crop production and animal husbandry) and significant industrial activity, leading to exceptionally high emission intensity.

Second, topography plays a crucial role. The region is bordered by the Taihang and Yanshan Mountains, forming a semi-enclosed plain. This topography can obstruct the dispersal of air pollutants under certain meteorological conditions, leading to their accumulation in the piedmont area and the formation of a persistent high-concentration center.

*Revision in Section 6:*

*Spatially, the region with the highest NH₃ emissions was located at the intersection of Henan, Hebei, and Shandong provinces. This is attributed to a combination of high emission intensity from dense agricultural and industrial activities and topographical effects that hinder the dispersal of pollutants.*

**Suggestions for Figures**

• **Figure 2:** Add a row showing differences (posterior – prior) for better visualization of emission changes by sector.

**Response:** Thank you for this helpful suggestion. We have added a figure in the Supplementary to express the spatial difference in emissions by sector before and after optimization. Our iterative algorithm optimizes emissions on a regional basis to best match the satellite's observed spatial patterns, rather than applying a uniform scaling factor across the domain. This methodological approach results in heterogeneous adjustments, leading to the significant spatial differences observed between provinces, with increases in some areas and decreases in others to achieve an optimal overall fit.

*Revision in Supplementary:*

[Figure]

*Figure S4. Spatial distribution of the difference in NH₃ emissions (Posterior − Prior) for AGR and non-AGR sources.*

• **Figure 3:** Clarify what the red box highlights; also discuss large underestimates in Shandong and BTH.

**Response:**

Thank you for these suggestions. We have revised the manuscript to improve the clarity of Figure 3 and to expand the discussion.

First, we have clarified the meaning of the red box in the figure caption. The box highlights the range we consider to represent good model performance, specifically where the RMSE is less than 10 ($\times 10^{15}$ molec cm⁻²) and the simulated-to-observed ratio is between 0.7 and 1.3 (i.e., within a ±30% margin).

Second, regarding the underestimation in certain regions, our analysis confirms that this residual bias is most pronounced during the autumn season. We have added a discussion attributing this remaining discrepancy to the inherent limitations of our

inversion methodology. Potential factors include our iterative stopping criterion (a 30% error tolerance) and inconsistencies arising from aggregating monthly optimizations to a seasonal scale. This discussion provides a more transparent assessment of our results.

*Revision in Figure 3:*

**Figure 3.** *Scatter plots of the prior and posterior $NH_3$ total column data versus IASI retrievals. Each point represents prior (or posterior) data for a specific season and a specific region. Circles, triangles, rhombuses, and rectangles correspond to the BTH, Henan, Shandong, and YRD regions, respectively. Orange and blue markers represent a prior and a posterior data, respectively. The red box indicates the performance area, with a model error within ±30% and an RMSE below 10(×1015 molec cm-2).*

*Revision in Section 4.3:*

*In summary, the posterior simulation improves the agreement between the simulated $NH_3$ column concentrations and satellite observations in both overall magnitude and spatial distribution, although some deviations remain, particularly in the colder seasons. These can likely be attributed to methodological limitations, such as the inherent tolerance of our 30% iterative stopping criterion and potential inconsistencies from aggregating monthly optimizations to a seasonal scale.*

• **Figure 6:** Add units to the color bar for clarity.

**Response:** Thank you for the suggestion. The units have been added to the color bar in Figure 6.

• **Figure S2:** Increase font size on color bar units.

**Response:** Thank you for the comment. We have increased the font size on the color bar units in Figure S2 as requested.

**References**

• Please ensure that all relevant IASI-based ammonia studies are cited appropriately to position your work in the broader context of satellite-based $NH_3$ retrievals and applications.

**Response:** We appreciate this guidance. To better position our study within the broader context of the field, we have incorporated citations to several additional relevant IASI-

based $NH_3$ studies in the revised manuscript.

**References**

Pan, Y., Tian, S., Zhao, Y., Zhang, L., Zhu, X., Gao, J., Huang, W., Zhou, Y., Song, Y., Zhang, Q., and Wang, Y.: Identifying Ammonia Hotspots in China Using a National Observation Network, Environ Sci Technol, 52, 3926–3934, https://doi.org/10.1021/acs.est.7b05235, 2018.

Qu, Z., Henze, D. K., Capps, S. L., Wang, Y., Xu, X., Wang, J., and Keller, M.: Monthly top-down $NO_x$ emissions for China (2005–2012): A hybrid inversion method and trend analysis, JGR Atmospheres, 122, 4600–4625, https://doi.org/10.1002/2016JD025852, 2017.

Xia, J., Zhou, Y., Fang, L., Qi, Y., Li, D., Liao, H., and Jin, J.: South Asia ammonia emission inversion through assimilating IASI observations, https://doi.org/10.5194/egusphere-2024-3938, 20 January 2025.

Xu, M., Jin, J., Wang, G., Segers, A., Deng, T., and Lin, H. X.: Machine learning based bias correction for numerical chemical transport models, Atmospheric Environment, 248, 118022, https://doi.org/10.1016/j.atmosenv.2020.118022, 2021.